# In vivo interaction screening reveals liver-derived constraints to metastasis

Costanza Borrelli[1], Morgan Roberts[1], Davide Eletto[1], Marie-Didiée Hussherr[1], Hassan Fazilaty[2], Tomas Valenta[2,3], Atefeh Lafzi[1], Jonas A. Kretz[1], Elena Guido Vinzoni[1], Andromachi Karakatsani[1], Srivathsan Adivarahan[1], Ardian Mannhart[1], Shoichiro Kimura[1], Ab Meijs[1], Farah Baccouche Mhamedi[1], Ilhan E. Acar[1], Kristina Handler[1], Xenia Ficht[1], Randall J. Platt[1], Salvatore Piscuoglio[4,5] & Andreas E. Moor[1✉]

It is estimated that only 0.02% of disseminated tumour cells are able to seed overt metastases[1]. While this suggests the presence of environmental constraints to metastatic seeding, the landscape of host factors controlling this process remains largely unclear. Here, combining transposon technology[2] and fluorescence niche labelling[3], we developed an in vivo CRISPR activation screen to systematically investigate the interactions between hepatocytes and metastatic cells. We identify plexin B2 as a critical host-derived regulator of liver colonization in colorectal and pancreatic cancer and melanoma syngeneic mouse models. We dissect a mechanism through which plexin B2 interacts with class IV semaphorins on tumour cells, leading to KLF4 upregulation and thereby promoting the acquisition of epithelial traits. Our results highlight the essential role of signals from the liver parenchyma for the seeding of disseminated tumour cells before the establishment of a growth-promoting niche. Our findings further suggest that epithelialization is required for the adaptation of CRC metastases to their new tissue environment. Blocking the plexin-B2–semaphorin axis abolishes metastatic colonization of the liver and therefore represents a therapeutic strategy for the prevention of hepatic metastases. Finally, our screening approach, which evaluates host-derived extrinsic signals rather than tumour-intrinsic factors for their ability to promote metastatic seeding, is broadly applicable and lays a framework for the screening of environmental constraints to metastasis in other organs and cancer types.

The importance of microenvironmental conditions of the host organ for metastatic seeding has long been recognized. The seminal 'seed and soil' hypothesis postulates that metastatic cells will seed and colonize only favourable environments[4]. However, we still lack a comprehensive overview of the signals from the metastasis-accepting organs that promote or suppress the establishment of secondary tumours. Indeed, retrospective analysis may identify factors that allow metastases to thrive long term, but does not capture early events of metastatic seeding, and cannot distinguish between cellular interactions that are cause or consequence of metastatic outgrowth. Here we devised a screening approach for functional testing of cell–cell interactions in vivo, aimed at the identification of host-derived factors that determine the fate of disseminated tumour cells (DTCs) at the time of seeding. We used it to interrogate which hepatocyte-derived signals promote or suppress seeding of colorectal cancer (CRC) liver metastases and identify plexin B2 as a crucial regulator of liver colonization.

## Screening interactions in a mosaic liver

Hepatocytes constitute 60% of the liver by cell number and 80% by mass[5]. We therefore hypothesized that early interactions with these cells might influence the ability of extravasated DTCs to seed metastases. To test this, we developed an experimental strategy for pooled perturbation of hepatocyte–tumour interactions during seeding (Fig. 1a). First, hundreds of genes are stably overexpressed in hepatocytes using CRISPR-mediated transcriptional activation (CRISPR-a), resulting in a 'mosaic liver' containing multiple perturbed environments; then, tumour cells are delivered to the liver by intrasplenic injection. We assumed that, after cancer inoculation, DTCs interacting with hepatocytes overexpressing a seeding-promoting factor would seed and grow, while DTCs interacting with hepatocytes overexpressing a suppressing factor would fail to form metastases. The effect of a perturbation on seeding can therefore be inferred by its enrichment in metastatic or non-metastatic areas, indicating a

[1]Department of Biosystems Science and Engineering, ETH Zurich, Basel, Switzerland. [2]Department of Molecular Life Sciences, University of Zurich, Zurich, Switzerland. [3]Laboratory of Cell and Developmental Biology, Institute of Molecular Genetics of the Czech Academy of Sciences, Prague, Czech Republic. [4]IRCCS Humanitas Research Hospital, Milan, Italy. [5]Institute of Medical Genetics and Pathology, University Hospital Basel, Basel, Switzerland. ✉e-mail: andreas.moor@bsse.ethz.ch

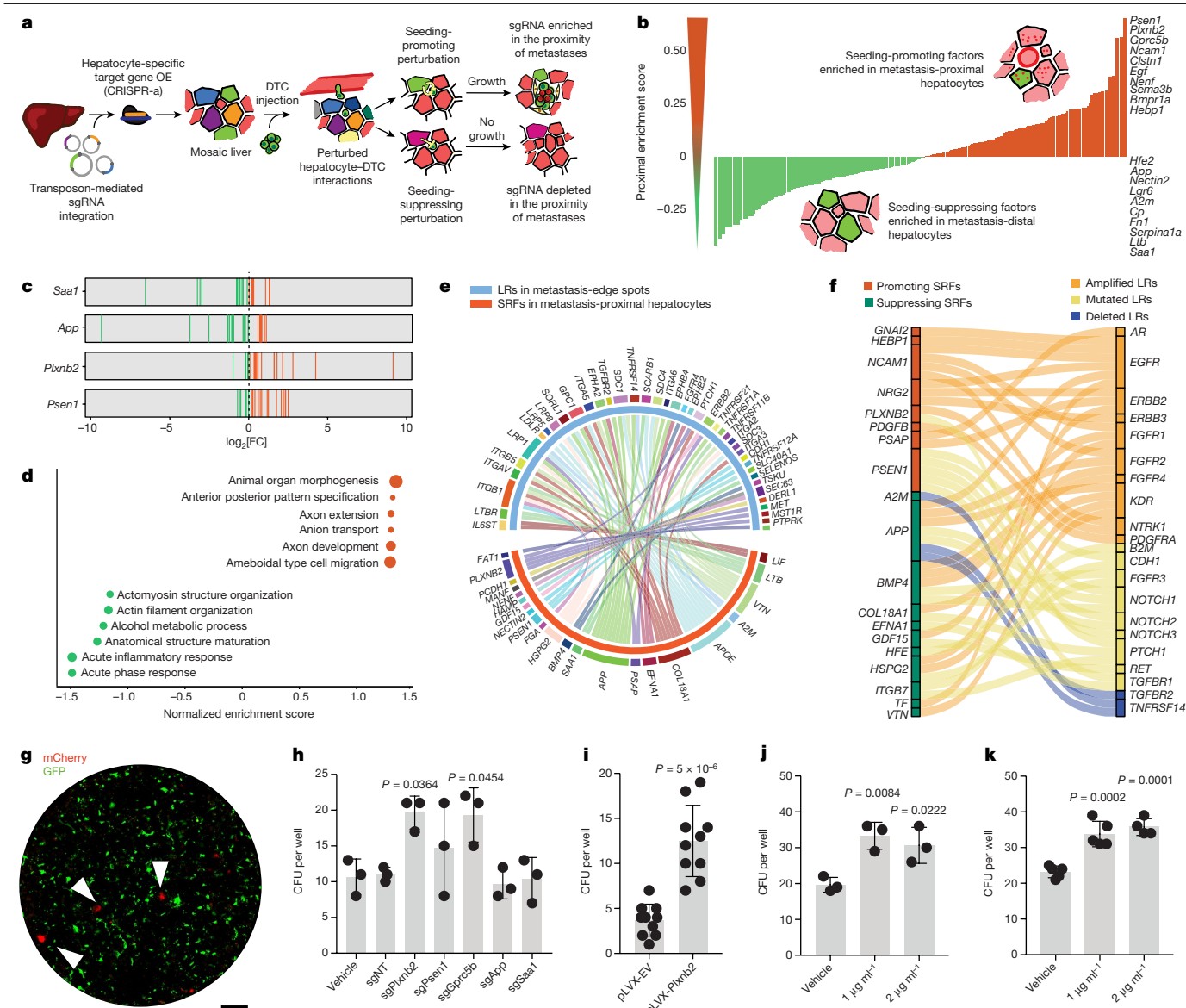

**Fig. 1 | Screening tumour–hepatocyte interactions in a mosaic liver.**
**a**, Schematic of the screen. DTCs interact with hepatocytes harbouring seeding-promoting or seeding-suppressing perturbations, influencing local metastatic outgrowth and therefore sgRNA enrichment in metastasis-proximal hepatocytes. **b**, Genes ranked by proximal enrichment score. Seeding-promoting factors are enriched in GFP⁺mCherry⁺ hepatocytes (red). Suppressing factors are enriched in GFP⁺mCherry⁻ hepatocytes (green). Top-scoring SRFs are listed on the right. **c**, The log-transformed fold change (FC) of individual sgRNAs (vertical lines) for two suppressing and two promoting factors across all mice ($n = 7$) and library batches ($n = 3$). **d**, GSEA of SRFs. The dot size indicates the gene set size. **e**, Interaction analysis in a human CRC liver metastasis. 22 SRFs expressed by metastasis-proximal hepatocytes (orange) are predicted to interact with LRs on tumour cells at the metastatic leading edge (turquoise). **f**, Predicted

interactions between SRFs and LRs that are frequently mutated in liver metastases. CNV status is shown in orange (amplified), yellow (mutation) or blue (deleted). **g**, Representative fluorescence micrograph showing co-culture of *Alb-cre;dCas9-SPH* hepatocytes overexpressing SRFs (GFP⁺) and AKPS^sLP-mCherry^ colonies (mCherry⁺, arrowheads). Scale bar, 100 µm. **h–k**, The CFU per well of AKPS^sLP-mCherry^ organoids co-cultured with AML12^dCas9-SPH^ cells overexpressing SRFs ($n = 3$ wells; **h**), AKPS^sLP-mCherry^ organoids co-cultured with AML12 cells overexpressing *Plxnb2* ($n = 10$ wells; **i**), AKPS^sLP-mCherry^ organoids treated with rmPlexin B2 ($n = 3$ wells; **j**) and PDOs treated with rhPlexin B2 ($n = 5$ wells; **k**). Statistical analysis was performed using ordinary one-way analysis of variance (ANOVA) with Dunnet's correction for multiple testing (**h–j**) or two-tailed unpaired *t*-tests (**k**). Data are mean ± s.d. For **b–d**, results are pooled from three independent experiments.

seeding-promoting or seeding-suppressing effect, respectively (Fig. 1a). Thus, even if perturbed seeding events are not directly observed, their outcome can be retrospectively assessed by the presence of a metastasis.

To achieve hepatocyte-specific CRISPR-a in vivo, *dCas9-SunTag-p65-HSF1*[6] mice crossed with albumin-Cre mice[7] (*Alb-cre;dCas9-SPH*) were subjected to hydrodynamic tail vein injection of transposon plasmids containing sgRNAs and Sleeping Beauty transposase (SB100X), which mediates stable integration of exogenous DNA in

mouse hepatocytes[2]. Indeed, injection of sgRNAs targeting glutamine synthetase resulted in increased *Glul* transcript expression and ectopic glutamine synthetase immunoreactivity (Extended Data Fig. 1a,b).

Molecular interactions between surface proteins on tumour cells and hepatocytes are probably among the first events to occur after DTC extravasation into the space of Disse. We therefore designed a library of sgRNAs targeting ligands and receptors (LRs) expressed by quiescent and regenerating mouse hepatocytes[8]. Moreover, we

included components of the Hippo signalling pathway, the activation of which suppresses melanoma metastases[9]; claudins recently implicated in CRC dissemination[10]; orphan class C G-protein-coupled receptors (GPCRs) with an unknown role in metastatic seeding; and 100 safe-targeting sgRNAs as negative controls[11] (Extended Data Fig. 1c). The library (3 sgRNAs per gene, 997 sgRNAs in total; Supplementary Table 1) was cloned into transposon vectors containing a *CMV-GFP* reporter, and co-injected with SB100X into *AlbCre;dCas9-SPH* mice at a concentration resulting in 0.5% GFP+ hepatocytes (Extended Data Fig. 1d,e). One week after injection, we isolated CD31−CD45−GFP+ hepatocytes using fluorescence-activated cell sorting (FACS) and performed targeted sgRNA amplification from genomic DNA followed by sequencing. Correlation analysis of sgRNA abundances in the pre- and post-injection library revealed stable sgRNA distribution and high library retention, indicating no loss of perturbation diversity (Extended Data Fig. 1f). Notably, the introduction of sgRNAs into non-proliferative hepatocytes, rather than into tumour cells, ensures unaltered library distribution throughout the experiment, avoids bottleneck effects arising from poor grafting of tumour cells in vivo and prevents library skewing towards perturbations that confer a proliferative advantage. Moreover, the elevated number of hepatocytes in the adult mouse liver (150 million)[12] enables screening of 997 sgRNAs library at high coverage (750×), while also ensuring a multiplicity of infection (MOI) lower than 1 (Extended Data Fig. 1g).

Our approach depends on the assumption that a DTC interacting with a hepatocyte with a seeding-promoting perturbation would have increased chances of survival, and the corresponding sgRNA would therefore be enriched in metastatic areas. To record the proximity of perturbed hepatocytes to metastases, we introduced the sLP-mCherry niche-labelling system[3] in *Villin-creER[T2];APC[fl/fl];Trp53[fl/fl];Kras[G12D];Smad4[KO]* (AKPS) organoids (AKPS[sLP-mCherry]) and observed efficient perimetastatic hepatocyte mCherry labelling in vivo after intrasplenic injection (Extended Data Fig. 1h–j). Thus, hepatocytes with successful transposon insertion (GFP+) can be separated by fluorescence-activated cell sorting (FACS) as either proximal to metastasis (metastasis-proximal, mCherry+GFP+) or distant from metastases (metastasis-distal, mCherry−GFP+) (Extended Data Fig. 1k).

We conducted three screening experiments with independently amplified sgRNA library batches (Extended Data Fig. 2a). In total, 7 *Alb-cre;dCas9-SPH* mice and 5 non-Cre littermate controls were injected with sgRNA library and intrasplenically injected with AKPS[sLP-mCherry] organoids. Metastases were allowed to grow for 2 weeks, after which metastasis-proximal and metastasis-distal hepatocytes were isolated using FACS. The amount of sorted cells across all experiments and mice resulted in cumulative coverage of 1,000× for *Alb-cre;dCas9-SPH* mice and 500× for the littermate controls (Extended Data Fig. 2b,c). We scored genes based on the enrichment of their inferring sgRNAs in metastasis-proximal versus metastasis-distal hepatocytes (Fig. 1b). The top-scoring differentially enriched perturbations were consistent across individual mice and library batches (Fig. 1c), indicating high robustness of our screening strategy, and were not differentially enriched in non-Cre littermates (Extended Data Fig. 2d). sgRNAs strongly enriched in metastasis-distal hepatocytes induced overexpression (OE) of the tumour necrosis factor family cytokine lymphotoxin-β (*Ltb*), as well as several genes involved in acute phase response such as serum amyloid 1 (*Saa1*), amyloid precursor protein (*App*), ceruloplasmin (*Cp*) and α2 macroglobulin (*A2m*) (Fig. 1d). The depletion of sgRNAs targeting these genes in metastatic areas suggests that their upregulation prevents seeding of DTCs, possibly by inducing local immune activation. Indeed, *Saa1* was suggested to attract macrophages to the tumour invasive front[13], whereas amyloid protein (APP) deposition recruits neutrophils in several cancers[14]. Conversely, sgRNAs enriched in the proximity of metastases induced OE of epithelial growth factor (*Egf*), a known driver of metastatic CRC[15], as well as other regulators of morphogenesis (*Gpc3* and *Psen1*),

and several genes that are involved in axon guidance such as *Plxnb2*, *Nrp2*, *Sema3b*, *Ncam1* and *Nenf* (Fig. 1b–d). Our screen therefore implicates neurotrophic factors as promoters of metastatic seeding in the liver. This is consistent with reports of DTCs hijacking axonal morphogenesis pathways to interact with endothelial cells[16,17]; however, their role in tumour–hepatocyte interactions has not been explored.

We next sought to cross-validate the results of our screen in transcriptional and mutational data of human liver metastases. We first tested whether any of the identified seeding-regulating factors (SRFs; 62 genes, top and bottom decile in GFP+mCherry+ hepatocytes) were predicted to engage in tumour–hepatocyte interaction at the metastatic edge. We therefore generated spatial transcriptomics data of a human CRC liver metastasis with an extensive tumour–liver border, and performed cell type deconvolution using published single-cell RNA-sequencing (scRNA-seq) datasets[18,19] (Extended Data Fig. 2e). We then predicted LR pairs between metastasis-proximal hepatocytes and the tumour edge that potentially regulate expression changes between the tumour edge and core (Methods and Extended Data Fig. 2f,g). Among the 109 active LR pairs, 22 involved SRFs, including the chemoattractants *App*, *Saa1*, *Cp* and *Ltb* and the axon guidance molecules *Plxnb2*, *Nenf* and *Nectin2* (Fig. 1e). Next, we extracted LRs from a published dataset of genomic alterations enriched in liver metastases compared with in matched primary tumours[20] and identified their interaction partners expressed by hepatocytes (Methods and Extended Data Fig. 3a). We found that 21 LRs mutated in liver metastases potentially interact with SRFs, with deleted LRs mainly predicted to interact with suppressing factors, and amplified LRs with promoting factors, possibly suggesting a selection of these interactions (Fig. 1f). Finally, SRFs are also predicted to interact with LR-encoding differentially expressed genes in liver metastases compared with in matched primary CRC in two independent scRNA-seq datasets[21,22] (Extended Data Fig. 2h). Together with our screening results, these analyses demonstrate the ability of our screening platform to capture disease-relevant interactions, and implicate hepatocyte-derived chemoattractants and axon guidance cues as regulators of metastatic seeding in the liver.

To test the direct effect of SRFs on cancer growth, we devised a small interaction screen based on co-culture of hepatocytes and cancer cells. sgRNAs targeting SRFs were transfected in an arrayed manner in primary hepatocytes isolated from *Alb-cre;dCas9-SPH* mice, or in immortalized mouse hepatocytes (AML12) stably expressing dCas9-SPH (Extended Data Fig. 3b). AKPS[sLP-mCherry] organoids dissociated into single cells were then sparsely seeded onto the hepatocyte monolayer and allowed to grow for 5 days before colony counting (colony forming units, CFU) (Fig. 1g). OE of *App* and *Saa1* did not result in significantly altered CFU values with respect to the non-targeting sgRNA (sgNT) or untransfected controls, suggesting that their effect on seeding in vivo might be mediated by local recruitment of a third cell type (Fig. 1h and Extended Data Fig. 3c). Conversely, we observed increased CFU values after *Plxnb2*, *Psen1* and *Gprc5b* OE, indicating their direct involvement in the interactions between hepatocytes and tumour cells (Fig. 1h). In particular, both CRISPR-a-mediated and lentiviral *Plxnb2* OE in AML12 cultures had a very potent effect on AKPS seeding (Fig. 1i and Extended Data Fig. 3d–g). Moreover, addition of recombinant mouse plexin B2 ectodomain (rmPlexin B2) on AKPS single cells significantly increased CFUs both in the presence and absence of hepatocytes, suggesting that plexin B2 directly binds to tumour cells (Fig. 1j and Extended Data Fig. 3h). Notably, these results could be recapitulated by adding recombinant human plexin B2 (rhPlexin B2) on patient-derived CRC organoids (PDOs) with or without immortalized human hepatocytes (PTA-5565) (Fig. 1k and Extended Data Fig. 3i,j). Together with our transcriptomic and mutational analysis, these results implicate plexin B2 in direct interactions between hepatocytes and metastatic cells, which we next sought to investigate in vivo.

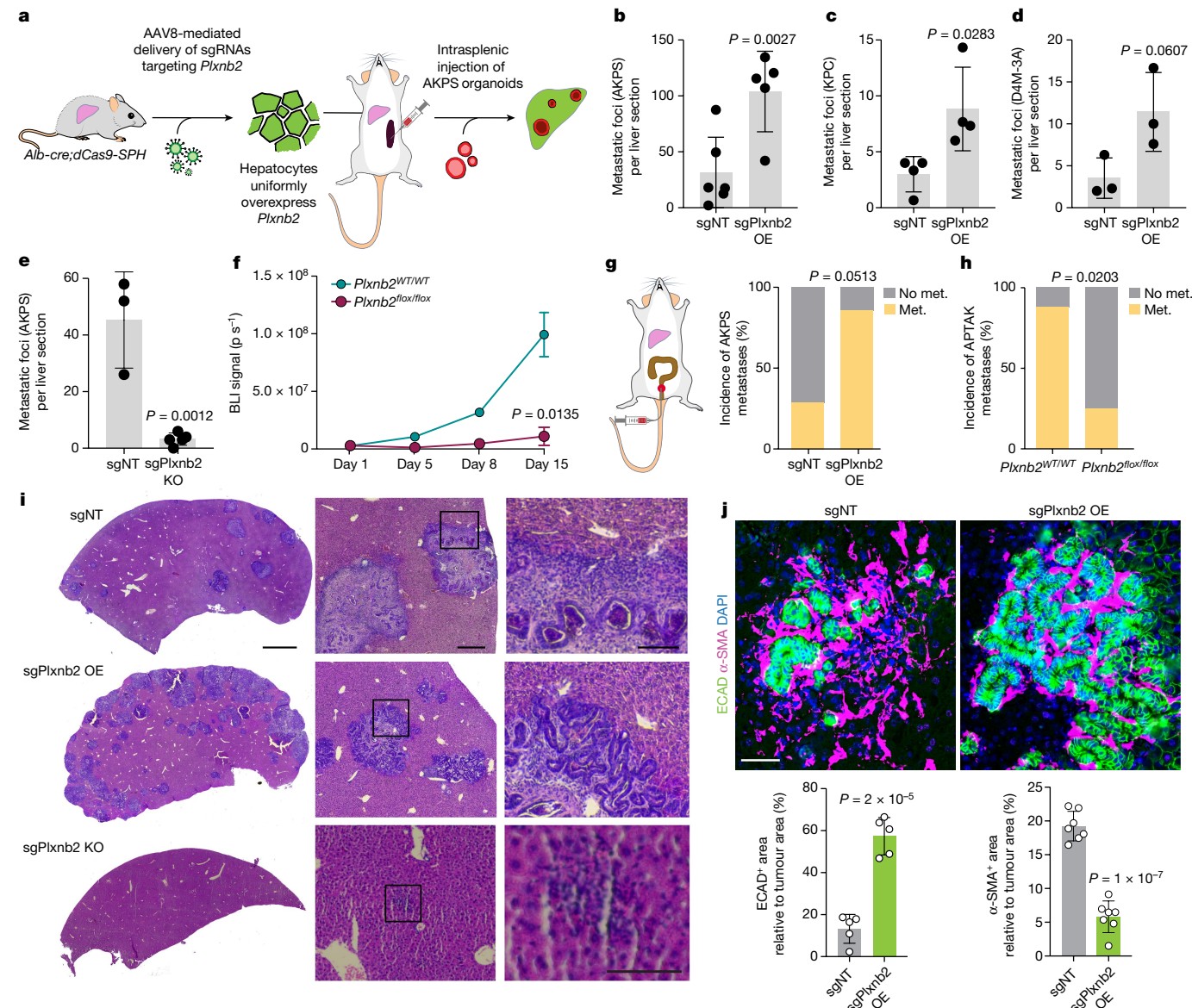

**Fig. 2 | Hepatocyte-derived plexin B2 is required for liver colonization.**
**a,** AAV8-U6-sgPlxnb2OE-EF1a-eGFP injection induces dCas9-SPH-mediated
*Plxnb2* OE in hepatocytes, followed by intrasplenic injection of AKPS organoids.
**b–d,** Metastatic foci per liver section 2 weeks after injection of AKPS organoids
in *Plxnb2*-OE (*n* = 5) or control mice (*n* = 6) (results are pooled from two
independent experiments; **b**), or KPC cells (*n* = 4 per group; **c**) or D4M-3A cells
(*n* = 3 per group; **d**) in *Plxnb2*-OE or control mice. **e,** Metastatic foci per liver
section 2 weeks after injection of AKPS organoids in *Plxnb2* KO (*n* = 4) or control
(*n* = 3) mice. **f,** The BLI signal over time in *Plxnb2*[flox/flox] (*n* = 3) or *Plxnb2*[WT/WT]
(*n* = 4) mice injected with AAV8-Alb-cre-moxGFP and subsequently with
AKPS[luciferase;zsGreen] organoids. Statistical analysis was performed using two-tailed
Fisher's exact tests. **g,h,** The incidence of spontaneous liver metastases (met.)
in *Plxnb2*-OE versus control mice bearing primary AKPS tumours (*n* = 7 mice

per group; **g**), or in *Plxnb2*-KO versus control mice bearing primary APTAK
tumours (*n* = 7 mice per group; **h**). Statistical analysis was performed using
two-tailed Fisher's exact tests. **i,** Representative haematoxylin and eosin (H&E)
staining of AKPS metastases in *Plxnb2*-OE, *Plxnb2*-KO and control livers. Insets:
the metastasis leading edge. Scale bars, 500 μm (left), 200 μm (middle) and
50 μm (right). **j,** Representative fluorescence micrograph and quantification
of α-SMA (magenta) and ECAD (green) immunoreactivity in AKPS metastases in
*Plxnb2*-OE and control livers. DAPI (nuclear counterstain) is shown in blue.
Scale bar, 100 μm. Data are the mean ± s.d. area relative to the tumour area.
The dots represent individual liver metastases (*n* = 5 (α-SMA) and *n* = 6 (ECAD))
pooled from 2 mice per group. For **b–e** and **j**, statistical analysis was performed
using two-tailed unpaired *t*-tests. Data are mean ± s.d. (**b–f** and **j**).

## Plexin B2 is required for liver seeding

Plexin B2 is widely expressed in epithelial cells of most mouse tissues,
where it localizes to the basolateral membrane[23]. Although its func-
tions are mostly characterized in neural development[24], the phyloge-
netic emergence of the plexin family predates the appearance of the
nervous system[25], and recent studies have unravelled roles of plexin
B2 in several tissue contexts[26,27]. We used adeno-associated virus 8
(AAV8) to broadly deliver sgRNAs targeting *Plxnb2* to hepatocytes of

*Alb-cre;dCas9-SPH* mice in vivo (Extended Data Fig. 4a–e). Consistent
with the results of our screen, *Plxnb2* OE induced a threefold increase
in metastatic foci after intrasplenic injection of AKPS organoids
(Fig. 2a,b). Notably, *Plxnb2* OE also promoted grafting of syngeneic
pancreatic ductal adenocarcinoma (*Ptf1a-cre;Kras*[G12D/+]*;Trp53*[flox/+], KPC)
and melanoma cells (*Tyr-creER;BRaf*[CA]*;Pten*[lox/lox], D4M-3A), suggesting
that its seeding-promoting effect also applies to other cancers that
frequently metastasize to the liver (Fig. 2c,d). To test the requirement
of hepatocyte-derived plexin B2 for metastatic seeding, we performed

hepatocyte-specific ablation of plexin B2 through AAV8-mediated delivery of *Plxnb2*-targeting sgRNAs in *Alb-cre;Cas9* mice or *Alb-cre* to *Plxnb2*^flox/flox^ mice (*Plxnb2* KO; Extended Data Fig. 4d,f,g). In both of the experimental models, loss of plexin B2 almost completely prevented metastatic outgrowth of AKPS liver metastases, as revealed by histological analysis and in vivo bioluminescence imaging (BLI; Fig. 2e,f and Extended Data Fig. 4h). We further assessed the influence of hepatocyte-derived plexin B2 on seeding of spontaneous liver metastases from colorectal tumours generated by colonoscopy-guided submucosal injection of organoids (Extended Data Fig. 5a). Notably, 86% of the *Plxnb2*-OE mice developed spontaneous liver metastases 8 weeks after orthotopic tumour inoculation (AKPS organoids), while only 29% of control littermates did (Fig. 2g and Extended Data Fig. 5b,c). Conversely, *Plxnb2* deletion significantly decreased the incidence and numbers of spontaneous liver metastases of *Apc*^flox/flox^; *Trp53*^flox/flox^;*Tgfbr2*^flox/flox^;*Kras*^G12D^;*Akt1*^myristoilated^ (APTAK) organoids, which exhibit a high metastatic potential[28] (Fig. 2h and Extended Data Fig. 5d–f).

At steady state, plexin B2 is widely expressed by hepatocytes—with higher expression in portal areas (Extended Data Fig. 5g,h). Notably, although plexin B2 immunoreactivity is higher in peritumoral hepatocytes (Extended Data Fig. 5i,j), *Plxnb2* expression is unaltered in the livers of mice bearing AKPS colon tumours, suggesting that *Plxnb2* is not upregulated by primary-tumour-secreted factors nor systemic effects but, rather, due to a local response to the presence of liver metastases (Extended Data Fig. 5k). Notably, ablation of *Plxnb2* 5 days after intrasplenic AKPS organoid injection delayed but did not prevent metastasis formation, indicating that the presence of plexin B2 on peritumoral hepatocytes is required for metastatic seeding, but not to sustain growth (Extended Data Fig. 5l,m).

Histological analysis revealed that hepatic plexin B2 levels substantially change the morphology and microenvironment of AKPS metastases. The few remaining lesions in *Plxnb2*-KO livers exhibited cellular disarray and often lacked gland structures, suggesting that the absence of plexin B2 impairs epithelial morphogenesis in liver metastases (Fig. 2i). Instead, lesions in *Plxnb2*-OE livers consisted mostly of epithelial cells, contained fewer fibroblasts positive for α-smooth muscle actin (α-SMA) and periostin (POSTN) that instead surround AKPS liver metastases in wild-type (WT) livers, and exhibited an extensive CD146^+ vascular network (Fig. 2i,j and Extended Data Fig. 6a,b). Importantly, α-SMA and CD146 immunoreactivity, as well as transcriptionally predicted cellular composition, were unaltered before tumour inoculation, indicating that increased metastatic seeding was not due to alterations in the liver environment induced by *Plxnb2* OE but, rather, to direct tumour–hepatocyte interactions (Extended Data Fig. 6c–f).

## Plexin B2 induces epithelialization

We next sought to investigate the mechanism by which plexin B2 controls liver colonization. scRNA-seq revealed that, in AKPS organoids, 2 h treatment with rmPlexin B2 was sufficient to induce a shift towards a more proliferative cell population, with increased expression gene sets related to MAPK and JNK signalling, cell junction assembly and morphogenesis (Fig. 3a). Consistent with an induction of proliferation, rmPlexin B2 increased frequencies of EdU^+ cells in AKPS organoids and PDOs (Extended Data Fig. 7a,b). Lesions in *Plxnb2*-OE livers further contained a higher density of Ki-67^+ epithelial cells, indicating a proliferative advantage in vivo, as well as lower levels of cleaved caspase 3 (Fig. 3b and Extended Data Fig. 7c). We profiled AKPS liver metastases in *Plxnb2*-OE and control livers using single-nucleus assay for transposase-accessible chromatin with sequencing (snATAC–seq) and snRNA-seq (Extended Data Fig. 7d,e). Gene set enrichment analysis (GSEA) in tumour cells confirmed upregulation of terms related to cell division, morphogenesis and assembly of cell projections (Fig. 3c,d and Extended Data Fig. 7f). Consistent with cytoskeletal remodelling, phalloidin staining

indicated notable apical F-actin accumulation in epithelial gland structures of AKPS liver metastases in *Plxnb2*-OE livers (Fig. 3e). Treatment of KPC cells and CRC PDOs with rmPlexin B2 similarly induced F-actin focal aggregation in vitro (Extended Data Fig. 7g,h). Notably, lesions in *Plxnb2*-OE livers also exhibited strong downregulation of genes involved in immune recognition, such as *Cd74*, *B2m* and *H2-D1*, coinciding with diminished CD4^+ T cell infiltration (Fig. 3d and Extended Data Fig. 7f,i).

To reveal the transcription factors mediating the effects of plexin B2 in metastases, we used the chromatin profile of tumour cells to identify differentially accessible peaks and their associated motifs (Fig. 3f). With respect to the controls, metastases growing in *Plxnb2*-OE livers exhibited increased activity of several members of the SP/KLF family of zinc-finger transcription factors (Fig. 3g). Consistent with these results, we detected increased nuclear levels of KLF4 in AKPS liver metastases growing in *Plxnb2*-OE livers, while it was absent from lesions in *Plxnb2*-KO livers (Fig. 3h). Moreover, expression of *Klf4* as well as its predicted target genes was increased in tumour cells in *Plxnb2*-OE livers, as well as in AKPS organoids treated with rmPlexin B2 (Extended Data Fig. 7j).

*Klf4* is expressed by differentiated cells of the colonic epithelium, but its expression is lost in CRC, in which it acts as tumour suppressor by inhibiting epithelial–mesenchymal transition (EMT)[29–31]. In AKPS metastases colonizing *Plxnb2*-OE livers, increased KLF4 coincided with elevated EPCAM immunoreactivity (Extended Data Fig. 7k). We therefore hypothesized that reactivation of KLF4 at secondary sites might promote epithelialization of tumour cells through reversion of EMT, which is thought to be essential for successful metastatic outgrowth of several carcinomas[32–34]. Treatment of two-dimensional (2D) AKPS cultures with the KLF4 inhibitor WX2-43 (ref. 35) induced a mesenchymal-like phenotype, altering colony morphology, size and actin cytoskeleton, and reduced the frequency of Ki-67^+ cells (Extended Data Fig. 7l–n), indicating that KLF4 suppresses mesenchymal traits and promotes proliferation in AKPS organoids. KLF4 inhibition further reduced seeding of AKPS organoids in vitro, which could be rescued by co-treatment with rmPlexin B2 (Extended Data Fig. 7o).

AKPS organoids in culture show a hybrid EMT state with non-overlapping transcriptional signatures of EMT and mesenchymal–epithelial transition (MET), and a mix of E-cadherin (ECAD)^high^ZEB1^low^ and ECAD^low^ZEB1^high^ cells (Extended Data Fig. 8a,b). Notably, nuclear ZEB1 levels are decreased after treatment with rmPlexin B2 and conversely increased by KLF4 inhibition (Extended Data Fig. 8c). In vivo, ZEB1 is absent from metastases growing in *Plxnb2*-OE and control livers, while AKPS lesions in *Plxnb2*-KO livers retain epithelial ZEB1 expression (Fig. 3i and Extended Data Fig. 8c). In the absence of plexin B2, AKPS metastases also lack expression of ELF3 and GRHL2, two transcription factors that preserve epithelial identity by suppressing EMT[36,37] (Extended Data Fig. 8d). These data implicate hepatocyte-derived plexin B2 as an inducer of epithelialization of AKPS liver metastases, and suggest that reversion of EMT is required for DTC seeding and adaptation to the liver environment. Consistent with a liver-induced epithelialization of metastases, AKPS organoid lines derived from metastases exhibit morphological and transcriptomic evidence of epithelial morphogenesis, and have an impaired ability to establish colonies when seeded in vitro as single cells, indicating increased susceptibility to anoikis (Extended Data Fig. 8d–f).

## Plexin B2 interacts with class IV semaphorins

Interactions between plexin B2 and its canonical ligands, class IV semaphorins, have been implicated in promoting tumour invasion and metastasis by means of cytoskeletal remodelling and activation of RAC1 signalling[38–41]. The seeding-promoting effect of rhPlexin B2 on PDOs in vitro was indeed prevented by antibody-mediated blockade of the semaphorin-binding domain of plexin B2 (Fig. 4a). Moreover, treatment of AKPS organoids with rmPlexin B2 increased in vitro and in vivo seeding in a RAC1-dependent manner (Extended Data Fig. 9a,b),

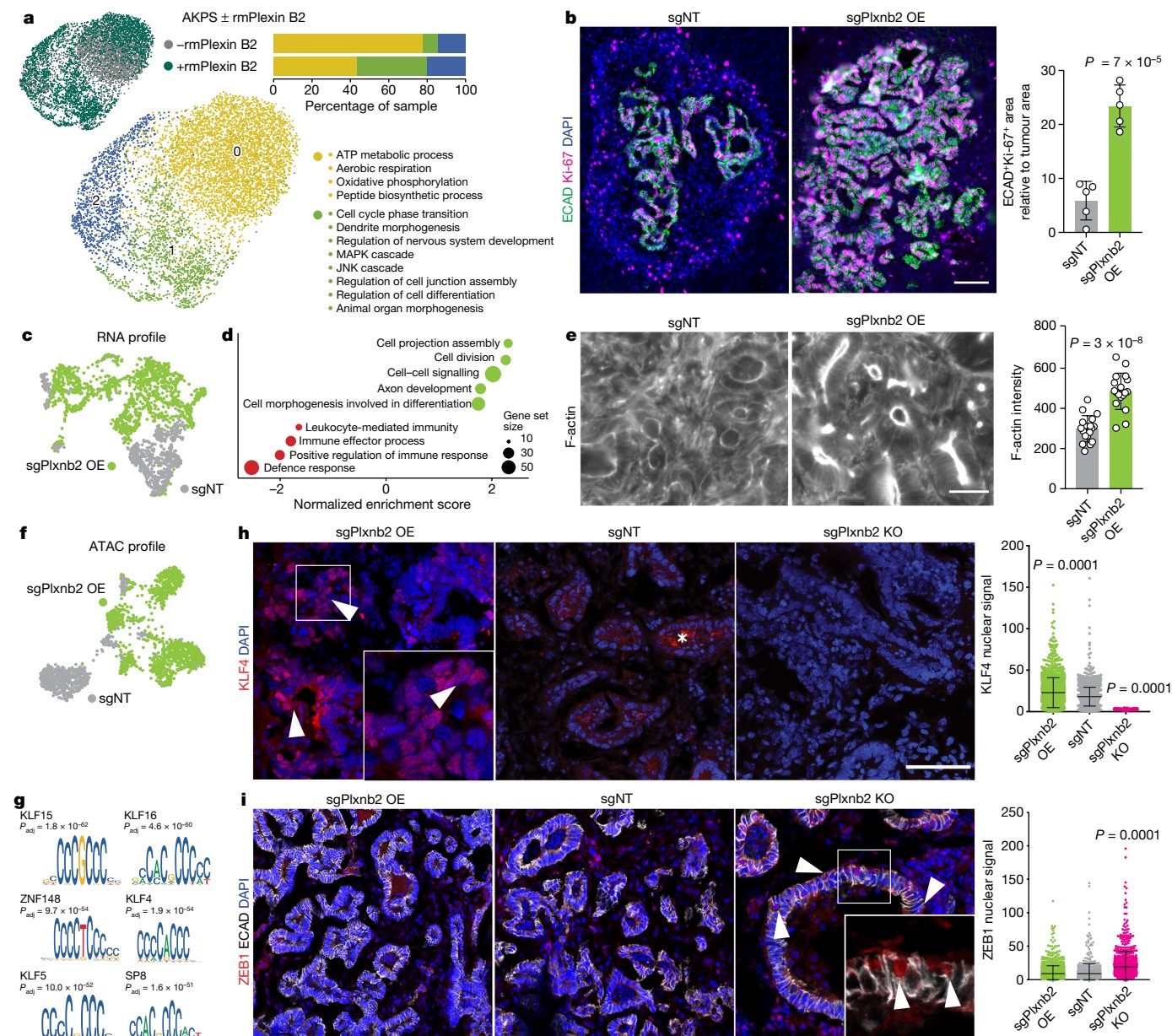

**Fig. 3 | Plexin B2 induces epithelialization of liver metastases. a**, Uniform manifold approximation and projection (UMAP) of AKPS organoids treated with or without rmPlexin B2. The dots represent single cells, coloured by treatment or transcriptional clusters (0–3). Significantly enriched gene ontology (GO) terms for clusters 0 and 1, and the cluster composition per sample are shown. **b**, Representative fluorescence micrograph and quantification of ECAD (green) and Ki-67 (magenta) immunoreactivity in AKPS metastases in *Plxnb2*-OE and control livers. DAPI (nuclear counterstain) is shown in blue. Scale bar, 100 μm. The dots represent individual liver metastases (*n* = 5) pooled from two mice per group. **c**, UMAP of tumour cells in *Plxnb2*-OE and control livers (RNA profile). The dots represent single nuclei coloured by condition. **d**, GSEA in tumour cells in *Plxnb2*-OE versus control livers. **e**, Representative fluorescence micrograph and quantification of F-actin in AKPS metastases in *Plxnb2*-OE and control livers. Scale bar, 20 μm. The dots represent apical F-actin segments in metastatic glands (*n* = 18) pooled from two mice per group. **f**, UMAP analysis of tumour cells in *Plxnb2*-OE and control livers (ATAC profile). Dots represent single nuclei coloured by condition. **g**, Enriched transcription-factor-binding motifs in differentially open peaks in tumour cells in *Plxnb2*-OE versus control livers. $P_{adj}$, adjusted *P*. **h**,**i**, Representative fluorescence micrograph and quantification of KLF4 (**h**) or ZEB1 (red) and ECAD (grey) (**i**) immunoreactivity in AKPS metastases in *Plxnb2*-OE, *Plxnb2*-KO and control livers. DAPI (nuclear counterstain) is shown in blue. Scale bar, 50 μm. Inset: KLF4+ or ZEB1+ nuclei (arrows). The asterisk indicates the background. The dots represent individual nuclei in metastases pooled from *n* = 3 mice per condition. Statistical analysis was performed using ordinary one-way ANOVA with Dunnet's correction for multiple testing (**h** and **i**) and two-tailed unpaired *t*-tests (**b** and **e**). Data are mean ± s.d. (**c**, **d** and **f**–**i**) pooled from two independent experiments, *n* = 2 mice per condition.

suggesting that class IV semaphorins mediate the seeding-promoting effects of hepatocyte-derived plexin B2 on DTCs.

In AKPS liver metastases, SEMA4A+ tumour cells contact hepatocytes at the metastatic leading edge, and class IV semaphorins (*Sema4a*, *Sema4c*, *Sema4d* and *Sema4g*) are widely expressed (Extended Data Fig. 9c,d). SEMA4A, SEMA4C, SEMA4D and SEMA4G are also detected in human colon and primary CRC[42] (Extended Data Fig. 9e). Notably, in two scRNA-seq datasets of human CRC[18], expression of the semaphorin genes and KLF4-target genes is significantly upregulated in high-relapse cells (HRCs)[43] and in intrinsic consensus molecular subtype 3 (iCMS3) cells[44], but not in Lgr5[high] cells (Fig. 4b–d and Extended Data Fig. 9f,g). Semaphorin expression also coincides with a MET signature and

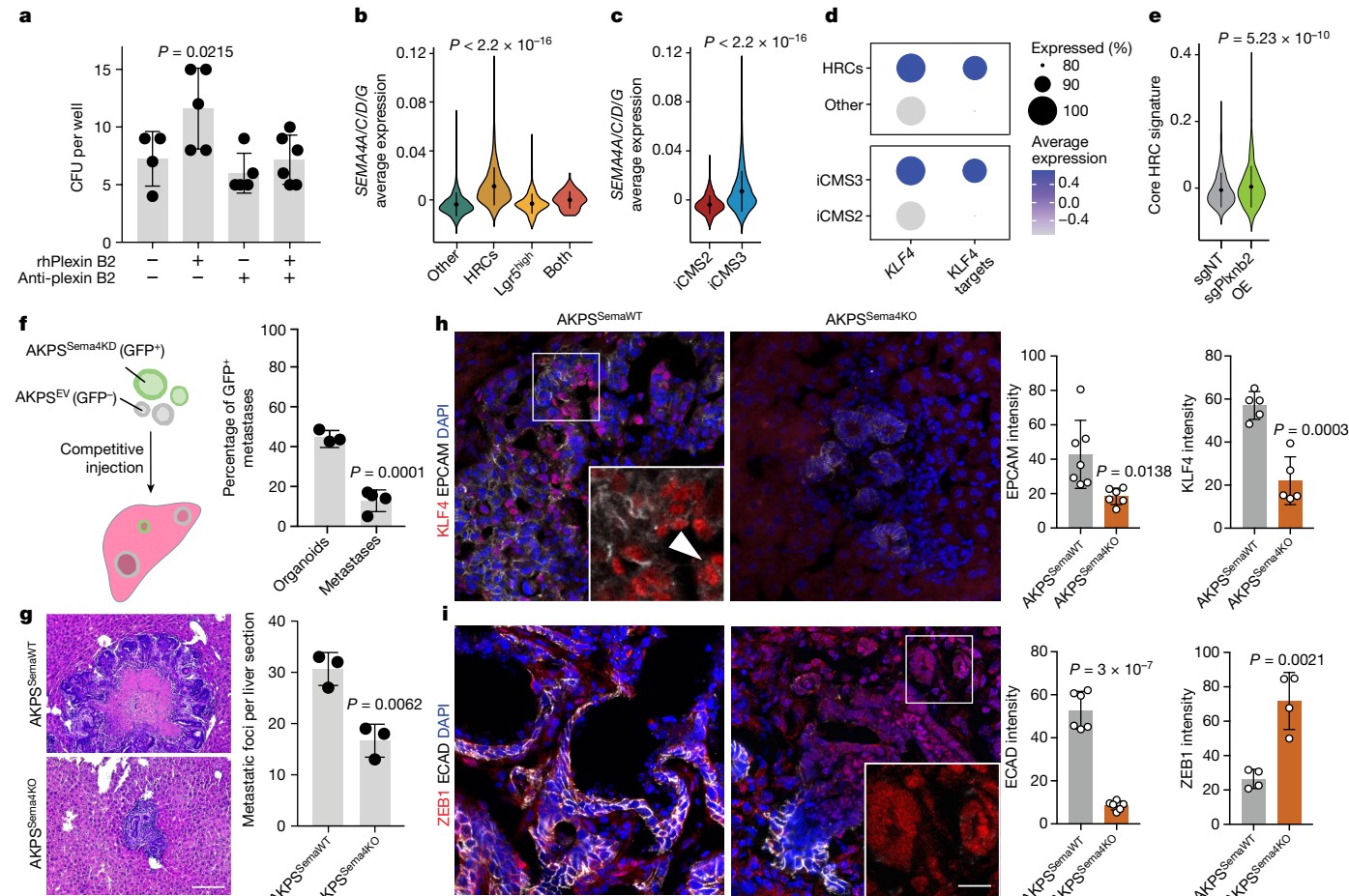

**Fig. 4 | Plexin B2 interacts with semaphorins on tumour cells. a**, The CFU per well of PDOs cultured with rhPlexin B2 ($n = 5$), anti-plexin B2 antibody ($n = 5$), both ($n = 6$) or IgG1 isotype control ($n = 4$). The dots indicate individual wells. Statistical analysis was performed using ordinary one-way ANOVA with Dunnet's correction for multiple testing. **b**,**c**, The average expression of *SEMA4A*, *SEMA4C*, *SEMA4D* and *SEMA4G* in epithelial cells of primary CRC (Samsung dataset[18]; $n = 25$ patients). Cells are grouped as HRCs, Lgr5[high], others or both (**b**), or iCMS2 versus iCMS3 (**c**). Statistical analysis was performed using two-tailed unpaired Wilcoxon tests. **d**, The expression of *KLF4* and KLF4 targets in HRCs versus other cells, and iCMS3 versus iCMS2 cells in primary CRC (Samsung dataset[18]; $n = 25$ patients). **e**, The expression of the core HRC signature in AKPS metastases in *Plxnb2*-OE versus control livers. Data were pooled from two independent experiments, $n = 2$ mice per condition. Statistical analysis was performed using two-tailed unpaired Wilcoxon rank-sum tests. **f**, Schematic of the competitive seeding assay (left). EV, empty vector. Right, the percentage of GFP[+] metastases (2 weeks after injection) versus the percentage of GFP[+] organoids in the injection mix. The dots represent individual mice ($n = 4$ mice per group). Results were pooled from two independent experiments. **g**, Representative H&E staining of AKPS[SemaWT] or AKPS[Sema4KO] metastases (left). Scale bar, 100 μm. Right, metastases per liver section 2 weeks after intrasplenic injection of AKPS[SemaWT] or AKPS[Sema4KO] organoids. The dots represent individual mice. $n = 3$ mice per group. **h**,**i**, Representative fluorescence micrograph and quantification of EPCAM (grey) and KLF4 (red) (**h**) or ECAD (grey) and ZEB1 (red) (**i**) immunoreactivity in AKPS[SemaWT] and AKPS[Sema4KO] metastases. Scale bar, 20 μm. Inset: KLF4[+] and ZEB1[+] nuclei (arrowhead). Scale bar, 10 μm. The dots indicate the average nuclear signal per individual metastasis ($n = 7$ (EPCAM), $n = 5$ (KLF4), $n = 6$ (ECAD), $n = 4$ (ZEB1)) pooled from three mice per condition. For **f**–**i**, statistical analysis was performed using two-tailed unpaired $t$-tests. For **a**–**c** and **e**–**i**, data are mean ± s.d.

expression of KLF4-target genes (Extended Data Fig. 9h). Conversely, the core epithelial HRC signature is significantly upregulated in metastatic cells grown in *Plxnb2*-OE livers, as well as in AKPS organoids after treatment with rmPlexin B2 (Fig. 4e and Extended Data Fig. 9i). Of note, semaphorin levels are unaltered in liver metastases compared with in matched primary tumours[21,22] (Extended Data Fig. 9j). These analyses indicate that high semaphorin expression marks a subpopulation in the primary tumour with elevated liver metastatic potential. Indeed, in a large cohort of patients with CRC, increased expression of *SEMA4A*, *SEMA4C* and *SEMA4D*, but not *SEMA4G*, is associated with reduced recurrence-free survival (Extended Data Fig. 9k). Moreover, copy-number variation (CNV) analysis in the COAD dataset[45] indicates that *SEMA4A*, *SEMA4C* and *SEMA4D* are commonly found amplified, while *SEMA4G* is often deleted in patients with CRC (Extended Data Fig. 9k). Cumulatively, these data support the role of plexin B2–semaphorin–KLF4 signalling in promoting liver seeding, and might explain

the differential ability of distinct tumour cell subpopulations to successfully form hepatic metastases.

To test the requirement of semaphorin genes for liver metastases, we performed simultaneous partial knockdown of *Sema4a*, *Sema4c*, *Sema4d* and *Sema4g* in AKPS organoids (AKPS[Sema4KD]), and observed downregulation of gene sets involved in epithelial morphogenesis, cell adhesion and RAC1 GTPase activity (Extended Data Fig. 10a–c). Notably, when co-injected with control organoids in competitive seeding assays, AKPS[Sema4KD] organoids exhibited reduced grafting ability in vivo (Fig. 4f and Extended Data Fig. 10d). To achieve complete deletion of semaphorins, we generated quadruple KO organoids (AKPS[Sema4KO]; Extended Data Fig. 10e and Supplementary Table 2). While AKPS[Sema4KO] organoids show unaltered proliferation in vitro, they exhibit significant grafting impairment when inoculated orthotopically (Extended Data Fig. 10f–h). To assay liver colonization, we injected AKPS[Sema4KO] organoids intrasplenically, and found significantly decreased

metastatic burden compared with control organoids (Fig. 4g). Notably, KLF4 immunoreactivity is lost in AKPS$^{Sema4KO}$ metastases, which further exhibit diminished EPCAM and ECAD expression and high ZEB1 levels (Fig. 4h,i). Loss of semaphorins in AKPS liver metastases therefore phenocopies loss of plexin B2 on hepatocytes, supporting the notion that plexin–semaphorin interactions are required for metastatic seeding.

## Discussion

Historically viewed as a late-stage event in cancer progression, spreading of DTCs has recently gained recognition as an early phenomenon in tumorigenesis[46,47]. Recent studies have elucidated mechanisms that promote DTC survival in circulation[48], regulate DTC dormancy[49–51] or promote recurrence[43]. Yet, owing to the technical challenges of tracking single extravasated tumour cells, the environmental determinants of DTC adaptation and survival in a foreign organ environment remain largely unclear. Identification of these factors might reveal a therapeutic window to target metastasis at its most vulnerable point: before the establishment of a growth-promoting metastatic niche. The results presented here identify the interaction between hepatocyte-derived plexin B2 and class IV semaphorins on tumour cells as a necessary inducer of KLF4-mediated epithelialization of liver metastases and lay a methodological framework to deepen our understanding of metastatic seeding.

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

# Methods

## Mice

All of the experiments were performed in male and female mice aged 6–16 weeks. *LSL-dCas9-SPH* (dCas9-SPH, 031645) and *Plxnb2*$^{flox/flox}$ mice (036883) were obtained from The Jackson Laboratory. *Alb-cre* (003574), *LSL-Cas9* (Cas9, 024858) and mT/mG mice (007576) were obtained from a local live mouse repository. *LSL-dCas9-SPH* mice were obtained in 2019 and initially exhibited spontaneous urination, as also reported on the Jackson Laboratory website. The phenotype disappeared when the mice were outcrossed to B6J mice and crossed with the *Alb-cre* strain. Chow and water were available ad libitum, unless specified otherwise. All of the mice were in the B6J background and maintained under a 12 h–12 h light–dark schedule. Mice were housed and bred under specific-pathogen-free conditions in accredited animal facilities. At the experimental end point, mice were euthanized by raising the $CO_2$ concentrations. Approved humane end points (a palpable tumour with a diameter of more than 1.5 cm, or weight loss of more than 15%) were not exceeded. All of the experimental procedures were performed in accordance with Swiss Federal regulations and approved by the Cantonal Veterinary Office. Genotyping was performed by Transnetyx genotyping services.

## Animal experiments

**Hydrodynamic tail vein injection.** Hydrodynamic injection, an efficient method to deliver nucleic acids to the liver, involves the rapid injection (6–8 s) of a large volume (8–12% body weight) of saline (0.9% sodium chloride) into the tail[52]. Mice were placed in a restraining device, and the tail was warmed with a red light lamp to induce lateral tail-vein dilatation. The injection site was cleaned and disinfected using an alcohol swab. Transposon plasmid (*PT4-CMV-GFP* (Addgene, 11704) or *PT4-U6-sgRNA-CMV-GFP*) and SB100X transposase OE plasmid pCMV(CAT)T7-SB100X (Addgene, 34879) were equilibrated at room temperature, transferred to a 3 ml syringe mounted with a 27 G needle, and then injected into the tail vein in a continuous motion. The mouse was removed from the restrainer and signs of recovery were monitored.

**Intrasplenic injection of tumour cells followed by splenectomy.** Intrasplenic injection of tumour cells followed by splenectomy was performed as previously described[53]. In brief, the mice were placed into a container connected to an oxygen–isoflurane inhalation device (4% for induction and 2–3% for maintenance). After 5 min, anaesthetized mice were placed onto a thermal pad (37 °C), isoflurane gas was continuously supplied by a nose cone and sterile eye ointment was applied to avoid corneal dehydration. Anaesthesia depth was monitored regularly by testing toe and tail pinch reflexes and by observing the rate, depth and pattern of respiration. All of the surgical instruments were sterilized before use and the surgical procedure was performed under aseptic conditions. The skin over the surgical site was shaved and disinfected with betadine. Using aseptic technique, an incision was made in the skin and peritoneal wall to expose the spleen. A sterile gauze was placed under the spleen. For each mouse, four 50 µl domes of AKPS organoids were collected, washed of Matrigel in ice-cold PBS and mechanically dissociated using a 20 G needle on a 20 ml syringe. Dissociated organoids were pelleted at 290$g$ for 3 min, and resuspended in 0.04 ml sterile PBS and injected under the splenic capsule with an insulin syringe (BD, MicroFine, 0.3 ml, 30 G). Alternatively, 100,000 KPC cells in 0.04 ml sterile PBS were injected. After 10 min, the splenic artery and vein were closed by ligation. Immediately after, the spleen was resected. Subsequently, the wound was washed three times with sterile PBS. The peritoneal wall was closed with an absorbable polyglactin suture (Vicryl 4-0 or 5-0 coated) and the skin was closed with wound clips. The mice were monitored for weight loss and the experiment was terminated maximally 3 weeks after tumour cell injection.

**Colonoscopy-guided submucosal injection of CRC organoids.** The procedure for the colonoscopy-guided orthotopic injection of mouse colonic organoids was adapted from a previously published protocol[54]. At 36 h after passaging, AKPS or APTAK organoids were mechanically dissociated, resuspended in OptiMEM (Gibco) (one 50 µl Matrigel dome in 50 µl of OptiMEM per mouse for AKPS, three 50 µl Matrigel domes in 50 µl of OptiMEM per mouse for APTAK). Mice were anaesthetized by isoflurane inhalation and placed onto their back on a heating pad (37 °C). The colons were evacuated of stool with prewarmed PBS (37 °C) (Gibco) using the plastic tubing from an intravascular catheter (BD) mounted onto a 50 ml syringe (B. Braun). The organoid solution was injected with custom injection needles (33 gauge, 400 mm length, point style 4, 45° angle, Hamilton), a syringe (Hamilton) and a colonoscope with integrated working channel (Storz). The needle was brought into contact with colonic mucosa and 50 µl of organoid solution was quickly delivered to form a submucosal injection bubble. The mice were then monitored until the experimental or humane end point was reached.

**Tail vein injection of AAV8.** A 100 µl solution of $1 × 10^{11}$–$2 × 10^{11}$ AAV viral genomes in sterile saline was loaded into a 1 ml syringe mounted with a 27 G needle. Mice were placed in a restraining device, and the tail was warmed with a red-light lamp to induce lateral tail-vein dilatation. The injection site was cleaned and disinfected using an alcohol swab, and then the AAV-containing solution was injected into the tail vein. The injections were performed 1 week before tumour inoculation for *Plxnb2* overexpression, 2 weeks before or 5 days after tumour inoculation for *Plxnb2* knockout, or 2 weeks after tumour inoculation for primary tumour experiments.

**In vivo BLI.** At 5 min before imaging, mice were injected intraperitoneally with luciferin substrate (D-luciferin, 150 mg per kg body weight in 0.15 ml PBS). During imaging on the Lumina S5 In Vivo Imaging System (IVIS, PerkinElmer), mice were anaesthetized with isoflurane and kept warm on a heated stage.

## In vivo CRISPR-a screen

**Library cloning.** sgRNA sequences for dCas9-SPH-mediated OE were retrieved from the Caprano library[55] and obtained as oPool from Integrated DNA Technologies with Gecko flanking sequences[56]. Three sgRNAs were selected per target from the Caprano Set A, for a total of 897 sgRNAs targeting 299 genes. One hundred safe-targeting sgRNAs[11] were also included. The library (Supplementary Table 1) was cloned into the transposon vector PT4-CMV-GFP (Addgene, 117046) and amplified as described previously[56]. In brief, oPools were resuspended to 1 µg µl$^{-1}$ in water and incubated for 2 h at 37 °C, then amplified by PCR (a list of the primer sequences is provided in Supplementary Table 3): 1 µl library (1 ng µl$^{-1}$) was mixed with 12.5 µl NebNext MasterMix, 1.25 µl oligo reverse primer (10 µM), 1.25 µl oligo forward primer (10 µM) and 9 µl water and incubated in a thermocycler (98 °C for 30 s; 20 cycles of 98 °C for 10 s, 63 °C for 10 s, 72 °C for 12 s; then 72 °C for 2 min). The PCR products were purified using the Qiagen QIAquick PCR Purification Kit, eluted in 30 µl buffer EB (Qiagen) and separated on a 2% agarose gel in Tris-borate EDTA (TBE) buffer with SybrSafe Dye. The transposon vector was digested overnight with the Bsmb-v2 enzyme and run on a 2% agarose gel in TBE buffer with SybrSafe Dye. The 150 bp sgRNA amplicon and the digested vector band (missing 1,000 bp filler) were excised and gel-extracted using the QIAquick Gel Extraction Kit. Both the vector and insert were processed for isopropanol purification by incubating 50 µl eluted DNA with 50 µl isopropanol, 0.5 µl GlycoBlue Coprecipitant and 0.4 µl 5 M NaCl. The reactions were vortexed and incubated at room temperature for 15 min, before centrifugation at 16,000$g$ for 15 min at room temperature. The precipitate was washed twice with 1 ml ice-cold 80% ethanol and air-dried for 1 min before resuspension in 10 µl water. The DNA concentration was measured

using the NanoDrop system. The Gibson assembly reaction mix containing 10 µl MasterMix, 330 ng vector, 50 ng insert and water to 20 µl was incubated at 50 °C for 1 h. After a second isopropanol precipitation, the cloned transposon libraries were resuspended in 5 µl Tris-EDTA (TE) buffer and incubated at 55 °C for 10 min. The DNA concentration was measured using the NanoDrop system.

**Library amplification.** *Escherichia coli* electrocompetent bacteria (E. cloni; Biosearch Technologies) were thawed on ice for 20 min before addition of 1 µl of Gibson reaction and electroporation (1,800 V) in a MicroPulser Electroporator (Bio-Rad). A total of 975 µl of prewarmed recovery medium was added and the bacteria were incubated 37 °C at 250 rpm for 1 h. Bacteria were plated onto prewarmed 30 cm² square agar plates with ampicillin and incubated overnight at 37 °C. The plates were scraped with 30–50 ml Luria–Bertani broth and plasmids were purified using the endotoxin-free MaxiPrep kit (Macherey-Nagel). Precipitated DNA was resuspended in 1 ml water and the concentration was measured using the NanoDrop system. pCMV(CAT)T7-SB100 (Addgene, 34879) was amplified in competent *E. coli* under chloramphenicol selection and purified using the Endotoxin-Free MaxiPrep kit (Macherey-Nagel).

**Library delivery to mouse liver and tumour inoculation.** The sgRNA transposon plasmid libraries (150 µg) and SB100X-encoding plasmids (50 µg) were co-injected hydrodynamically into *Alb-cre;dCas9-SPH* mice or littermate controls lacking *Alb-cre*. Then, 1 week later, dissociated AKPS^sLP-mCherry organoids were inoculated by intrasplenic injection, followed by splenectomy. Metastases were allowed to grow for 2 weeks, during which several (10–100) small metastases (0.5–1.5 mm) formed.

**Isolation of metastasis-distal and metastasis-proximal hepatocytes.** Mice were euthanized by raising the $CO_2$ concentrations, then the abdomen was opened and a G22 cannula was inserted into the inferior vena cava. The liver was perfused with 20 ml Hanks buffer (0.5 mM EDTA and 25 mM HEPES in HBSS) followed by 15 ml digestion buffer (15 mM HEPES and 32 µg ml⁻¹ liberase (Roche) in low-glucose DMEM). After initial swelling of the liver, the portal vein was cut to allow outflow. After perfusion, the gallbladder was removed and the liver was transferred to a Petri dish with 10 ml digestion buffer and squished with a cell scraper to release the hepatocytes. Liberase was inactivated by adding 10 ml isolation buffer (10% fetal bovine serum (FBS) in low-glucose DMEM). The cell suspension was filtered through a 100 µm cell strainer and centrifuged at 50*g* for 2 min. The supernatant was removed and the pellet was washed again twice with 20 ml isolation buffer. Cells were resuspended in 2 ml FACS buffer (2 mM EDTA, 0.5% BSA in PBS) and stained with Zombie Violet (1:500) (BioLegend, 423113), TruStain FcX (anti-mouse CD16/32) antibody (BioLegend, 101320, 1:50), PE/Cy7 anti-mouse CD31 (BioLegend, 102418, 1:300) and BV570 anti-mouse CD45 (BioLegend, 103135, 1:300) for 25 min at 4 °C. Cells were washed and filtered through a 70 µm strainer. CD45⁻CD31⁻ hepatocytes that contained a sgRNA (GFP⁺), were divided into metastasis-proximal (mCherry⁺) and metastasis-distal (mCherry⁻) by drawing different sorting gates on an AriaIII sorter (BD Biosciences) with 100 µm nozzle (the gating strategy is shown in the Supplementary Information). Cells were collected in PBS, centrifuged at 800*g* for 5 min and the pellet was stored at −20 °C.

**Genomic DNA extraction and targeted guide amplification and sequencing.** Genomic DNA extraction was performed as described in the 'Isolation of genomic DNA with NucleoSpin Blood Kits and PCR pre-check' protocol of the Broad Institute's Genetic Perturbation Platform. In brief, the pellet was equilibrated at room temperature, resuspended in 200 µl PBS and incubated with proteinase K and lysis buffer mixture at 70 °C overnight. Then, 1 µl RNase A was added for 5 min at room temperature, followed by column purification using

the NucleoSpin Blood Mini Kit (Macherey Nagel). DNA was eluted in 25 µl elution buffer (prewarmed at 70 °C), incubating on column for 5 min before centrifugation, then the concentration was measured using the NanoDrop system. sgRNAs were target amplified from both post-injection libraries and the plasmid library using an equimolar mixture of staggered P5 primers and P7 primers with sample-specific barcode (the sequences are provided in Supplementary Table 3) under the following conditions: 10 µl titanium buffer, 8 µl dNTPs, 5 µl DMSO, 0.5 µl P5 primer mix (100 µM), 40 µl water, 1.5 µl Titanium Taq polymerase, 25 µl DNA (maximum, 10 µg), 10 µl P7 primer (5 µM). The reactions were incubated in a thermocycler with the following programme: 95 °C for 5 min; 28 cycles of 95 °C for 30 s, 53 °C for 30 s and 72 °C for 20 s; and 72 °C for 10 min. PCR products from different samples were pooled according to the number of reads required to ensure 200–1,000 reads per sorted cell. After 1× SPRIselect bead (Beckman Coulter, B23319) purification, PCR products were eluted in 50 µl TE buffer. The quality and quantity of libraries were assessed using the dsDNA high-sensitivity kit (Life Technologies, Q32854) on the Qubit 4 fluorometer (Thermo Fisher Scientific) and using the high-sensitivity D1000 reagents and tapes (Agilent, 5067-5585, 5067-5584) on the TapeStation 4200 or Bioanalyzer (Agilent Technologies) system. Libraries were sequenced using the NextSeq kit (Illumina) with 75 bp single-end read chemistry and 9 bp index read, adding 10% PhiX spike-in (Illumina).

**Replicates and coverage.** The procedure outlined above (from cloning to sequencing) was repeated independently three times (three batches). In summary, the sgRNA coverage (that is, the number of sorted cells/997 sgRNAs) added up to a total of 1,000× for *Alb-cre;dCas9-SPH* mice (*n* = 7) and 500× for non-Cre littermate controls (*n* = 5).

**Analysis.** FASTQ files were demultiplexed using Bcl2fastq v.2.20.0.422 (Illumina) and adaptors were trimmed with cutadapt[57] using the following parameters (-g CACCG and -a GTTTT). The sequences were aligned to the sgRNA library using Bowtie2[58,59]. sgRNAs were counted using the MAGeCK count function (--norm-method total)[60,61]. sgRNA enrichment was calculated using the MAGeCK paired test function. Metastasis-proximal and metastasis-distal libraries from the same mouse were considered as paired samples. As sgRNAs in paired samples are considered to be independent sgRNAs (3 sgRNAs in 7 mice are thus considered to be 21 independent sgRNAs), paired testing yields consistent effects between paired replicates. This analysis was repeated for the individual library batches. The paired function was not used to compare perturbation enrichment in *Alb-cre;dCas9-SPH* versus non-Cre littermates, as an equal number of samples is required. Instead, sgRNA counts were added up from all *Alb-cre;dCas9-SPH* and non-Cre mice and then the standard MAGeCK RRA test function was applied. Results were visualized using MAGeCKFlute[60] and ggplot2[62].

## Spatial transcriptomics of human CRC liver metastases

**Visium library preparation.** A sample of human CRC hepatic metastasis (CB522586, 44 year old, male) with clear tumour-liver borders was selected from a commercial biobank (Origene). Non-consecutive sections were cut with a thickness of 10 µm and placed onto two capture areas of the 10x Visium Spatial Gene expression slide using the cryostat (Leica, CM3050S). The tissue optimization kit was used to determine the permeabilization times (24 min), and cDNA libraries were generated according to the manufacturer's instructions (10x Genomics). The quality and quantity of libraries were assessed using the dsDNA high-sensitivity kit (Life Technologies, Q32854) on the Qubit 4 fluorometer (Thermo Fisher Scientific) and the high-sensitivity D1000 reagents and tapes (Agilent, 5067-5585, 5067-5584) on the TapeStation 4200 (Agilent Technologies) system. Paired-end sequencing was performed for all of the libraries (read 1:28 bp; index read: 10 bp; read 2: 82 bp) on the NovaSeq 6000 (Illumina) system using NovaSeq SP Reagent Kits (100 cycles).

**Data analysis.** Binary base call (BCL) files were demultiplexed using Bcl2fastq v.2.20.0.422 (Illumina) and preprocessed using Space Ranger (v.1.1.0 or v.1.2.0; 10x Genomics). Spot transcriptomes were deconvoluted with Spotlight[63] using published scRNA-seq data as reference. Specifically, two datasets of primary CRC[18] and liver tumour microenvironment[19] were integrated using the Seurat integration method[64]. Edge spot selection was performed using the CellSelector function in Seurat, and the FindMarkers function was used for differential gene expression analysis of metastasis proximal versus distal and metastasis core versus centre. NicheNet[65] was used to predict LR interactions at the metastatic leading edge (503 possible interactions). The ligand activity analysis from NicheNet was used to estimate the potential of these interactions to regulate differentially expressed genes (DEGs) between metastasis edge and core, yielding 109 LR pairs with regulatory potential. These were then intersected with SRFs (top and bottom decile of the screen, 62 factors).

### Cross-validation with human and transcriptional mutational data

**Interactors of liver-metastasis-specific mutations.** The Genomic Features of Organotropism dataset[20] was used to extract genes with mutations more frequently occurring in liver metastases as compared to primary tumours. This set of genes was then parsed with CellPhoneDB (v.3)[66], CellTalkDB[67] and NicheNet[65] to filter for ligand and receptors and compile a list of their known interactors. The obtained interactors were then filtered for expression by hepatocytes according to the Human Protein Atlas (v.22.0; https://www.proteinatlas.org/)[68]. Specifically, we filtered out genes with transcript per million (TPM) ≤ 0.5 in the RNA GTEx tissue gene data, normalized TPM ≤ 0.5 in the RNA single-cell data (v.22.0 https://www.proteinatlas.org/humanproteome/single+cell+type) and 'not detected' in the normal tissue data (v.22.0; https://www.proteinatlas.org/; Ensembl v.103.38). The data used for the analyses described in this Article were obtained from the GTEx Portal on 30 May 2022 and/or dbGaP phs000424 on 30 May 2022. The interactors of LRs frequently mutated in liver metastases were then intersected with SRFs.

**Interactors of liver-metastasis-specific DEGs.** Two published datasets of primary CRC tumours and matched liver metastases[21,22] were downloaded and imported into Seurat. Tumour cells were subsetted on the basis of EPCAM expression, then DEGs were calculated between liver metastases and primary CRC using the FindMarker function in Seurat. DEGs were parsed using CellPhoneDB (v.3)[66], CellTalkDB[67] and NicheNet[65] to filter for ligand and receptors and compile a list of their known interactors, which were intersected with SRFs (top and bottom decile, 62 factors).

### Organoid culture and modification

**Mouse CRC organoid culture.** *Vil-creER*[T2]*;APC*[fl/fl]*;Trp53*[fl/fl]*;Kras*[G12D/WT] *Smad4*[KO] (AKPS) and *Vil-creER*[T2]*;APC*[fl/fl]*;Trp53*[fl/fl]*;Kras*[G12D/WT]*Tgfbr2*[flox/flox] *Akt1*[myristoilated] (APTAK) organoids were cultured in 50 μl Matrigel domes (Corning) as described previously[69]. To make complete medium, Advanced DMEM/F12 (Life Technologies) was supplemented with 10 mM HEPES (Life Technologies), 2 mM L-glutamine (Life Technologies), 100 mg ml⁻¹ penicillin–streptomycin (1%), 1× B27 supplement (Life Technologies), 1× N2 supplement (Life Technologies) and 1 mM *N*-acetylcysteine (Sigma-Aldrich). Organoids were split every 3–5 days by mechanical dissociation. Splitting was always performed on the day before intrasplenic injection and 36 hours prior to orthotopic inoculation.

**RNP-mediated *Smad4* KO.** *Vil-creER*[T2]*;APC*[fl/fl]*;Trp53*[fl/fl]*;Kras*[G12D/WT] mice were obtained from Owen Sansom (Beatson Institute for Cancer Research) and cultured under the above described conditions with supplementation of 100 ng ml⁻¹ mouse recombinant noggin (LuBioScience, 250-38-250). Four domes of organoids were treated with 5 mM nicotinamide for 2–3 days before transfection. sgRNAs comprising both crRNA and tracrRNA sequences were obtained from IDT (Alt-R system). Targeting sequences were obtained from a previous study[70]. Organoids were collected, washed of Matrigel, and dissociated into single cells by incubation at 37 °C for 10 min in 1 ml prewarmed TrypLE Express Enzyme (Gibco). Cells were centrifuged at 190*g* for 3 min, resuspended in 1 ml complete medium with 5 mM nicotinamide and 10 μM Y-27632 dihydrochloride (Rock inhibitor) (StemCell, 72304), and seeded into two wells of a 48-well plate. Transfection was performed using the CRISPRmax kit (Thermo Fisher Scientific). In brief, 25 μl OPTImem supplemented with 1,250 ng Cas9 nuclease V3 (1081058), 240 ng sgRNA and 2.5 μl Cas9 Reagent Plus was mixed with 25 ml OPTImem with 1.5 μl CRISPRmax reagent, incubated for 10 min and added to each well. The cells were spinoculated for 1 h at 600*g* at 32 °C, then incubated for 4 h at 37 °C. Cells were then collected, resuspended in Matrigel and plated in complete medium supplemented with Rock inhibitor. Then, 3 days later, selection was started by adding medium supplemented with 10 ng ml⁻¹ TGFβ (Peprotech, 100-21C-10UG) and lacking noggin. Selection was continued for 3 passages, then TGFβ was removed from the medium. Successful editing was confirmed by T7 endonuclease assay. Primers asymmetrically flanking the cut site were designed so as to yield fragments distinguishable by electrophoresis. Organoid DNA was extracted using the QuickExtract DNA Extraction Solution (Lucigen) and then PCR amplified. The PCR reaction mix consisted of 10 μl Q5 master mix, 6 μl H₂O, 2 μl primer mix and 2 μl DNA. Then, 10 μl PCR products was mixed with 1.5 μl 10× NEB buffer 2 and 1.5 μl H₂O and incubated in a thermocycler as follows: 10 min at 95 °C; from 95 °C to 85 °C with a ramp rate of −2 °C s⁻¹; from 85 °C to 25 °C with a ramp rate of −0.3 °C s⁻¹. Formed heteroduplexes were incubated with 2 μl T7 mix (10 μl NEBuffer 2, 10 μl T7 and 80 μl H₂O) at 37 °C for 1 h. The reaction was stopped by the addition of 1 μl of 0.5 M EDTA. The samples were analysed by electrophoresis on a 2% agarose gel. Abrogation of SMAD4 signalling was further confirmed by the loss of *Id3* expression, as assessed using quantitative PCR with reverse transcription (RT–qPCR).

**Integration of the sLP-mCherry labelling system.** The pcPPT-mPGK-attR-sLP-mCherry-WPRE vector was obtained from Ximbio and lentiviruses were generated according to a published protocol[71]. In brief, HEK293T cells were cultured on poly-D-lysine-coated plates and transfected with 4.4 μg PAX2, 1.5 μg VSV-G and 5.9 μg lentiviral vector with JetOptimus reagents. The supernatant was collected after 2 days, centrifuged 5 min at 500*g* and filtered through a 0.45 μm filter. Then, 4 domes of AKPS organoids were washed from Matrigel and dissociated into single cells by incubating them for 10 min in 1 ml prewarmed TripLE at 37 °C. Cells were centrifuged at 190*g* for 3 min, then resuspended in 2 ml infection medium containing 1.8 ml virus, 200 μl complete medium, 5 mM nicotinamide, 1.6 μl polybrene and 2 μl Rock inhibitor. The cells were plated into two wells of a 48-well plate and spinoculated for 1 h at 600*g* at 32 °C, then incubated 4 h at 37 °C. Cells were then collected, resuspended in Matrigel and plated into complete medium supplemented with Rock inhibitor. Then, 3 days later, the organoids were dissociated into single cells and mCherry⁺ cells were isolated by FACS using an the Aria III sorter (BD Biosciences). Organoids were further selected with 2 μg ml⁻¹ puromycin for a week.

**Integration of the luciferase reporter for BLI.** To generate the pLVX-fireflyLuc-IRES-zsGreen1 vector, the protein coding sequence of firefly luciferase was amplified from an in-house plasmid, and cloned into EcoRI/BamHI-linearized pLVX-IRES-zsGreen1 (Takara) by InFusion (InF-fireflyLuc-F: TATTTCCGGTGAATTCCACCATGGAAGACG CCAAAAAC and -R: GAGAGGGGCGGGATCCTTACACGGCGATCTTT CCGCC). Sanger (Microsynth) and whole-plasmid (PlasmidSaurus) sequencing confirmed the identity of the construct and the absence

of unwanted mutations. Lentiviral preparation and transduction of organoids was conducted as described above. Successfully transduced organoids were selected by FACS on the basis of GFP fluorescence and gating for live cells.

**shRNA-mediated semaphorin KD.** The sequences of shRNAs targeting *Sema4a*, *Sema4c*, *Sema4d* and *Sema4g* were obtained from The RNAi Consortium shRNA Library (Broad Institute) and cloned in an arrayed manner in a lentiviral vector expressing GFP as a selection marker based on a published plasmid backbone[72]. The EV control expressed a puromycin-resistance cassette for selection. Sanger (Microsynth) and whole-plasmid (PlasmidSaurus) sequencing confirmed the identity of each construct and the absence of unwanted mutations. Lentiviral preparation and transduction of organoids was conducted as described above. Organoids transduced with shRNAs targeting semaphorins were dissociated into single cells, incubated with Zombie Violet (1:500, BioLegend, 423113) and selected by FACS based on GFP fluorescence and gating for live cells. AKPS organoids transduced with the EV were also dissociated into single cells and subjected to sorting (only live-cell gate), and then selected by puromycin as described above. Semaphorin knockdown was assessed using RT–qPCR. For competitive seeding assays, AKPS^Sema4KD and AKPS^EV organoids were grown separately and then mixed at a 1:1 ratio and mechanically dissociated for intrasplenic injection. A small fraction of the injection mix was seeded in three domes to estimate the injection ratios.

**enAsCas12a-mediated KO of class IV semaphorins.** AKPS^Sema4KO organoids were generated using enAsCas12a technology[73,74]. sgRNAs targeting *Sema4a*, *Sema4c*, *Sema4d* and *Sema4g* were obtained using the guide design tool CRISPick[74,75] with the mouse GRCm38 reference genome and enAsCas12a CRISPRko mode. Two high-scoring sgRNAs (on-target efficacy score > 0.75) targeting different exons for each semaphorin were combined into an 8-mer pre-crRNA array (Sema4KO array). The Sema4KO array was cloned into a in a lentiviral vector expressed under the CMV promoter together with GFP and puromycin (pLVX-EF1a-EGFP-2A-Puro-Triplex-Sema4KO-array-WPRE)[73]. Lentiviral preparation and transduction of organoids was conducted as described above. AKPS^Sema4KO organoids were generated by co-transduction of pLVX-enAsCas12a-BSD (pRDA174, Addgene, 136476)[74] and pLVX-EF1a-EGFP-2A-Puro-Triplex-Sema4KO-array-WPRE, and selected with 5 µg ml⁻¹ blasticidin and 2 µg ml⁻¹ puromycin for 7 days. Control organoids were transduced with pLVX-enAsCas12a and selected with 5 µg ml⁻¹ blasticidin. GFP expression was confirmed by microscopy, then clonal lines were generated by single-cell picking, and screened for indels by next-generation-sequencing-based amplicon sequencing. For immunofluorescence staining of class IV semaphorins, organoids were seeded in Matrigel on eight-well chambers (Thermo Fisher Scientific). Then, 3 days after, the medium was removed from the wells and organoids were fixed with 400 µl 4% PFA for 20 min at room temperature. After washing twice with 400 µl 3% BSA in PBS, the organoids were permeabilized for 20 min at room temperature in 0.5% Triton X-100 in PBS. The organoids were incubated with mouse anti-SEMA4A (BioLegend, 148402), rat anti-SEMA4C-AF647 (Biotechne, FAB8497R), rat anti-SEMA4D-PE (BioLegend, 147603) and rabbit anti-SEMA4G (Thermo Fisher Scientific, BS-11479R) overnight in 1% BSA, 0.2% Trizol and 0.05% Tween-20. After washing three times with working solution, the organoids were incubated 1 h at room temperature with anti-mouse Alexa Fluor 405 and anti-rabbit Alexa Fluor 647 (Thermo Fisher Scientific, 1:400, in working solution). Nuclei were stained with DAPI (Sigma-Aldrich, 1:1,000).

**Patient-derived CRC organoids.** Human CRC organoids were obtained from the Visceral Surgery Research Laboratory at the University of Basel. Tissues from primary and liver metastases of patients with CRC were obtained from the University Hospital Basel after patient consent

and ethical approval (Ethics Committee of Basel, EKBB, 2019-00816). To generate PDOs, tissue was cut into small pieces and, subsequently, enzymatically digested in 5 ml advanced DMEM/F-12 (Thermo Fisher Scientific, 12634028) containing 2.5 mg ml⁻¹ collagenase IV (Worthington, LS004189), 0.1 mg ml⁻¹ DNase IV (Sigma-Aldrich, D5025), 20 µg ml⁻¹ hyaluronidase V (Sigma-Aldrich, H6254), 1% BSA (Sigma-Aldrich, A3059) and 10 µM LY27632 (Abmole Bioscience, M1817) for 1 h and 30 min at 37 °C under slow rotation and vigorous pipetting every 15 min. The tissue lysate was filtered through a 100 µM cell strainer and centrifuged at 300*g* for 10 min. The cell pellet was resuspended with growth-factor-reduced Matrigel (Corning, 356231), plated into 50 µl domes and overlaid with medium composed of advanced DMEM/F-12 supplemented with 10 mM HEPES (Thermo Fisher Scientific, 15630056), 100 µg ml⁻¹ penicillin–streptomycin (Thermo Fisher Scientific, 10378-016), 1× GlutaMax (Thermo Fisher Scientific, 9149793), 100 µg ml⁻¹ primocin (invivoGen, ant-pm-1), 1× B27 (Thermo Fisher Scientific, 17504044), 1.25 mM *N*-acetylcysteine (Sigma-Aldrich, A9165-25G), 10 mM nicotinamide (Sigma-Aldrich, N0636), 500 ng ml⁻¹ R-spondin (EPFL SV PTPSP), 100 ng ml⁻¹ noggin ((EPFL SV PTPSP), 50 ng ml⁻¹ EGF (PeproTech, AF-100-15), 500 nM A83-01 (R&D Systems, 2939), 10 µM SB202190 (Sigma-Aldrich, S7076), 10 nM prostaglandin E2 (Tocris Bioscience, 2296), 10 nM gastrin (Sigma-Aldrich, G9145) and 10 µM Y-27632 dihydrochloride. The medium was changed every 3 days, and the organoids were passaged using 0.25% trypsin-EDTA (Life Technologies, 25200-056).

## Cell lines

**Mouse immortalized hepatocytes.** AML12 cells were obtained from ATCC (CRL-2254) and cultured in DMEM:F12 Medium (Gibco) supplemented with 10% FBS, 10 µg ml⁻¹ insulin, 5.5 µg ml⁻¹ transferrin, 5 ng ml⁻¹ selenium, 40 ng ml⁻¹ dexamethasone and 1% penicillin–streptomycin.

**Human immortalized hepatocytes.** PTA-5565 cells (ATCC) stably labelled by H2B–mCherry were obtained from the Bentires-Alj laboratory (University of Basel) and cultured in William's E Medium supplemented with 1% GlutaMax (Gibco), 5% FBS and 1% penicillin–streptomycin.

**KPC cells.** *Ptf1a-Cre;Kras^G12D/+;Trp53^flox/+* (KPC) pancreatic ductal adenocarcinoma cells (B6J syngeneic) were donated by I. Guccini and cultured in 2D cultures in DMEM:F12 supplemented with 10% FBS and 1% penicillin–streptomycin. Cells were split every 3–5 days and on the day before surgery. Before intrasplenic injection, KPC cells were detached from culture flasks with 1 mM EDTA.

**Melanoma cells.** The *Tyr-CreER;Braf^CA;Pten^lox/lox* (D4M-3A) B6 mouse melanoma line was generated previously[76], obtained from Merck Millipore and cultured in Advanced DMEM:F12 supplemented with 10% FBS and 1% penicillin–streptomycin and 1× GlutaMax (Gibco).

All of the cell lines were tested for mycoplasma, no cell lines were authenticated.

## In vitro assays

**Arrayed screen with primary hepatocytes or AML12 cells.** Plates (96 or 384 well) were coated with the Collagen-I Cell Culture Surface Coating Kit (ScienCell Research Laboratories) according to the manufacturer's instructions. Primary mouse hepatocytes from *Alb-cre;dCas9-SPH* mice were isolated by perfusion as described above. After two washes in isolation buffer, the hepatocyte pellet was further purified by density separation according to a published protocol[77]. In brief, the pellet was resuspended in 10 ml isolation buffer and 10 ml Percoll solution (9 ml Percoll, 1 ml 10× PBS), then mixed thoroughly by inverting the tube several times. After centrifugation at 200*g* for 10 min at 4 °C, the hepatocytes were resuspended in isolation medium (supplemented with 1% penicillin–streptomycin) and plated at high density (15,000 hepatocytes per well in 96-well plates, 5,000 hepatocytes per well in 384-well

plates). The same plating density was used for AML12[dCas9-SPH] cells, which were generated introducing doxycycline-inducible dCas9-SPH into the Rosa26 safe-harbour by recombinase-mediated cassette exchange[78,79] and kept in culture with 2 µg ml[−1] doxycycline. The next day, primary *Alb-cre;dCas9-SPH* hepatocytes or AML12-SPH were transfected with SB100X and transposon vectors harbouring sgRNAs against selected gene targets using Lipofectamine 3000 (Thermo Fisher Scientific). For every target, three sgRNAs were independently cloned and amplified into transposon vectors, and then pooled before transfection. Three wells were transfected for each target, and three wells were left untransfected. The next day, the transfection efficiency was estimated on the basis of GFP fluorescence. AKPS[sLP-mCherry] organoids were dissociated into single cells as described above, then 50 cells were seeded per well. After 5 days, colony formation was assessed by microscopy.

**Stable OE of *Plxnb2*.** AML12[dCas9-SPH] were treated for a week with 2 µg ml[−1] doxycycline and then transfected with SB100X and a pool of three PT4-U6-sgRNA-CMV-GFP transposon vectors targeting *Plxnb2*, or *sgNT*, using Lipofectamine 3000 (Thermo Fisher Scientific). On the next day, cells were sorted for GFP[+] fluorescence on the Aria III sorter (BD Biosciences) with a 70 µm nozzle and used for CFU assays as described above. *Plxnb2* upregulation was assessed by RT−qPCR. To generate the pLVX-VSV-mmPLXNB2-IRES-Blast vector, the mouse *Plxnb2* coding sequence was amplified from pmPB2-VSV (Roland Friedel, Addgene, 68038) and cloned into a BamHI/EcoRI-linearized and engineered pLVX-backbone (Takara), bearing BlasticidinR, by InFusion (Takara; InF-VSV-Plxnb2-F: TATTTCCGGTGAATTCACCATGTGGG TGACCAAACT and -R: GAGAGGGGCGGGATCTCAGAGGTCTGTAA CCTTATTCTCA). Lentiviruses were generated as described above and used to transduce AML12 cells. The correct membrane localization was assessed by immunofluorescence with rabbit anti-VSV-G antibody (Thermo Fisher Scientific, PA1-29903). VSV-G[+] cells were selected by FACS. Plexin B2 upregulation was assessed by flow cytometry using mouse anti-plexin-B2-PE (BioLegend, 145903).

**Treatment with recombinant mouse and human plexin B2.** Recombinant human plexin B2 (5329-PB-050, Biotechne) and mouse plexin B2 (6836-PB-050, Biotechne) were reconstituted at 100 µg ml[−1] in PBS. Human or mouse organoids were dissociated into single cells as described above, mixed with recombinant plexin B2 in growth medium, then seeded into 384-well plates at a density of 50 cells per well, in the absence or presence of 5,000 human or mouse hepatocytes. Colony formation was scored by microscopy. Where indicated, cultures were supplemented with 50 µM RAC1 inhibitor NSC23766 or 1 ng µl[−1] anti-plexin B2 monoclonal antibody (67265-1, Proteogenic). The EdU-incorporation assay was performed using the Click-iT EdU Cell Proliferation Kit for Imaging, Alexa Fluor 647 dye (Invitrogen). In brief, 3 days after treatment of PDOs with rhPlexin B2, half of the culture medium was removed and replaced with 2× EdU-containing medium for 1 h, then the manufacturer's instructions were followed.

**Treatment with KLF4 inhibitor.** AKPS organoids were seeded as single cells into 384-well plates in the presence of the KLF4 inhibitor WX2-43 (10 µM, Aobious). Then, 4 days later, colonies were fixed in 4% PFA, blocked and stained overnight with rabbit anti-Ki-67 (Abcam, ab15580). After washing, wells were incubated with anti-rabbit Alexa Fluor 594 (Thermo Fisher Scientific), F-actin was stained with Alexa Fluor 647 Phalloidin (1:400, Invitrogen, A22287) and nuclei with DAPI.

**Generation of AAVs**
Three sgRNA sequences for dCas9-SPH-mediated OE of *Plxnb2* were obtained from the Caprano library[55], four sgRNA sequences for Cas9-mediated knockout (KO) and two control non-targeting sgRNA were obtained from the Brie library[80]. A list of the sequences is provided in Supplementary Tables 1 and 3. Each guide was cloned into

U6-sgRNA-EF1a-eGFP (Addgene, 117046) vector and amplified using the Maxi prep kit (Macherey-Nagel). To generate the AAV-mmALBpr-Cre-2A-moxGFP vector, the mouse *Alb* promoter was amplified from pALB-GFP (S. Thorgeirsson, Addgene, 55759) and cloned by InFusion (Takara; InF-mmAlb-F: CTGCGGCCGCACGCGTCTAGCTTCCTTAGCA TGACGTTCCA and -R: GCATGGTGGCACCGGTGGGGTTGATAGGAA AGGTGATCTGT) into an AAV8 backbone, provided by A. Santinha (Platt laboratory, ETHZ-BSSE). GFP was replaced by PCR-out and InFusion with moxGFP (E. Snapp, Addgene, 68072; InF-noATG-moxGFP-F: AGGAGGTAGCGGATCCGTGTCCAAGGGCGAGGAG and R: TAGCGCTC GGTATCGATTTACTTGTACAGCTCGTCCATGCC). AAVs were generated and purified according to a slightly modified version of the AddGene protocols 'AAV Production in HEK293T Cells' and 'AAV Purification by Iodixanol Gradient Ultracentrifugation'. In brief, 250 million HEK293T cells were seeded in a Five Chambers Cell-Stack (Corning). Then, 24 h later, the vectors were pooled and co-transfected with pAdH helper plasmid and pAAV2/8 capsid (Addgene, 112864) using polyethylenimine (PEI). After viral genome production and purification, total viral genomes were quantified using digital droplet PCR according to the Addgene protocol 'ddPCR Titration of AAV Vectors'. AAVs were injected into the tail vein as described above.

**Immunofluorescence**
**Formalin-fixed and embedded tissues.** Livers were perfused with PBS, then the medial lobe was fixed with 4% PFA in PBS for 48 h, followed by 48 h PBS incubation and storage in 75% ethanol. Dehydration, formalin embedding and H&E staining were performed by the histology core facility of the University of Basel. Sections (5 µm) were deparaffinized with descending alcohol series and subjected to heat-induced epitope retrieval in 2.4 mM sodium citrate and 1.6 mM citric acid, pH 6, for 25 min in a steamer. Sections were washed with PBST (0.1% Tween-20 in PBS) and blocked for 1 h at room temperature in blocking buffer (5% BSA, 5% heat-inactivated normal goat serum in PBST). Sections were incubated overnight at 4 °C with the following primary antibodies (1:100, in blocking buffer): anti-CD146 (Abcam, ab75769), anti-α-SMA (Abcam, ab5694), anti-periostin (Abcam, ab227049) and anti-GFP (Aves Labs, GFP-1020). Sections were repeatedly washed in PBST and incubated with the following secondary antibodies (1:400, in blocking buffer) for 1 h at room temperature: AlexaFluor goat anti-rabbit 594 (A-11012), AlexaFluor goat anti-rabbit 647 (A-21244), Alexa Fluor goat anti-chicken 647 (A32933), all from Thermo Fisher Scientific. Nuclei were stained with DAPI (Sigma-Aldrich, 1:1,000) in blocking buffer for 15 min at room temperature. The sections were mounted with ProLong Gold (P36930, Invitrogen) and imaged on the Leica THUNDER Imager 3D Cell Imaging system, equipped with the Leica LED8 Light engine, Leica DFC9000 GTC sCMOS camera and the following filter sets: filter cube CYR71010 (excitation: 436/28, 506/21, 578/24, 730/40; dichroic: 459, 523, 598, 763; emission: 473/22, 539/24, 641/78, 810/80); filter cube DFT51010 (excitation: 391/32, 479/33, 554/24, 638/31; dichroic: 415, 500, 572, 660; emission: 435/30, 519/25, 594/32, 695/58) and extra emission filters (460/80, 535/70, 590/50, 642/80, 100%).

**Fixed frozen tissue.** After liver perfusion with PBS, the left lobe was incubated in 4% PFA at 4 °C for 1 h, then in 30% sucrose in PBS overnight at 4 °C and then embedded in Tissue-Tek OCT Compound (Sakura, 4583) for cryosectioning. Sections (8 µm) were washed three times, blocked and stained as described above with the following primary and secondary antibodies: anti-glutamine synthetase (1:100, BioLegend 856201), anti-plexin-B2-PE (1:500, BioLegend, 145903), anti-GFP-AlexaFluor-488 (1:200, Thermo Fisher Scientific, A-21311), anti-ZEB1 (1:400, Novus, NBP1-05987), anti-α-SMA (1:1,000, Sigma-Aldrich, A2547), anti-CD146 (Abcam, ab75769), anti-E-cadherin (Biotechne, AF748), anti-EPCAM (Abcam 2884975), anti-GRHL2 (Abcam, ab271023), anti-KLF4 (Biotechne, AF3158), anti-ELF3 (Thermo Fisher Scientific, PA5-120996) and anti-Sema4A (BioLegend, 148402). DAPI counterstain, mounting

and imaging was performed as described above. F-actin was stained by incubating blocked slides for 2 h at room temperature with Alexa Fluor 647 Phalloidin (1:400, Invitrogen, A22287).

**Multiplexed immunofluorescence and quantification.** Multiplexed immunofluorescence was performed on the Comet instrument (Lunaphore) with the following antibodies: anti-cleaved caspase 3 (Cell Signaling, 9661), anti-CD68 (Abcam, ab125212), anti-CD4 (Abcam, ab183685), anti-Ki-67 (Abcam, ab15580), anti-E-cadherin (Cell Signaling, 3195), anti-α-SMA (1:1,000, Sigma-Aldrich, A2547), anti-CD146 (Abcam, ab75769). The fields of view (FOVs) containing individual liver metastases were cropped and saved using the HORIZON software (Lunaphore). Each condition (sgNT or sgPlxnb2 OE) had a minimum of five FOVs representing five different lesions taken from two mice. The individual FOVs were analysed in FIJI. In brief, each channel was thresholded manually, followed by application of a median filter for signal smoothing and filling of holes. Each image was overlaid with its corresponding thresholded image to verify the accuracy of the thresholding. The region corresponding to tumour within a FOV was demarcated as a ROI and the area covered by a specific antibody signal was quantified as the number of pixels within the thresholded image with respect to the total number of pixels within the ROI. For quantification of dividing tumour cells, signals from both Ki-67 and ECAD were used: the overlap between the two signals was calculated, then the area of dividing tumour cells was determined as Ki-67$^+$ pixels within the overlap area over the area occupied by the nuclei of all cells (calculated using DAPI as a marker).

**Organoids.** After fixation in 4% PFA at 4 °C for 2 h, and blocking in 5% BSA-PBS solution with 0.2% Triton X-100, the samples were stained with primary (overnight) and secondary antibodies (4 h) (anti-ZEB1 (1:400, Novus, NBP1-05987), anti-E-cadherin (1:200, BD Biosciences, 610181)) and DAPI counterstain, and mounted in 3% low-melting-point agarose in glass-bottom plates and then imaged.

## Quantification of metastatic foci and lesion area

H&E sections were imaged on the Leica DMi8 inverted microscope, equipped with a FLEXACAM C1 12 MP CMOS camera and analysed using QuPath software[81]. Whole-tissue area and single-liver metastases were manually isolated, producing a measure for whole-section area, metastatic area (μm$^2$) and metastasis number per section. Two non-consecutive sections quantified per animal, and a mean was calculated for the number of metastatic foci per liver section.

## In situ hybridization

**Single-molecule in situ hybridization.** Custom DNA smFISH probes for *Plxnb2* were designed in house and synthesized by Biosearch Technologies containing a 3′ amine reactive group (a list of probes is provided in Supplementary Table 3). All of the probes were pooled and labelled with AlexaFluor 594 dye according to a previously published protocol[82]. Mouse tissues were collected and fixed with 4% PFA in PBS for 3 h followed by overnight incubations in 30% sucrose, 4% PFA in PBS at 4 °C. Fixed tissues were embedded in Tissue-Tek OCT Compound (Sakura, 4583). Tissue sections (8 μm) were sectioned onto poly-L-lysine-coated coverslips, allowed to adhere by drying at room temperature for 10 min, followed by 15 min fixation in 4% PFA and overnight permeabilization in 70% ethanol. Probe hybridization was performed according to a previously published protocol[82]. Images were acquired using a ×63 oil-immersion objective with NA = 1.4 on the Leica THUNDER Imager 3D Cell Imaging system, equipped with a Leica LED8 Light engine and Leica DFC9000 GTC sCMOS camera. For quantification, 3–4× FOVs covering the entire width of the tissue were acquired for each sample and the images were processed using the Thunder deconvolution algorithm. Maximum-intensity projections of the processed images were rendered using ImageJ. Dot counting to determine the transcript numbers for each FOV was performed with FISHQuant[83] using the automatic thresholding function and the cell number was determined by segmenting and counting the nuclei using CellPose[84]. Spot counting and nucleus numbers were manually verified to ensure correctness. The average number of spots per cell was then measured by dividing the number of spots within the FOV by the number of nuclei.

**Multiplexed in situ hybridization (Molecular Cartography).** Probe design, sample preparation imaging and processing were conducted as previously described[85]. The analysis of the data, including cell segmentation, cell type annotation and portal versus central area annotation was described previously[86]. Visualizations were generated in ImageJ using genexyz Polylux tool plugin from Resolve BioSciences.

## RT−qPCR

RNA extraction from fresh organoids, cells or liver tissue was performed using the Qiagen RNeasy purification kit. Then, 1 ng of total RNA was reverse transcribed using the cDNA synthesis kit (Takara Bio) according to the manufacturer's instructions. Expression of genes of interest was quantified with primers listed in Supplementary Table 3, by RT−qPCR using the Applied Biosystems SYBR Green Kit monitored by the QuantStudio3 system (Applied Biosystems). The samples were analysed in technical triplicates and the average cycle threshold values were normalized to *Gapdh* using the $\Delta\Delta C_T$ method[87].

## Bulk RNA-seq experiments and analysis

**Library preparation.** RNA was extracted as described above from the livers of *Alb-cre;SPH* mice 7 days after injection with *AAV8-sgPlxnb2-OE-EF1a-eGFP* or *AAV8-sgNT-EF1a-eGFP*, livers of B6 mice bearing AKPS colon tumours, and organoids. Libraries for bulk RNA-seq were prepared using the mcSCRB-seq protocol[88] (organoids) or the Takara SMART-Seq Stranded Kit (634762, mouse livers). Libraries were quality-controlled using the dsDNA high-sensitivity kit (Life Technologies, Q32854) on the Qubit 4 fluorometer (Thermo Fisher Scientific) and using the high-sensitivity D1000 reagents and tapes (Agilent, 5067-5585, 5067-5584) on the TapeStation 4200 (Agilent Technologies) and sequenced on the NovaSeq 6000 (Illumina) system using the NovaSeq SP Reagent Kits (100 cycles).

**Analysis.** Reads were demultiplexed with Bcl2fastq v.2.20.0.422 (Illumina) and quality-checked with FastQC[89]. Adaptors were trimmed with cutadapt[57]. Data were processed using the zUMIs (v.2.9.4) platform to convert reads to count matrices per sample. Differential gene expression analysis was performed using edgeR[90]. GSEA was performed using the Bioconductor package fgsea with the default parameters on genes ranked by log[fold change][91]. The Gene Ontology Biological Process and Hallmarks gene set collections from the Molecular Signatures Database were imported into R using the package msigdbr[92]. Cell type composition was estimated for significantly up- and downregulated genes in Enrichr[93] using Tabula Muris[94] as a reference (odds ratio test).

## scRNA-seq

**Library preparation.** AKPS organoids were dissociated into single cells and incubated with 2 μg ml$^{-1}$ rmPlexin B2 or vehicle in culture medium for 2 h at 37 °C. Cells were filtered, counted and loaded onto the GemCode Single-cell Instrument (10x Genomics). Libraries were generated according to the manufacturer's instructions from the Chromium Next GEM Single Cell 3′ end Reagent Kits v1.1 protocol. The quality and quantity of all of the libraries were assessed using the dsDNA high-sensitivity (HS) kit (Life Technologies, Q32854) on the Qubit 4 fluorometer (Thermo Fisher Scientific) and using the high-sensitivity D1000 reagents and tapes (Agilent, 5067-5585, 5067-5584) or high sensitivity D5000 reagents and tapes (Agilent, 5067-5593, 5067-5592) on the TapeStation 4200 system (Agilent Technologies). Paired-cell

sequencing was performed for all libraries using the NovaSeq SP Reagent Kits (100 cycles).

**Analysis.** BCL files were demultiplexed using Bcl2fastq v.2.20.0.422 from Illumina, then single-cell count matrices were generated using Cell Ranger (v.5.0.0, 10x Genomics) with GRCm38 v.2020-A gene code. Datasets were integrated and processed using Seurat. Downstream analysis was conducted in R (v.4.1.0) using the Seurat (v.4.0.3197) package. The Seurat objects (rmPlexinB2-treated and control) were merged and cells with <100 or >2,500 detected genes were excluded. After log-normalization, the data were scaled regressing for mitochondrial reads, and principal component analysis was performed based on the 2,000 most variable features. Clustering and UMAP visualization were performed using ten principal components and a resolution of 0.2 for the shared nearest-neighbour clustering algorithm. Cluster markers were computed using the FindAllMarkers function, and GSEA was performed as described above. KLF4-target genes were obtained from the CHEA Transcription Factor Binding Site Profiles database[95] and computed using the AddModuleScore function. EMT and MET signatures were obtained from the GO Biological Process dataset. A list of all signatures and gene sets is provided in Supplementary Table 3.

### snRNA-seq and snATAC–seq

**Nucleus extraction and library construction.** Combined profiling of gene expression and chromatin accessibility was performed from fresh frozen OCT-embedded livers. For each sample (2 sgNT and 2 sgPlxnb2 OE livers), three 50 μm liver sections were transferred into a prechilled gentleMACS C-tube (Miltenyi) and homogenized in the gentleMACS Octo Dissociator with 2 ml nucleus extraction buffer (Miltenyi). The nucleus suspension was filtered through a 70 μm SmartStrainer into a DNA-low-binding 5 ml tube (Eppendorf) and centrifuged at 150*g* for 3 min, at 4 °C. The pellet was resuspended in 5 ml 1% BSA in PBS and strained through a 30 μm SmartStrainer into a new tube. After centrifugation, the pellet was washed again in 5 ml 1% BSA in PBS. Nuclei were resuspended in 500 μl 1% BSA in PBS, counted and visually inspected. 16,000 nuclei per sample were profiled using the Chromium Single Cell Multiome ATAC + Gene Expression kit (10x) according to manufacturer's instructions. Libraries were quality controlled and sequenced as described above.

**Analysis.** BCL files were demultiplexed using Bcl2fastq v.2.20.0.422 from Illumina, then single-nucleus count matrices were generated using Cell Ranger Arc (10x Genomics) with GRCm38 v2020-A gene code. RNA and chromatin profiles of the four datasets were integrated with Signac[96] (v.1.12.0) using the FindIntegrationAnchors function. Ambient RNA was removed with the decontX package[97] (v.1.0.0), then cell types were annotated based on the RNA profile. Tumour cells were subsetted and DEGs were calculated using the FindMarker function, and GSEA was performed as described above. Chromatin peaks were called with the CallPeaks function, then differentially open peaks and motifs were identified using the AddPeaks, FindPeaks and FindMotifs functions.

### Analysis of class IV semaphorins in human CRC

**Protein atlas stainings.** SEMA4A, SEMA4D, SEMA4C and SEMA4G antibody stainings were obtained from the Human Protein Atlas[42] with the R package HPAanalyze[98].

**CNV analysis of class IV semaphorin genes.** The CNV status of *SEMA4A*, *SEMA4D*, *SEMA4C* and *SEMA4G* in 290 patients with CRC was obtained from the TGCA-COAD dataset[45].

**Published scRNA-seq datasets.** Preprocessed and annotated scRNA-seq profiles of epithelial cells in the KUL and Samsung dataset were obtained from the Synapse repository syn34942428[43]. scRNA-seq datasets of matched liver metastases and primary tumours were obtained from the Gene Expression Omnibus under the accession numbers GSE225857 (ref. 21) and GSE178318 (ref. 22), and imported into Seurat. Epithelial cells were subsetted on the basis of *EPCAM* expression. Averaged expression of *SEMA4A*, *SEMA4D*, *SEMA4C* and *SEMA4G* was computed using the AddModuleScore function.

**Kaplan–Meier analysis.** Kaplan–Meier analysis of 1,211 patients with CRC was performed using an online tool (http://kmplot.com). Recurrence-free survival was stratified by *SEMA4A*, *SEMA4C, SEMA4D* and *SEMA4G* expression in the Affymetrix colon dataset, using best cut-off.

### Statistics and reproducibility

Statistical analysis and visualization were performed using R (v.4.1.0, R Foundation for Statistical Computing), R Studio and the package ggplot2[62] or Prism v.8.2.0. Statistical significance tests were performed as described in each figure legend, and *P* values were adjusted for multiple testing. Micrographs are representative of multiple biological replicates ($n \geq 2$).

### Reporting summary

Further information on research design is available in the Nature Portfolio Reporting Summary linked to this article.

## Data availability

The sequencing data generated in this study are available at the Gene Expression Omnibus under the accession numbers GSE267981 and GSE267982 and at Zenodo[99] (https://doi.org/10.5281/zenodo.7737590). Source data are provided with this paper.

## Code availability

The code used in this study is available at GitHub (https://github.com/Moors-Code/coco_mosaic_liver).

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

**Acknowledgements** We thank A. Ozga and G. Aguilar and all of the members of the Moor laboratory for discussions and insights; the staff at the Single Cell Facility and Animal Facility of D-BSSE and the members of the laboratories of K. Basler, M. van den Broek, G. Schwank and R. Platt for technical support; O. Sansom for donating the AKP organoids; F. Greten for donating the APTAK organoids; M. Bentires-Alj for providing the human immortalized hepatocytes; I. Guccini for donating the KPC cells; and T. Stühmer for donating the shRNA plasmid backbone. This work was funded by the Swiss National Science Foundation (PCEFP3_181249) to A.E.M. and the Swiss Cancer League (KFS-5444-08-2021-R) to C.B. and A.E.M. T.V. was partially supported by the National Institute for Cancer Research (Programme EXCELES, LX22NPO5102) funded by the European Union (Next Generation EU). A. Meijs and R.J.P. are supported by the National Centres of Competence Molecular Systems Engineering (51NF40-182895). S.P. is funded by the Swiss Cancer Research foundation (KFS-4988-02-2020-R) and by the Prof. Dr. Max Cloëtta Foundation.

**Author contributions** C.B. conceived the study, performed and analysed the screen and key validation experiments, and wrote the manuscript. M.R., M.-D.H., H.F., T.V., A.K. and F.B.M. performed mouse experiments. D.E. cloned and designed all vectors. A.L., S.A., E.G.V., K.H. and I.E.A. performed and assisted with bioinformatic analyses. J.A.K. generated the Visium dataset. A. Mannhart and S.K. performed in vitro assays. A. Meijs and R.J.P. helped with enAsCas12a experiments. X.F. edited the manuscript. S.P. donated the patient-derived organoids. A.E.M. conceived, supervised and acquired funding for the study.

**Funding** Open access funding provided by Swiss Federal Institute of Technology Zurich.

**Competing interests** The authors declare no competing interests.

**Additional information**
**Correspondence and requests for materials** should be addressed to Andreas E. Moor.

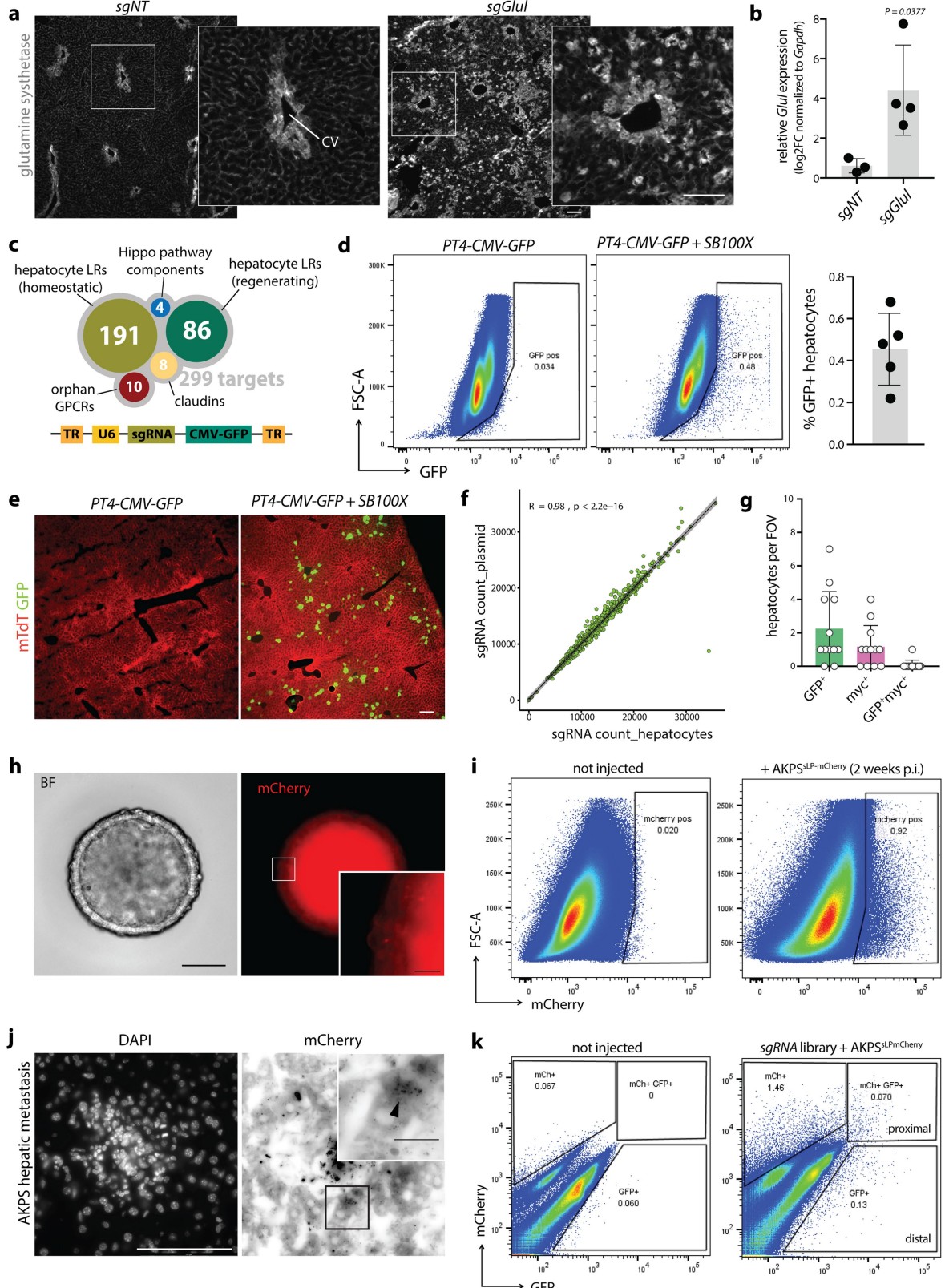

**Extended Data Fig. 1** | See next page for caption.

**Extended Data Fig. 1 | Screening tumour-hepatocytes interactions in a mosaic liver. a**, Glutamine synthetase immunoreactivity in *AlbCre;dCas9-SPH* mice upon co-injection of SB100X transposase and *PT4-U6-sgGlul* or *PT4-U6-sgNT*. Scale bar, 100 μm. CV, central vein. **b**, *Glul* gene expression of whole liver extracts of *AlbCre;dCas9-SPH* mice upon co-injection of SB100X and *PT4-U6-sgGlul* (n = 4) or *PT4-U6-sgNT* (n = 3), as assessed by qRT-PCR. Log2FC normalized to *Gapdh*, two-tailed unpaired t-test. **c**, Top, genes targeted by the sgRNA library (see Table S1). Bottom, sgRNA transposon vector design. TR, terminal repeats. **d**, Representative flow cytometry plot and quantification of GFP⁺ hepatocytes (gated as CD45⁻ CD31⁻) one week after HTVI of SB100X transposase and *PT4-CMV-GFP* transposon in *AlbCre;dCas9-SPH* mice (n = 5). **e**, GFP expression one week after HTVI. Scale bar, 100 μm. Membrane TdTomato (mTdT) in red. **f**, sgRNA abundance in plasmid vs. sorted GFP⁺ hepatocytes. Two-tailed Pearson's correlation test. **g**, Quantification of single and double positive hepatocytes in livers co-injected with transposon vectors harbouring myc or GFP reporters (n = 12 FOVs, pooled from 2 mice). **h**, Representative micrograph of AKPS organoids expressing sLP-mCherry. Scale bar, 50 μm. Inset shows sLP-mCherry punctae in red. Scale bar, 10 μm. **i**, Representative flow cytometry plots of mCherry niche labelling in vivo upon AKPS intrasplenic injection. Gated on CD31⁻CD45⁻ hepatocytes. **j**, Fluorescent micrograph of AKPS^sLP-mCherry metastases 1 week post intrasplenic injection. Nuclei are labelled with DAPI. Scale bar, 100 μm. Inset and arrowhead indicate sLP-mCherry punctae (in black) in proximal hepatocytes. Scale bar, 20 μm. **k**, Representative FACS plots and gating strategy for the isolation of hepatocytes (CD31⁻CD45⁻) harbouring a sgRNA (GFP⁺) that are proximal to (mCherry⁺) or distal from (mCherry⁻) metastases. b,d,g, Barplots indicate mean ± s.d.

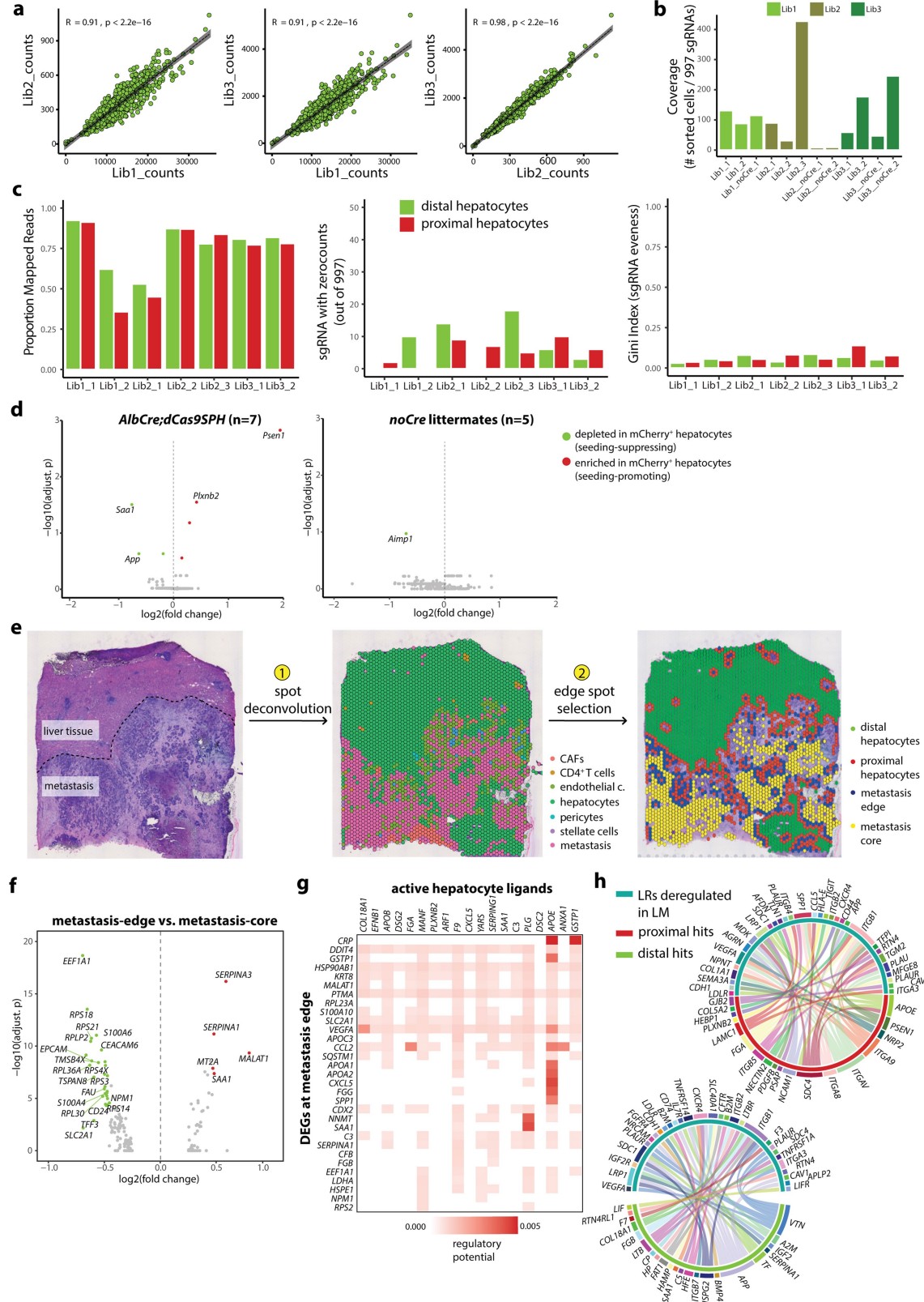

**Extended Data Fig. 2** | See next page for caption.

**Extended Data Fig. 2 | In vivo CRISPR-a screen identifies hepatocyte-derived factors that regulate metastatic seeding. a**, Two-tailed Pearson correlation of sgRNA abundance in three independent library batches (Lib_1, Lib_2 and Lib_3). **b**, sgRNA coverage per mouse, calculated by dividing the number of sorted cells by 997 (sgRNAs) (n = 7 *AlbCre;dCas9-SPH* mice and n = 5 *nonCre* littermate controls). **c**, Proportion of reads mapped (left), zero count sgRNAs (middle), and Gini index (right) of sgRNA libraries amplified from distal (mCherry⁻, green) and proximal (mCherry⁺, red) hepatocytes across three independent batches and recipient *AlbCre;dCas9-SPH* mice (n = 7). **d**, Volcano plots showing aggregated results for all *AlbCre;dCas9-SPH* mice (left) and *nonCre* littermate controls (right). Red dots indicate seeding-promoting factors enriched in the proximity of metastases, green dots indicate seeding-suppressing factors depleted in the proximity of metastases. LogFC and adjusted p value calculated with unpaired robust rank aggregation (α-RRA) from the MAGeCK algorithm[60]. **e**, H&E staining (left) of a human CRC liver metastasis sample used for Visium spatial transcriptomics (10x Genomics) (1 representative image shown of 2 analysed replicates). After spot deconvolution, spots are annotated according to the predominant cell types, indicated in a colour-coded map (middle). The annotation is used as input for edge spot selection, identifying metastasis-proximal and distal hepatocytes as well as metastasis core vs. edge (right). **f**, DEGs between metastasis-edge and metastasis-core spots. Two-tailed unpaired Wilcoxon test. Bonferroni correction for multiple testing. **g**, Top 20 active hepatocyte ligands and their regulatory potential on DEGs at metastatic edge (analysis conducted with Nichenet[65]). **h**, Predicted interactions between top-scoring hits of the screen (proximal hits in red in top plot, distal hits in green in bottom plot) and cognate LRs whose expression is altered (logFC > |0.25|, adjusted P < 0.01, two-sided Wilcoxon test) between epithelial cells in liver metastases (LM) vs. matched primary tumours in two independent datasets of metastatic CRC (Wang et al. n = 5[21], Che et al. n = 6[22]).

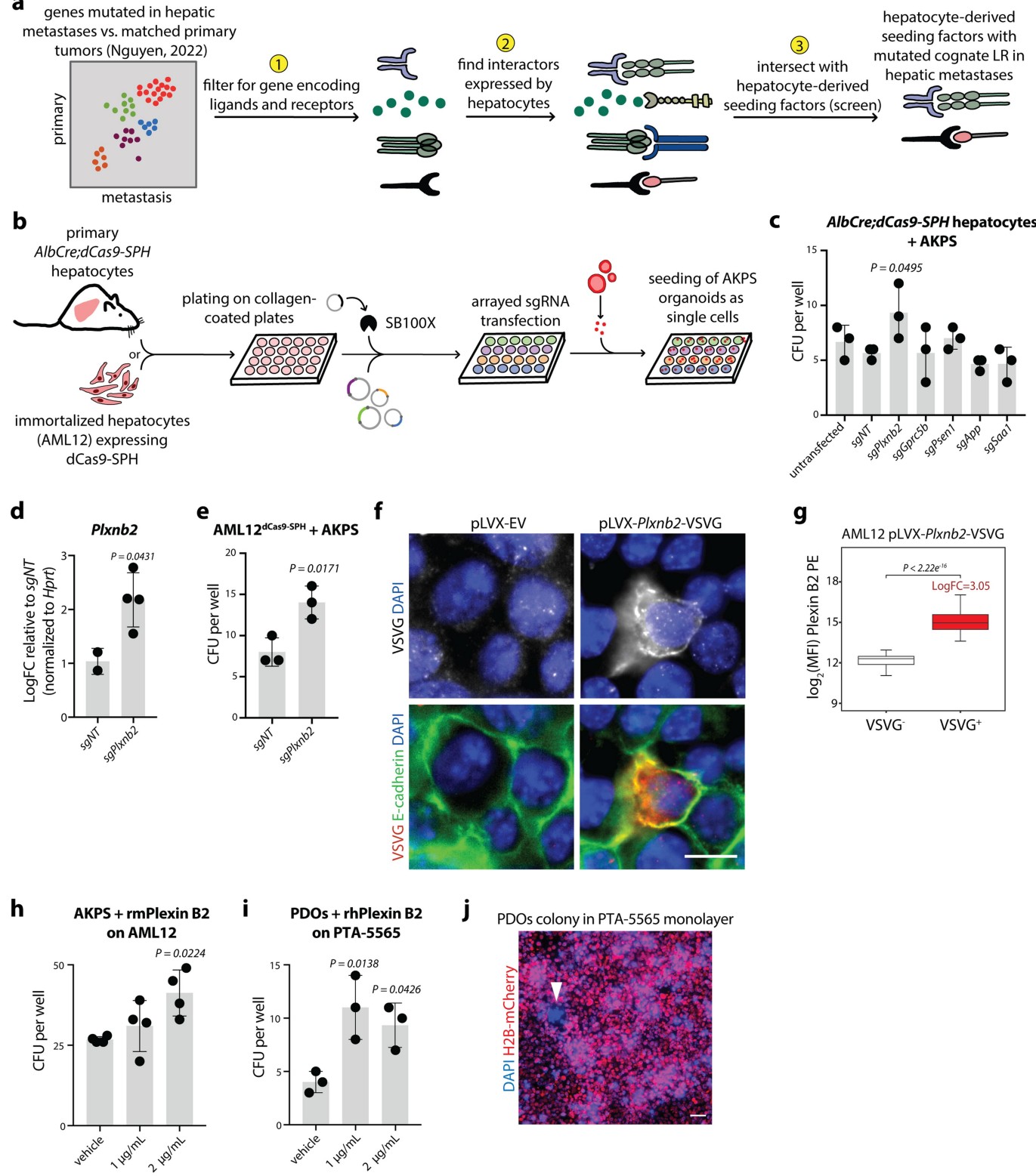

**Extended Data Fig. 3** | See next page for caption.

**Extended Data Fig. 3 | In vitro screen and Plexin B2 overexpression.**
**a**, Schematic representation of workflow for mutational analysis. 1) Frequently mutated genes in hepatic metastases vs. primary tumours in a publicly available dataset[20] are filtered for LR-encoding genes. 2) Cognate interaction partners are identified by cell-cell interaction prediction tools and filtered for hepatocyte expression and 3) intersected with SRFs. **b**, Experimental workflow for in vitro arrayed screens. 1) primary *AlbCre;dCas9-SPH* hepatocytes or AML12[dCas9-SPH] are plated on collagen-coated plates and 2) transfected with sgRNA-encoding transposon vectors and SB100X, prior to 3) seeding of dissociated AKPS[sLP-mCherry] organoids. After 5 days, CFUs are scored by fluorescent microscopy. **c**, CFU per well of AKPS[sLP-mCherry] organoids after co-culture with primary *AlbCre;dCas9-SPH* hepatocytes transfected with indicated sgRNAs. Dots indicate individual wells (n = 3 per condition). **d**, *Plxnb2* gene expression in AML12[dCas9-SPH] transfected with *sgPlxnb2* (n = 4) or *sgNT* (n = 2), as assessed by qRT-PCR. Log2FC normalized to *Hprt*. **e**, CFU per well of AKPS[sLP-mCherry] organoids after co-culture with AML12[dCas9-SPH] cells overexpressing *Plxnb2*. Dots indicate individual wells (n = 3 per condition). **f**, Representative fluorescent micrograph of VSVG (red) and E-cadh (green) immunoreactivity in AML12 transduced with *pLVX-Plxnb2-VSVG* or *pLVX-EV*. DAPI in blue as nuclear counterstain. Scale bar, 5 µm. **g**, Plexin B2-PE mean fluorescent intensity (MFI) in VSVG[+] and VSVG[-] AML12 cells, as assessed by flow cytometry. Two-tailed unpaired Wilcoxon test. Boxplots indicate median, first and third quartiles. Whiskers extend from the hinges to the largest value no further than 1.5× the inter-quartile range. **h**, CFU per well of AKPS organoids seeded as single cells on AML12 cells treated with rmPlexin B2. Dots indicate individual wells (n = 4 per condition). **i**, CFU per well of PDOs cultured on PTA-5565 cells treated with rhPlexin B2. Dots indicate individual wells (n = 3 per condition). **j**, Representative fluorescent micrograph of a PDO colony (indicated by arrowhead) growing on PTA-5565[H2B-mCherry] cells. DAPI in blue as nuclear counterstain. Scale bar, 20 µm. c-e, h, i, Barplots indicate mean ± s.d. c,h,i Ordinary one-way ANOVA, comparing each treatment group to vehicle control. Dunnet's correction for multiple testing. d,e Two-tailed unpaired t-test.

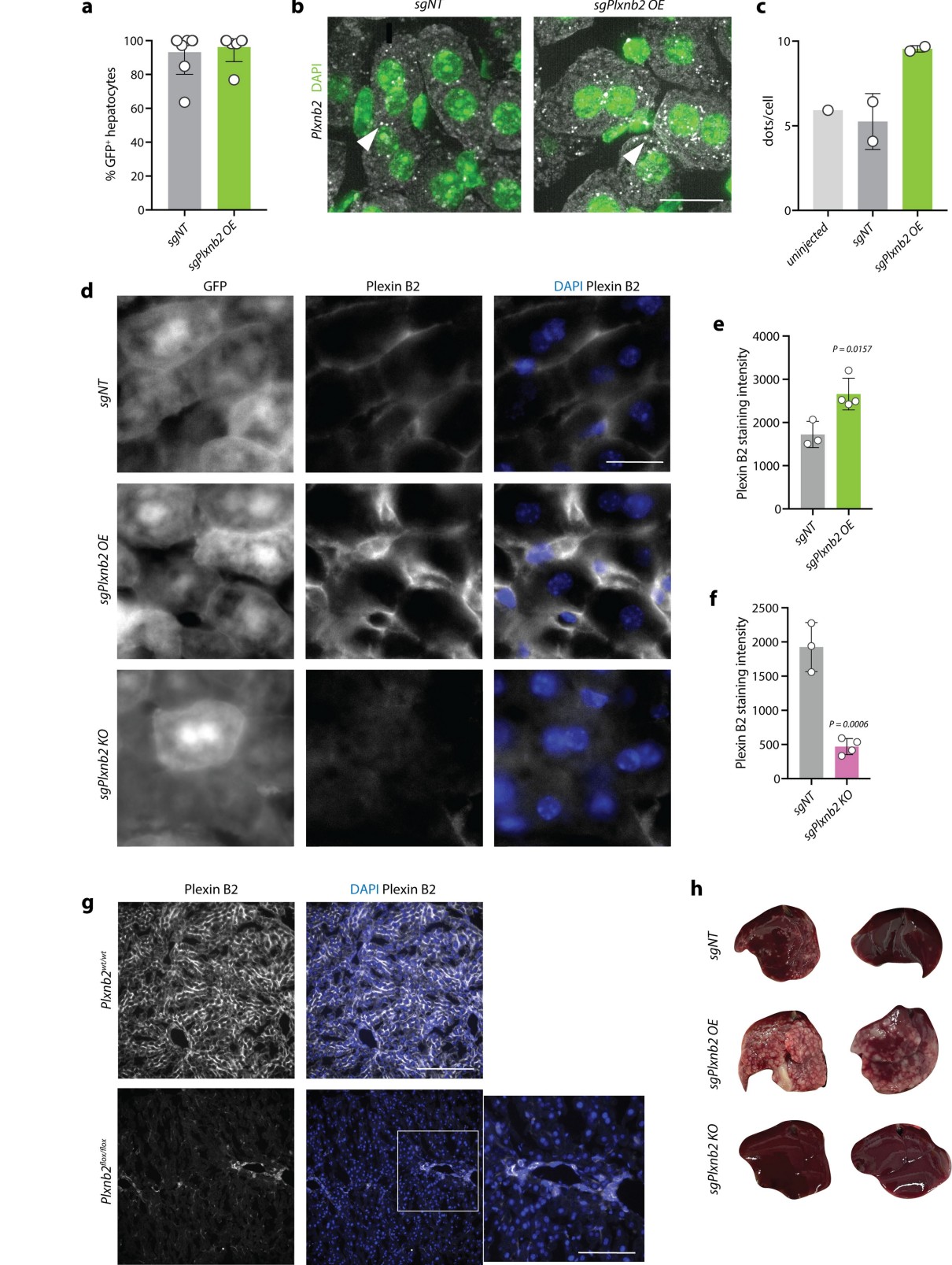

**Extended Data Fig. 4** | See next page for caption.

**Extended Data Fig. 4 | *Plxnb2* OE in mouse hepatocytes after AAV-mediated sgRNA transduction. a**, Percentage of GFP⁺ hepatocytes in *AlbCre;dCas9-SPH* mice injected with *AAV8-U6-sgPlxnb2OE-EF1a-EGFP* (*sgPlxnb2* OE, n = 5) or *AAV8-U6-sgNT-EF1a-EGFP* (*sgNT*, n = 7). **b**, Representative fluorescent micrograph of *Plxnb2* mRNA (white) as detected by smFISH. DAPI in green as nuclear counterstain. Arrowheads indicate single mRNA molecules. Scale bar, 20 μm. **c**, Quantification of *Plxnb2* mRNA molecules per cell, average of 3 fields of view per mouse in n = 2 mice per condition. **d,e,f**, Representative fluorescent micrographs and quantification of Plexin B2 immunoreactivity (grey) in *AlbCre;dCas9-SPH* mice injected with *AAV8-U6-sgPlxnb2OE-EF1a-EGFP* (*sgPlnxb2* OE, n = 4) or *AAV8-U6-sgNT-EF1a-EGFP* (*sgNT*, n = 5), and *AlbCre;Cas9* mice injected with *AAV8-U6-sgPlxnb2KO-EF1a-EGFP* (*sgPlxnb2* KO, n = 4). DAPI in blue as a nuclear counterstain. Scale bar, 20 μm. Two-tailed unpaired t-test. **g**, Representative fluorescent micrographs of Plexin B2 immunoreactivity (grey) in *Plxnb2^{wt/wt}* and *Plxnb2^{flox/flox}* mice injected with *AAV8-AlbCre-moxGFP*. DAPI in blue as a nuclear counterstain. Scale bar, 200 μm. Inset shows residual Plexin B2 on endothelial cells lining the central vein. Scale bar, 100 μm. **h**, Representative micrographs of AKPS metastases growing in WT, *Plxnb2* OE or KO livers. a,c,e,f, Dots represent individual mice. Barplots indicate mean ± s.d.

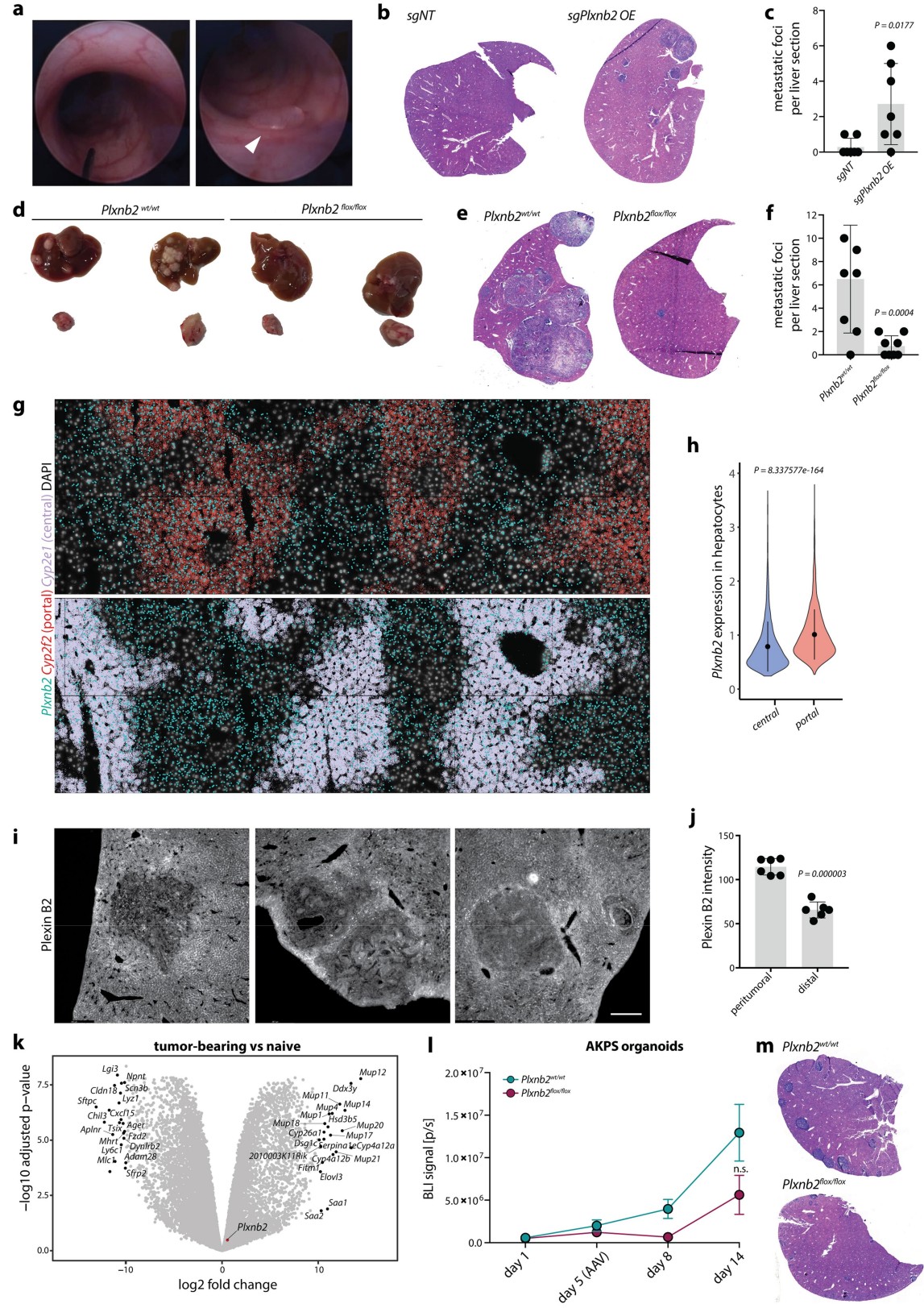

**Extended Data Fig. 5 |** See next page for caption.

**Extended Data Fig. 5 | Plexin B2 is required for spontaneous metastatic seeding. a**, Representative micrograph of colonoscopy-guided submucosal injection of CRC organoids. Arrowhead indicates the site of injection. **b,c**, Representative H&E section and quantification of spontaneous APTAK metastases in *AlbCre;dCas9SPH* mice injected with *sgPlxnb2 OE* or *sgNT* (n = 7 mice per group). Two-tailed unpaired T-test. **d**, Representative micrographs of primary APTAK tumours (bottom) and liver metastases (top) in *Plxnb2^wt/wt* and *Plxnb2^flox/flox* mice injected with *AAV8-AlbCre-moxGFP*. **e, f**, Representative H&E section and quantification of spontaneous APTAK metastases in *Plxnb2^wt/wt* and *Plxnb2^flox/flox* mice injected with *AAV8-AlbCre-moxGFP* (n = 7 mice per group). Two-tailed unpaired T-test. **g**, Multiplexed in situ hybridization (Molecular Cartography) of *Plxnb2* as well as portal and central hepatocyte markers in WT mouse liver. **h**, Quantification of *Plxnb2* mRNA abundance in cells from pericentral n = 89; periportal n = 66 areas pooled from 4 mice from. Two-tailed unpaired Wilcoxon test. **i**, Representative fluorescent micrograph of Plexin B2 (grey) immunoreactivity in metastatic livers. Scale bar, 100 μm. **j**, Plexin B2 staining intensity in peritumoral (2-3 hepatocyte layers) vs. distal areas. Dots represent individual peritumoral areas (n = 6) pooled from 3 mice per group. Two-tailed unpaired T-test. **k**, DEGs in livers of primary CRC-bearing vs. naive mice (n = 3 mice per condition). Quasi-likelihood F-test, FDR-adjusted p-values. **l**, BLI signal over time in *Plxnb2^flox/flox* (n = 4) or *Plxnb2^wt/wt* (n = 3) mice injected with AKPS^Luciferase;zsGreen organoids and subsequently *AAV8-AlbCre-moxGFP* (AAV). Two-tailed Fisher's exact test. Data presented as mean ± s.d. **m**, Representative H&E section of AKPS^Luciferase;zsGreen metastases in livers of *Plxnb2^flox/flox* or *Plxnb2^wt/wt* mice injected with *AAV8-AlbCre-moxGFP*. c,f,h,j,l, Data are presented as mean ± s.d.

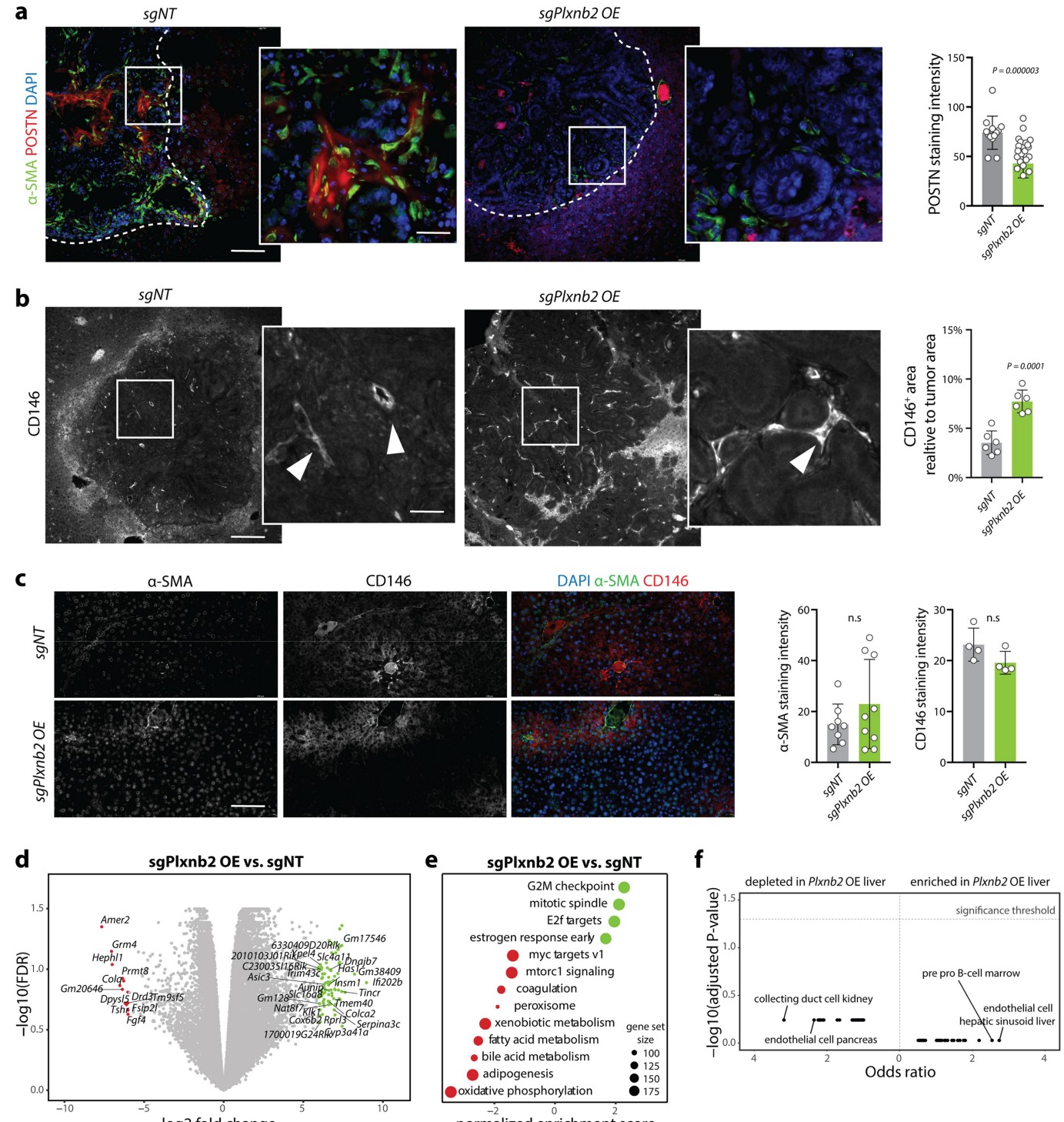

**Extended Data Fig. 6 | AKPS metastases in *Plxnb2* OE livers have an altered stromal and endothelial compartment. a**, Representative fluorescent micrographs and quantification of α-SMA (green) and periostin (POSTN, red) immunoreactivity in AKPS metastases in *Plxnb2* OE and control livers, DAPI in blue as nuclear counterstain. Dashed line indicates the tumour border. Scale bar, 100 μm, inset, 20 μm. Dots indicate average staining intensity of individual metastases (n = 20) pooled from 3 mice per group. **b**, Representative fluorescent micrographs and quantification of CD146⁺ immunoreactivity in AKPS metastases in *Plxnb2* OE and control livers. Arrowheads indicate vessels. Scale bar, 200 μm, inset, 100 μm. Dots indicate average staining intensity of individual metastases (n = 6) pooled from 3 mice per group. **c**, Representative fluorescent micrographs and quantification of α-SMA (green) and CD146 (red) immunoreactivity in *Plxnb2* OE and control livers, DAPI in blue as nuclear counterstain. Scale bar, 100 μm. Dots indicate average staining intensity of individual FOV (n = 8 for α-SMA, n = 4 for CD146) pooled from 2 mice per group. **d,e**, DEGs and GSEA in *Plxnb2* OE vs. control livers (n = 3 mice per group). Quasi-likelihood F-test, FDR-adjusted p-values. **f**, Cell type enrichment analysis using Tabula Muris as reference (Odds ratio test) in bulk RNASeq of *Plxnb2* OE vs. control livers (n = 3 mice per group). a-c, Two-tailed unpaired t-test, barplots indicate mean ± s.d.

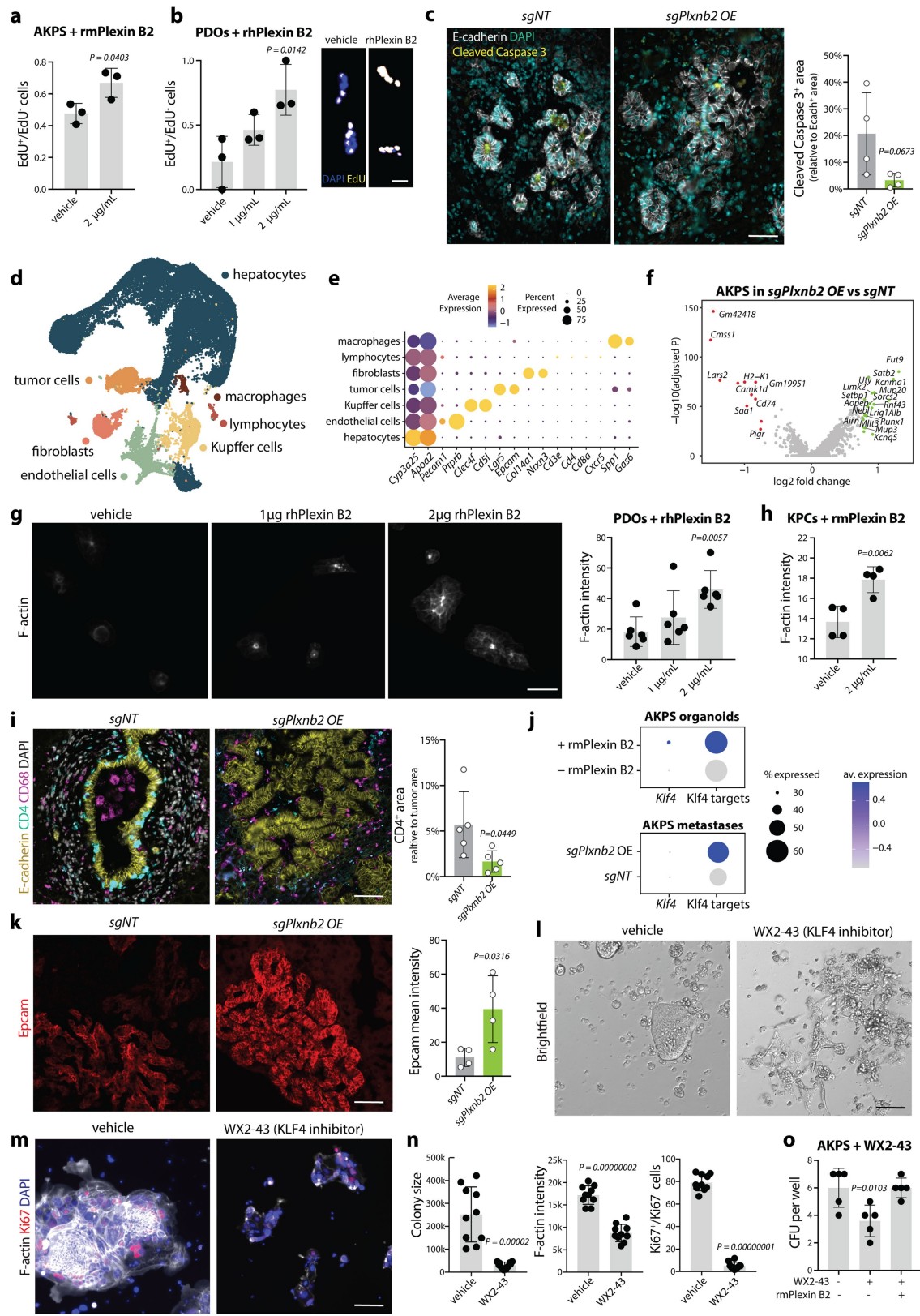

**Extended Data Fig. 7 |** See next page for caption.

**Extended Data Fig. 7 | Multiomic characterization of AKPS metastases in** *Plxnb2* **OE and control livers. a**, Edu⁺/Edu⁻ cells in 2D AKPS organoid cultures treated with rmPlexin B2. Dots represent individual wells (n = 3). **b**, Left, Edu⁺/Edu⁻ cells in 2D PDO cultures treated with rhPlexin B2. Dots represent individual wells (n = 3). Right, representative fluorescent micrograph of Edu⁺ PDO colonies. Scale bar, 10 μm. **c**, Representative fluorescent micrograph and quantification of E-cadh (grey) and cleaved caspase-3 (yellow) immunoreactivity in AKPS metastases in *Plxnb2* OE and control livers, DAPI in cyan as nuclear counterstain. Dots represent individual liver metastases (n = 4) pooled from 2 mice per group. **d**, UMAP of RNA profile of nuclei isolated from *Plxnb2* OE and control livers harbouring AKPS metastases. Dots indicate single nuclei, coloured by cell type. **e**, Marker gene expression across cell types. **f**, DEGs in tumour cells isolated from *Plxnb2* OE vs. control livers. Two-tailed unpaired Wilcoxon test. **g**, Representative fluorescent micrograph and quantification of F-actin in 2D PDO cultures treated with rhPlexin B2. Scale bar, 10 μm. Dots indicate individual wells (n = 6). **h**, F-actin intensity in KPC cultures treated with rmPlexin B2. Dots indicate individual wells (n = 4). **i**, Representative fluorescent micrograph and quantification of E-cadh (yellow), CD4 (cyan, T cells) and CD68 (magenta, macrophages) immunoreactivity in AKPS metastases in *Plxnb2* OE and control livers, nuclear DAPI stain in grey. Scale bar, 20 μm. Dots represent individual metastases (n = 5) pooled from 2 mice per group. **j**, Expression of *Klf4* and KLF4-target genes in AKPS organoids treated with rmPlexin B2, and of AKPS metastases grown in *Plxnb2* OE livers. **k**, Representative fluorescent micrograph and quantification of Epcam (red) immunoreactivity in AKPS metastases in *Plxnb2* OE and control livers. Scale bar, 20 μm. Dots represent individual metastases (n = 4) pooled from 3 mice per group. **l**, Representative brightfield micrograph of 2D AKPS cultures treated with the KLF4 inhibitor WX2-43. **m,n**, Representative fluorescent micrograph and quantification of F-actin (grey) and EdU (red) in 2D AKPS cultures treated with the KLF4 inhibitor WX2-43, DAPI in blue as nuclear counterstain. Scale bar, 10 μm. Dots represent individual wells (n = 10). **o**, CFU per well of AKPS organoids treated with WX2-43 (KLF4 inhibitor) with or without rmPlexin B2. Dots represent individual wells (n = 5) a-c, g-i,k,n,o, Barplots indicate mean ± s.d. a,c,h,i,k,n Two-tailed unpaired T-test. b,g,o Ordinary one-way ANOVA with Dunnet's correction for multiple testing. d-f,k, Data pooled from 2 independent experiments, n = 2 mice per condition.

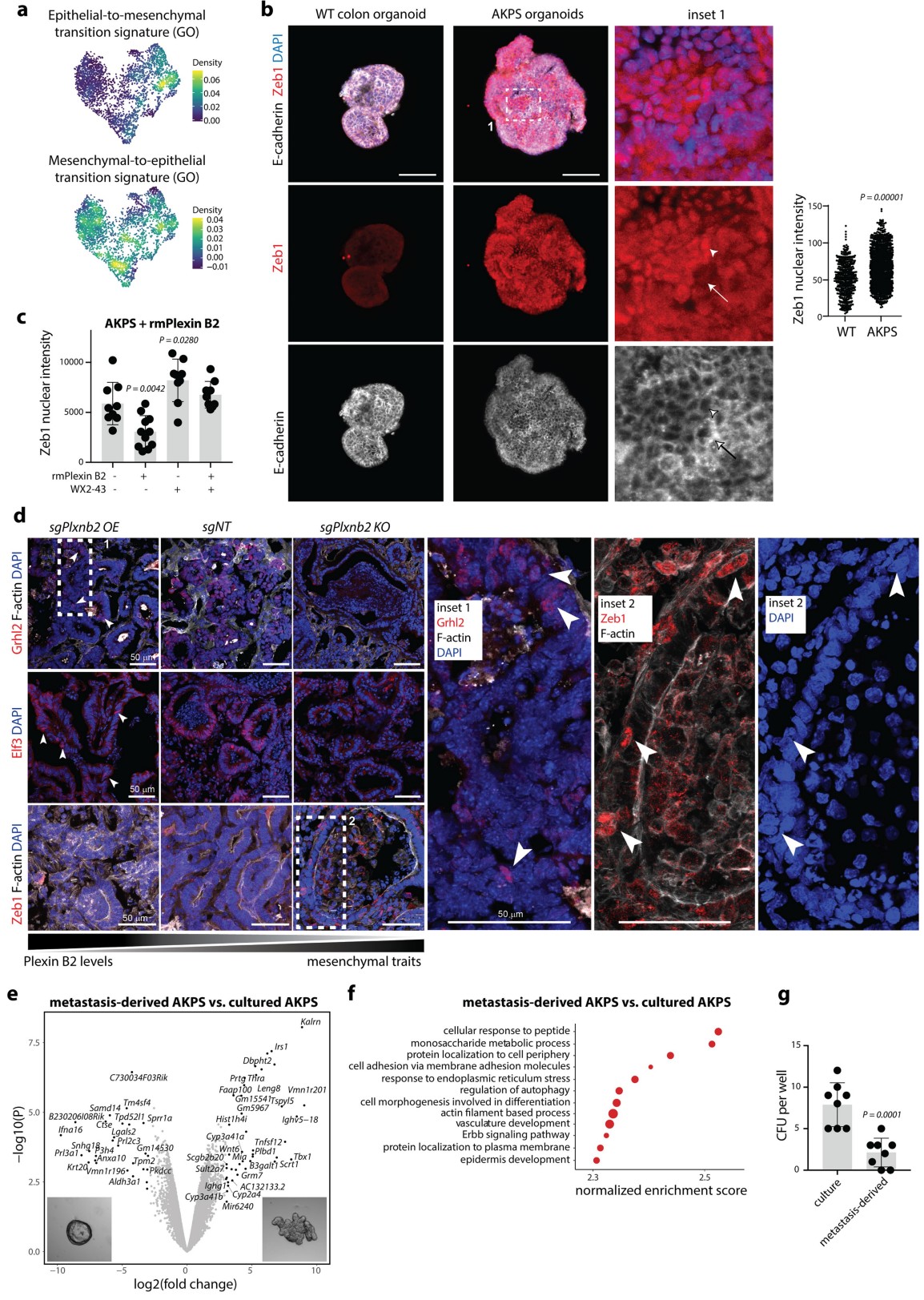

**Extended Data Fig. 8 |** See next page for caption.

**Extended Data Fig. 8 | Plexin B2 promotes acquisition of epithelial traits in AKPS metastases. a**, Expression density of EMT and MET gene signatures in AKPS organoids profiled by scRNASeq. **b**, Representative fluorescent micrographs and quantification of E-cadh (grey) and Zeb1 (red) immunoreactivity in WT colon vs. AKPS organoids. Arrow shows E-cadh$^{high}$Zeb1$^{low}$ cell, arrowhead indicates E-cadh$^{low}$Zeb1$^{high}$. Bars show mean ± s.d. Two-tailed unpaired T-test. **c**, Zeb1 nuclear intensity in 2D AKPS cultures treated with rmPlexin B2 and/or with WX2-43. Barplots indicate mean ± s.d. Barplots indicate mean ± s.d., dots represent nuclear values averaged for individual wells (n = 9). Ordinary one-way ANOVA with Dunnet's correction for multiple testing. **d**, Representative fluorescent micrograph and quantification Grhl2, Elf3 and Zeb1 (red) and F-actin (grey) immunoreactivity of AKPS metastases in *Plxnb2* OE, *Plxnb2* KO and control livers. DAPI in blue as nuclear counterstain. Scale bar, 50 μm. Inset 1 shows Grhl2$^+$ nuclei in *Plxnb2* OE livers. Inset 2 shows Zeb1$^+$ nuclei in *Plxnb2* KO livers. Scale bar, 50 μm. **e**, DEGs between AKPS organoid lines derived from metastases (n = 2) and AKPS-organoid lines kept only in culture (n = 3). Quasi-likelihood F-test. **f**, GO terms enriched in AKPS lines derived from metastases. **g**, CFU per well upon single-cell seeding of AKPS organoids kept in culture (culture) or derived from metastases. Barplots indicate mean ± s.d., dots represent individual wells (n = 8), two-tailed unpaired t-test.

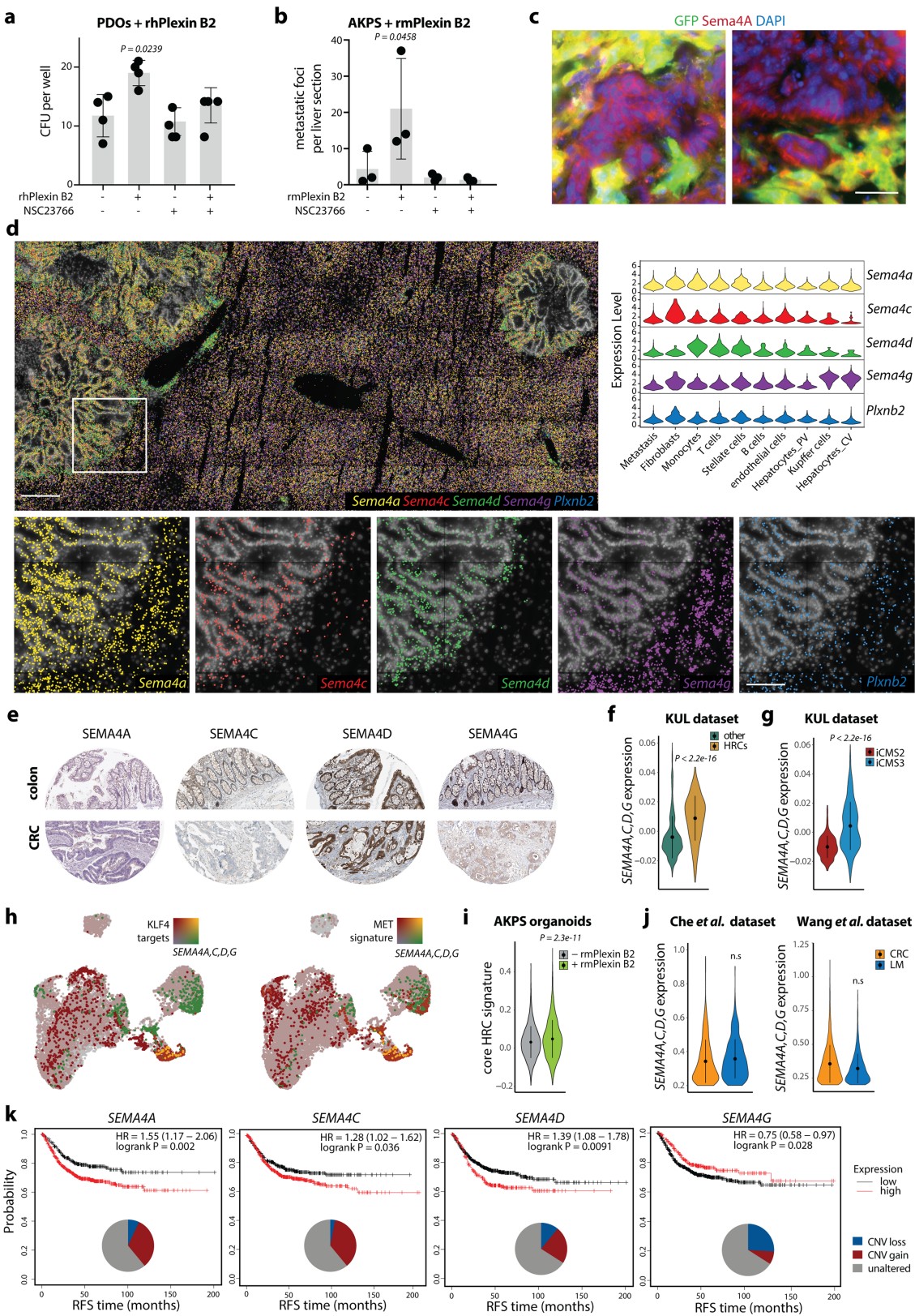

**Extended Data Fig. 9** | See next page for caption.

**Extended Data Fig. 9 | Class IV semaphorin expression in mouse and human CRC. a**, CFU per well of PDOs cultured with rhPlexin B2 and NSC23766. Dots represent individual wells (n = 4). **b**, Metastases per liver section of AKPS organoids treated ex vivo with rmPlexinB2 and/or NSC323766. Dots represent individual mice (n = 3 mice per group). **c**, Representative fluorescent micrograph of Sema4A⁺ tumour cells (red) in contact with GFP⁺ hepatocytes (green). 2 FOV shown. DAPI in blue as nuclear counterstain. **d**, Multiplexed in situ hybridization (Molecular Cartography) and normalized expression level per cell type of *Sema4a, 4c, 4d, 4g* and *Plxnb2* transcripts in AKPS liver metastases. Data pooled from n = 3 mice. **e**, Immunohistochemistry for SEMA4A, C, D, and G in healthy human colon and CRC (n = 4) obtained from the Human Protein Atlas (https://www.proteinatlas.org/). **f,g**, Averaged expression of *SEMA4A, 4C, 4D, 4G* in human primary CRC (epithelial cells only, KUL dataset includes 5 patients[18]). Cells grouped as high-relapse cells (HRCs) vs. other cells, or iCMS2 vs. iCMS3 cells. **h**, Averaged expression of *SEMA4A, SEMA4C, SEMA4D, SEMA4G* (green) projected on epithelial cells of the Samsung dataset and blended with expression of KLF4-target genes or of the MET signature (red). **i**, Expression of the core HRC signature in AKPS organoids treated with or without rmPlexin B2. **j**, Averaged expression of *SEMA4A, SEMA4C, SEMA4D, SEMA4G* in epithelial cells from CRC liver metastases (LM) or matched primary tumours in two independent datasets of metastatic CRC (Wang et al. n = 5[21], Che et al. n = 6[22]). **k**, Kaplan-Meier analysis (http://kmplot.com) of the recurrence free survival (RFS) of CRC patients (n = 1211) stratified by *SEMA4A, SEMA4C, SEMA4D* or *SEMA4G* expression. Log Rank test. Pie charts indicate CNV analysis *SEMA4A, SEMA4C, SEMA4D* and *SEMA4G* in the COAD dataset (n = 290 patients). f,g,i,j, Two-tailed unpaired Wilcoxon test. a,b,f,g,i,j Data are presented as mean ± s.d.

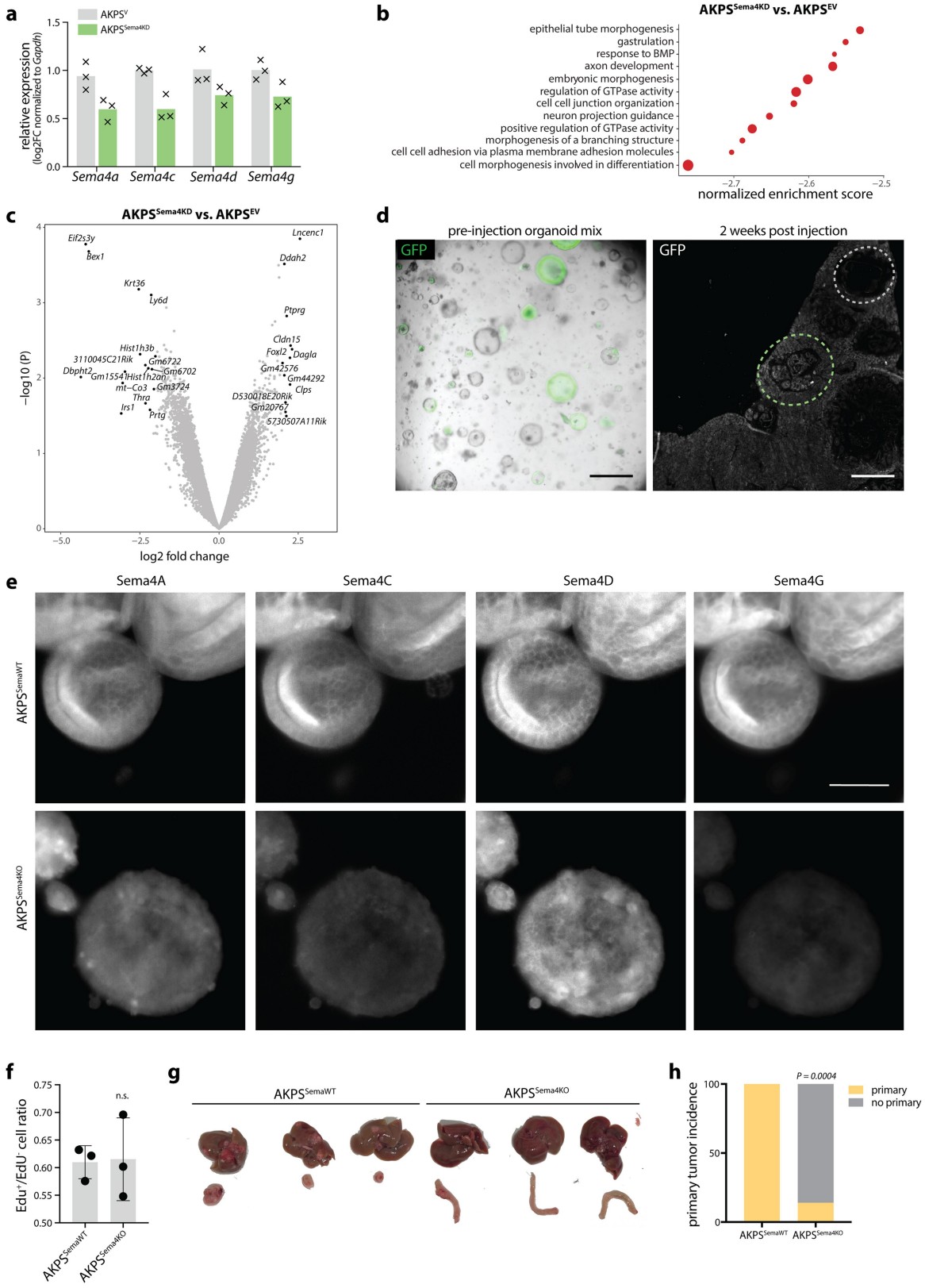

**Extended Data Fig. 10** | See next page for caption.

**Extended Data Fig. 10 | Class IV semaphorins on tumour cells are required for metastatic seeding. a**, Gene expression of *Sema4a, Sema4c, Sema4d* and *Sema4g* in AKPS[Sema4KD] vs. AKPS[EV] organoids. Log2FC normalized to *Gapdh*. Crosses indicate technical replicates (n = 3). **b,c**, GO terms and DEGs in AKPS[Sema4KD] vs. AKPS[EV] organoids. Quasi-likelihood F-test, n = 3 per group. **d**, Left, representative fluorescent micrographs of pre-injection 1:1 mix of AKPS[Sema4KD] organoids (GFP[+]) and control organoids (GFP[-]). Scale bar, 100 μm. Right, representative GFP immunoreactivity of AKPS[Sema4KD] metastases (GFP[+], green dashed line) and AKPS[EV] (GFP[-], grey dashed line). Scale bar, 200 μm. **e**, Representative fluorescent micrograph of Sema4A, C, D and G immunoreactivity in AKPS[Sema4KO] and AKPS[SemaWT] organoids. Scale bar, 20 μm. **f**, Edu[+]/Edu[-] cells in 2D AKPS[Sema4KO] and AKPS[SemaWT] organoids. Dots indicate individual wells (n = 3), barplots indicate mean ± s.d. Two-tailed unpaired t-test. **g**, Representative images of livers (top) and primary colon tumours (bottom) in mice orthotopically injected with AKPS[Sema4KO] or AKPS[SemaWT] organoids. **h**, Incidence of spontaneous liver metastases in mice orthotopically injected with AKPS[Sema4KO] or AKPS[SemaWT] organoids. Two-tailed unpaired t-test, n = 8 mice per group.

# Reporting Summary

## Statistics

For all statistical analyses, confirm that the following items are present in the figure legend, table legend, main text, or Methods section.

| n/a | Confirmed | |
|---|---|---|
| ☐ | ☒ | The exact sample size (*n*) for each experimental group/condition, given as a discrete number and unit of measurement |
| ☐ | ☒ | A statement on whether measurements were taken from distinct samples or whether the same sample was measured repeatedly |
| ☐ | ☒ | The statistical test(s) used AND whether they are one- or two-sided<br>*Only common tests should be described solely by name; describe more complex techniques in the Methods section.* |
| ☐ | ☒ | A description of all covariates tested |
| ☐ | ☒ | A description of any assumptions or corrections, such as tests of normality and adjustment for multiple comparisons |
| ☐ | ☒ | A full description of the statistical parameters including central tendency (e.g. means) or other basic estimates (e.g. regression coefficient) AND variation (e.g. standard deviation) or associated estimates of uncertainty (e.g. confidence intervals) |
| ☐ | ☒ | For null hypothesis testing, the test statistic (e.g. $F$, $t$, $r$) with confidence intervals, effect sizes, degrees of freedom and $P$ value noted<br>*Give P values as exact values whenever suitable.* |
| ☒ | ☐ | For Bayesian analysis, information on the choice of priors and Markov chain Monte Carlo settings |
| ☒ | ☐ | For hierarchical and complex designs, identification of the appropriate level for tests and full reporting of outcomes |
| ☐ | ☒ | Estimates of effect sizes (e.g. Cohen's *d*, Pearson's *r*), indicating how they were calculated |

*Our web collection on statistics for biologists contains articles on many of the points above.*

## Software and code

Policy information about availability of computer code

| Data collection | The code used in this study is available at: https://github.com/Moors-Code/coco_mosaic_liver |
|---|---|
| Data analysis | GraphPad Prism v8.2.0 GraphPad Software Schneider https://www.graphpad.com/scientific- software/prism/<br>R software 4.1.0 GNU project https://www.r-project.org<br>R Studio Server 1.4.1717, RStudio https://www.rstudio.com<br>Bcl2fastq v2.20.0.422 (Illumina https://support.illumina.com/sequencing/sequencing_software/bcl2fastq-conversion-software.html)<br>Space Ranger (v1.1.0 or v1.2.0, 10x Genomics)<br>edgeR R package Robinson et al, 2010 https://bioconductor.org/packages/release/bioc/html/edgeR.html<br>msigdbr R package R Bioconductor https://cran.r-project.org/web/packages/msigdbr/vignettes/msigdbr-intro.html<br>fgsea R package  Sergushichev et al, 2016 https://bioconductor.org/packages/release/bioc/html/fgsea.html<br>ggplot2 R package Wickham, 2016 https://cloud.r-project.org/web/packages/ggplot2/index.html<br>NicheNet v1.0.0 R package, Browaeys et al, 2019<br>CellPhoneDB v2.0.0 R package, Efremova et al, 2020<br>Seurat v4, R package, Hao, Hao et al, 2021<br>Signac v1.12, R package, Stuart et al, 2021<br>DecontX v1.0.0, R package, Yang et al, 2020<br>Biomart R package v2.46.3<br>Image J Fiji Schindelin et al, 2012 https://imagej.net/Fiji/<br>FlowJo v10.7.1 (Becton Dickinson & Company)<br>cutadapt, Martin et al 2011<br>FISHquant v2, python package, Imbert et al, 2022 |

HORIZON v2.1.0.0, Lunaphore technologies
Bowtie2, Langmead et al, 2012
MAgECK, Li et al, 2014

For manuscripts utilizing custom algorithms or software that are central to the research but not yet described in published literature, software must be made available to editors and reviewers. We strongly encourage code deposition in a community repository (e.g. GitHub). See the Nature Portfolio guidelines for submitting code & software for further information.

## Data

Policy information about availability of data

All manuscripts must include a data availability statement. This statement should provide the following information, where applicable:
- Accession codes, unique identifiers, or web links for publicly available datasets
- A description of any restrictions on data availability
- For clinical datasets or third party data, please ensure that the statement adheres to our policy

The sequencing data generated in this study are available at the Gene Expression Omnibus under the accession numbers GSE267981 and GSE267982 and the Zenodo repository: 10.5281/zenodo.7737590. The code used in this study is available at https://github.com/Moors-Code/coco_mosaic_liver.
Kaplan-Meier analysis of 1211 CRC patients was performed using the online tool http://kmplot.com. Recurrence free survival (RFS) was stratified by SEMA4A, SEMA4C, SEMA4D and SEMA4G expression in the Affymetrix colon dataset, using best cutoff.
Protein atlas stainings: SEMA4A, D, C, and G antibody stainings were obtained from the Human Protein Atlas with the R package HPAanalyze.
CNV analysis of class IV semaphorin genes: the CNV status of SEMA4A, D, C, and G in 290 CRC patients was obtained from the TGCA-COAD dataset88.
Published scRNASeq datasets: Pre-processed and annotated scRNASeq profiles of epithelial cells in the KUL and Samsung dataset were obtained from the Synapse repository syn34942428. scRNASeq datasets of matched liver metastases and primary tumors were obtained from the Gene Expression Omnibus under the accession numbers GSE22585786 and GSE17831887, and imported in Seurat. Epithelial cells were subsetted based on EPCAM expression. Expression of was computed with the AddModuleScore function.

## Research involving human participants, their data, or biological material

Policy information about studies with human participants or human data. See also policy information about sex, gender (identity/presentation), and sexual orientation and race, ethnicity and racism.

| Reporting on sex and gender | na |
|---|---|
| Reporting on race, ethnicity, or other socially relevant groupings | na |
| Population characteristics | The sample of human CRC hepatic metastasis (#CB522586) originated from a 44 years old male. |
| Recruitment | na |
| Ethics oversight | CRC-patient derived organoids from primary and liver metastases were obtained from the University Hospital Basel following patient consent and ethical approval (Ethics Committee of Basel, EKBB, number EKBB numbers 2019-00816). |

Note that full information on the approval of the study protocol must also be provided in the manuscript.

# Field-specific reporting

Please select the one below that is the best fit for your research. If you are not sure, read the appropriate sections before making your selection.

☒ Life sciences   ☐ Behavioural & social sciences   ☐ Ecological, evolutionary & environmental sciences

For a reference copy of the document with all sections, see nature.com/documents/nr-reporting-summary-flat.pdf

# Life sciences study design

All studies must disclose on these points even when the disclosure is negative.

| Sample size | In accordance with the 3Rs, the smallest sample size was chosen that could give a significant difference. Given the robustness of the phenotypes across all methods used, the minimum sample size assuming no overlap in control versus experimental is three animals per experiment. |
|---|---|
| Data exclusions | No animals were excluded, unless data acquisition quality was insufficient. For experiments with adeno-associated virus, animals with GFP coverage < 60% were excluded. Thresholds applied in bioinformatic analyses (standard parameters) are available in the github repository of the code. |
| Replication | Data was combined from independent biological replicates (mice) and analyzed together. Only consistent observations were reported. When indicated in the figure legends, data from multiple experiments was pooled for quantification. |

| Randomization | Treatment and control animals were randomized across cages, sex and age. For other experiments, randomization was not required. |
| Blinding | The researcher was blinded to the genotype or treatment group during the processing and analysis. |

# Reporting for specific materials, systems and methods

We require information from authors about some types of materials, experimental systems and methods used in many studies. Here, indicate whether each material, system or method listed is relevant to your study. If you are not sure if a list item applies to your research, read the appropriate section before selecting a response.

## Materials & experimental systems

| n/a | Involved in the study |
|---|---|
| ☐ | ☒ Antibodies |
| ☐ | ☒ Eukaryotic cell lines |
| ☒ | ☐ Palaeontology and archaeology |
| ☐ | ☒ Animals and other organisms |
| ☒ | ☐ Clinical data |
| ☒ | ☐ Dual use research of concern |
| ☒ | ☐ Plants |

## Methods

| n/a | Involved in the study |
|---|---|
| ☒ | ☐ ChIP-seq |
| ☐ | ☒ Flow cytometry |
| ☒ | ☐ MRI-based neuroimaging |

## Antibodies

| Antibodies used | For sorting, hepatocytes were resuspended in 2 mL FACS buffer (2 mM EDTA, 0.5% BSA in PBS) and stained with Zombie Violet (1:500) (Biolegend #423113), TruStain FcX™ (anti-mouse CD16/32) antibody (BioLegend #101320, 1:50), PE/Cy7 anti-mouse CD31 (BioLegend #102418, 1:300), BV570 anti-mouse CD45 (BioLegend #103135, 1:300) for 25 min at 4 °C.<br><br>Immunofluorescence on formalin fixed paraffin embedded tissue: Sections were incubated overnight at 4°C with the following primary antibodies (1:100, in blocking buffer): anti-CD146 (Abcam #ab75769); anti-α-SMA (Abcam #ab5694) anti-periostin (Abcam #ab227049) and anti-GFP (Aves Labs #GFP-1020). Sections were repeatedly washed in PBST and incubated with the following secondary antibodies (1:400, in blocking buffer) for 1h at RT: AlexaFluor goat anti-rabbit 594 (#A-11012), AlexaFluor goat anti-rabbit 647 (#A-21244), AlexaFluor goat anti-chicken 647(#A32933), all from ThermoFisher.<br><br>Multiplexed immunofluorescence was performed on the Comet instrument (Lunaphore) with the following antibodies (1:100): anti-cleaved Caspase 3 (Cell Signaling #9661), anti-CD68 (Abcam #ab125212), anti-CD4 (Abcam #ab183685), anti-Ki67 (Abcam #ab15580), anti-E-cadherin (Cell Signaling, #3195), anti-α-SMA (1:1000, SIGMA #A2547), anti-CD146 (Abcam, #ab75769).<br><br>Immunofluorescence on fixed frozen tissue: Sections were stained as above with the following primary antibodies: anti-glutamine synthetase (1:100, Biolegend #856201), anti c-Myc (1:100, 9E10, Thermo Fisher Scientific), anti-Plexin B2-PE (1:500, Biolegend #145903), anti-GFP-AlexaFluor488 (1:200, ThermoFisher #A-21311), anti-Zeb1 (1:400, Novus #NBP1-05987), anti-α-SMA (1:1000, SIGMA #A2547), anti-E-cadherin (1:100, Biotechne, AF748), anti-Epcam (1:100, Abcam #2884975), anti-GRHL2 (1:100, Abcam #ab271023), anti-Klf4 (1:100, Biotechne #AF3158), anti-ELF3 (1:100, Thermo Fisher Sceintific #PA5-120996) and anti-Sema4A (1:100, Biolegend #148402). DAPI counterstain, mounting and imaging was performed as above. F-actin was stained by incubating blocked slides for 2 hours at RT with Alexa Fluor 647 Phalloidin (1:400, Invitrogen #A22287). Sections were repeatedly washed in PBST and incubated with the following secondary antibodies (1:400, in blocking buffer) for 1h at RT: AlexaFluor goat anti-rabbit 594 (#A-11012), AlexaFluor goat anti-rabbit 647 (#A-21244), AlexaFluor goat anti-chicken 647(#A32933), all from ThermoFisher.<br><br>Immunofluorescence in organoids: anti-Zeb1 (1:400, Novus #NBP1-05987), anti-Ecadherin (1:200, BD Biosciences #610181), mouse anti-Sema4A (Biolegend #148402), rat anti-Sema4C-AF647 (Biotechne #FAB8497R), rat anti-Sema4D-PE (Biolegend #147603), rabbit anti-Sema4G (Thermo Fisher Scientific #BS-11479R).<br><br>Immunofluorescence in cell lines: rabbit anti-VSV-G antibody (Thermo Fisher Scientific #PA1-29903), mouse anti-PlexinB2-PE (Biolegend #145903). |
| Validation | All antibodies have been previously validated by the manufacturer for use in immunofluorescence. See manufacturer's website for further information. All antibodies were additionally validated in the study by including an unstained control (no primary antibody, only secondary). |

## Eukaryotic cell lines

Policy information about cell lines and Sex and Gender in Research

| Cell line source(s) | HEK293T cells - ATCC<br>AML12 - CRL-2254 - ATCC<br>PTA-5565 H2B-mCherry (Mohamed Bentires-Alj, University of Basel) |

| | KPC cell line (Ilaria Guccini, ETH Zurich)<br>D4M-3A (Merck Millipore) |
|---|---|
| Authentication | None of the cell lines were autheticated |
| Mycoplasma contamination | Cells tested negative for micoplasma contamination. |
| Commonly misidentified lines<br>(See ICLAC register) | No commonly misidentified cell lines were used in this study. |

# Animals and other research organisms

Policy information about studies involving animals; ARRIVE guidelines recommended for reporting animal research, and Sex and Gender in Research

| Laboratory animals | Albumin-Cre mice (AlbCre, stock no. 003574), LSL-dCas9-SPH (dCas9-SPH, stock no. 031645), LSL-Cas9 (Cas9, stock no. 024858), mT/mG mice (stock no. 007576) and Plxnb2flox/flox mice (stock no. 036883) were obtained from a local live mouse repository. Chow and water were available ad libitum, unless specified. All mice were in the B6J background and maintained on a 12h light / 12h darkness schedule. Mice were housed and bred under specific pathogen-free conditions in accredited animal facilities. All experiments were performed on 6-16 week-old male and female mice. |
|---|---|
| Wild animals | No wild animals were used. |
| Reporting on sex | All experiments were performed on 6-16 week-old male and female mice. |
| Field-collected samples | No field-collected samples. |
| Ethics oversight | All experimental procedures were performed in accordance with Swiss Federal regulations and approved by the Cantonal Veterinary Office. |

Note that full information on the approval of the study protocol must also be provided in the manuscript.

# Flow Cytometry

## Plots

Confirm that:

☒ The axis labels state the marker and fluorochrome used (e.g. CD4-FITC).

☒ The axis scales are clearly visible. Include numbers along axes only for bottom left plot of group (a 'group' is an analysis of identical markers).

☒ All plots are contour plots with outliers or pseudocolor plots.

☒ A numerical value for number of cells or percentage (with statistics) is provided.

## Methodology

| Sample preparation | mice were sacrificed by raising CO2 concentrations, then the abdomen was opened and a G22 cannula was inserted into the inferior vena cava. The liver was perfused with 20 mL Hanks buffer (0.5 mM EDTA and 25 mM HEPES in HBSS) followed by 15 mL digestion buffer (15 mM HEPES and 32 µg/mL Liberase in low glucose DMEM). After initial swelling of the liver, the portal vein was cut to allow outflow. After perfusion, the gallbladder was removed and the liver was transferred to a petri dish with 10 mL digestion buffer and squished with a cell scraper to release the hepatocytes. Liberase was inactivated by adding 10 mL isolation buffer (10% fetal bovine serum (FBS) in low glucose DMEM). The cell suspension was filtered through a 100 µm cell strainer and centrifuged at 50 g for 2 min. The supernatant was removed and the pellet was washed again twice with 20 mL isolation buffer. Hepatocytes were resuspended in 2 mL FACS buffer (2 mM EDTA, 0.5% BSA in PBS) and stained with Zombie Violet (1:500) (Biolegend #423113), TruStain FcX™ (anti-mouse CD16/32) antibody (BioLegend #101320, 1:50), PE/Cy7 anti-mouse CD31 (BioLegend #102418, 1:300), BV570 anti-mouse CD45 (BioLegend #103135, 1:300) for 25 min at 4 °C. Hepatocytes were washed and filtered through a 70 µm strainer. CD45-CD31- hepatocytes which contained a sgRNA (GFP+) were divided into metastasis-proximal (mCherry+) and metastasis-distal (mCherry-) by drawing different sorting gates on an AriaIII sorter (BD Biosciences) with 70-micron nozzle. Cells were collected in PBS, spun down at 800 g for 5 min and the pellet was stored at -20 °C. |
|---|---|
| Instrument | AriaIII sorter (BD Biosciences) with 70-micron nozzle. |
| Software | Aquired data were analyzed using FlowJo software. |
| Cell population abundance | Post-sort populations were not analyzed. Genomic DNA was extracted from all sorted cells. |
| Gating strategy | Events were initially gated by FSC-A and SSC-A, then by FSC-A and FSC-H (to exclude doublets). Hepatocytes were gated as CD45-CD31-. |

☒ Tick this box to confirm that a figure exemplifying the gating strategy is provided in the Supplementary Information.

