## [Peer Review File · Nature]

Manuscript Title: In vivo interaction screening reveals liver-derived constraints to metastasis

Reviewer Comments & Author Rebuttals

Reviewer Reports on the Initial Version:

Referee #1 (Remarks to the Author):

In this study, Borrelli and colleagues develop an in vivo CRISPR-A-based screen that combines transposon liver transduction of guide RNAs with proximity labeling of hepatocytes to identify host factors involved in liver metastasis formation. Using this methodology, they identify PlexinB2 as a liver-expressed molecule that facilitates metastasis formation by colorectal cancer (CRC) cells. Mechanistic studies suggest that PlexinB2 in hepatocytes interacts with class 4 Semaphorins expressed in metastatic cells and the activity of this pathway induces EMT reversion and survival of disseminated tumor cells during liver colonization. This is an interesting and relevant study. I found the screening method elegant and outstanding from a technical perspective. The data indicating that liver PlexinB2 expression levels modulate metastasis burden in models of intrasplenic tumor cell inoculation is also robust. In contrast, results supporting the proposed mechanistic action, i.e., reversion of EMT and survival, is very preliminary and not convincing. I am skeptical about this part. I detail my criticisms below:

1) Recent studies indicate liver metastasis formation upon implantation of CRC organoids in the primary site is due to dissemination of Lgr5 negative cell populations, which in the primary cancer undergo invasion and extravasation (Cañellas-Socias et al. Nature 2022; Fumagalli et al. Cell Stem Cell. 2020). In contrast, more simplistic metastasis models based on portal vein injection of CRC cells (as authors in this study) showed a dependency on Lgr5+ cancer stem cells (de Sousa E Melo. Nature 2017). These observations reveal substantial differences due to technical biases depending on the site of inoculation. It is, therefore crucial that authors show that PlexinB2-Semaphorin interactions are also required for the formation of liver metastases from the primary CRCs.

2) The data supporting that EMT reversion is the mechanism that explains the requirement for PlexinB2 in hepatocytes is far from convincing. This part should be largely strengthened.

-First, evidence that EMT occurs or is required for colonization in the model system used throughout the study is weak. Authors suggest that some cells in organoids remain in an EMT-like state, but this is largely speculative. In addition, they inoculate single-cell suspensions via the spleen, which should be able to directly enter the liver parenchyma through the sinusoids without the need for invasion or migration.

-Second, the expression of Zeb1 is cytoplasmic (fig. 4d) and the staining seems not specific. None of the EMT hallmarks is modulated upon PlexinB2 knockdown or overexpression. The geneset enrichment related to epithelial morphogenesis does not directly support EMT reversion. Evidence is

solely based on FoxP2 and Id3 regulation, but the relevance of these two genes in regulating the colonic epithelial phenotype is unclear.

To address these questions, authors should characterize by single cell profiling organoids and metastases with gain and loss of PlexinB2-Semaphorin signaling and provide additional evidence for alteration in the EMT state. Experiments of inoculation at the primary site (point 1) may also help clarify this issue.

3) The proposed model is difficult to reconcile with the data shown. PlexinB2 is a membrane-bound protein expressed in hepatocytes. Authors propose that PlexinB2-Semaphorin interactions between disseminated cancer cells and hepatocytes at the time of metastatic colonization are necessary for MET. But why the acquisition of epithelial features is sustained in large metastases of Plxnb2 deficient livers where hepatocytes no longer interact with tumor cells because of TME recruitment? Even if Semaphorin are shed and act as soluble ligands, how would cancer cells be instructed by hepatocytes that are not interacting directly with cancer cells, and through which mechanism MET is enforced in these metastases? can authors rule out that decreased desmoplastic reactions that characterize metastasis growing in livers overexpressing PlexinB2 indirectly affect the organization of epithelial cells, including changes in Epcam levels?

4) The manuscript will also gain relevance if the authors could provide insights into the mechanism by which Semaphorin signaling triggers EMT reversion in AKPS organoids.

5) The data on cancer cell death versus proliferation triggered by PlexinB2-Semaphorin signaling on DTCs is neither clear. How is it connected to the MET reversion process? According to the authors, organoids are in an EMT state yet proliferate and survive. To address these issues, authors must investigate the kinetics of metastasis (using bioluminescence and/or histology) and assess the numbers of proliferative and apoptotic cells during liver colonization.

6) A more detailed analysis of PlexinB2 expression in the liver and interactions with Semaphorin+ tumor cells during organ colonization and outgrowth is also needed it. Are PlexinB2-expressing hepatocytes localized in proximity to the portal vein and liver sinusoids? Do they contact Semaphorin+ tumor cells?

6) Many assumptions are based on the interpretation of individual stainings, but no quantifications are provided. This is particularly relevant in Figure 4d and extended data in fig 5.

7) Authors show that combined shRNA-mediated knockdown of four semaphorin-4 members reduces metastatic capacities of AKPS organoids. However:

- Downregulation of each individual semaphorin at RNA levels only ranges between 20%-50%, which makes me suspicious that the lower metastatic capacity of the KD organoids could be due to off-target effects.
- Proper controls (shRNA targeting of 4 neutral sequences and additional shRNA targeting semaphorins) are not included.

Authors should provide appropriate controls for this experiment and characterize changes in MET/epithelial organization of liver metastasis upon Semaphorin downregulation. Including additional experiments of semaphoring knock-down or knock-out by other genetic means or activity inhibition in vivo (e.g., using soluble PlexinB2 recombinant protein inoculation) would make this part more convincing.

8) The relevance of the author's findings for human CRC is only supported by a few Kaplan-Meier plots based on bulk transcriptomic data. Given the wealth of scRNAseq datasets of CRC samples, it would be interesting to assess whether Semaphorins are expressed in particular tumor cell states that have been associated with metastasis formation (Lgr5 negative cells, HCRs, etc). Plexin B2 and Semaphorin expression analyses by ISH in human samples would also strengthen the study.

Referee #3 (Remarks to the Author):

The study by Costanza Borrelli and colleagues seeks to explore the host determinants that modulate the seeding and outgrowth of metastatic cells in the liver. To address this, the authors have developed a novel and elegant in vivo approach to systematically interrogate hepatocyte intrinsic factors that promote the establishment of metastases. The screening approach used is based on a CRISPR-activation system that achieves specific gene activation in hepatocytes. The focused approach undertaken is reasonable and targets a wide range of factors (ligands, receptors, and relevant pathways) for their role in metastasis. Authors posit that metastatic outgrowth is contingent on interactions between hepatocytes and disseminating tumor cells, which they infer using a clever in vivo niche-labeling system that enables to distinguish between (niche) perturbations that are proximal and distal to metastases.

To prioritize candidates for validation, authors perform spatial transcriptomics of liver CRC mets and narrow down the list to 22 factors involved in tumor-hepatocyte interaction, and further filter the list according to mutated factors in liver mets. They then pursue orthogonal in vitro validation of sgRNAs of top-ranked hits on hepatocytes co-cultured with dissociated CRC organoids. Overexpression of PlexinB2 by sg-activation or treatment with recombinant protein, consistently promotes metastatic proliferation of organoids through interaction with class 4 semaphorins. This is validated in vivo with the AKPS model and KPC cells.

Collectively, the authors have developed an interesting approach that may be leveraged to investigate the biological relevance of host factors in metastatic colonization. This reviewer congratulates the authors on pioneering such a methodological advance which will certainly enable further studies. However, the mechanistic insights related to the biology of metastatic seeding, human relevance, and generalizability of their findings are underdeveloped. I hope the following comments will help authors improve their manuscript.

Major comments:

1. Given the novelty and broad-use potential of the in vivo screening system, additional details should be included. Specifically, authors should validate the “inferred proximity” measurement (as now inferred from flow cytometric analysis of dissociated livers) by providing histological/imaging data that directly shows a differential spatial distribution (proximal vs distal from grafted metastasis) of hepatocytes transduced with seeding-promoting vs seeding-suppressing gRNAs (top hits). To do so, authors could sub-clone gRNAs targeting “seeding-promoting” host factors into a modified version of the initial transposon vector in which the GFP reporter is substituted by a different fluorophore (eg BFP). The prediction would be that BFP+ hepatocytes (expressing “seeding promoting” hits: *Plxnb2*, *Nrp2*, *Ncam1*, *Nenf*, *Egf*, *Psen1*, *Gpc3*) would be more proximal to metastases than GFP+ hepatocytes (expressing “seeding-suppressive” hits: *Saa*, *Ltb*, *Serpina1a*, *App*, *Fn1*, *Cp*) in *AlbCre*;*dCas9*-SPH but not *NonCre* mice. The % of BFP/GFP double-positive hepatocytes will also be informative to support a low MOI in the pooled screening system.

2. Authors should assess the generalizability of their findings in additional lines and primary tumors of origin, and validate the role of Plexin B2 as a gatekeeper of liver metastasis in models beyond intrasplenic injection (eg. GEMMs or at least orthotopic transplantation-based) that better recapitulate the natural course of progression from an existing primary tumor.
3. The mechanism of Plxnb2 upregulation is unclear and should be explored further. This is important to support the physiological relevance of their approach. Specifically, authors should show the basal expression of Plxnb2 in unperturbed hepatocytes and test whether Plxnb2 is upregulated in response to tumor-secreted factors (prior and after seeding?), by direct tumor-hepatocyte interactions, and/or via other host/liver-related traits linked with metastasis burden and/or susceptibility, to clarify whether this is a tumor-induced or host-induced mechanism.
4. The proposed effects by which Plxnb2 OE induces mesenchymal-to-epithelial transition (and its blockade has an inhibitory effect) relies on illustrative panels of established markers that should include more extensive immunolabeling and quantification, and be accompanied by OE/KD and epistasis experiments to support the claims that Plxnb2 modulates metastasis via regulation of EMT/MET states, and that this is mediated by Rac1 signaling and actin cytoskeleton remodeling. It would also be relevant to assess whether Plxnb2-Sema axis is required to maintain the growth of established metastases after epithelial traits have been acquired. Expanding the mechanistic depth would enhance the significance of the study.
5. The human relevance of the findings should be strengthened by providing additional insights on the regulation of Plxnb2 levels and/or signaling in patients with differential liver metastasis burden, and/or assessing whether Plxnb2 levels associate with patient prognosis or metastasis relapse in human datasets – to complement the data in Ext Data Fig 6. Authors could exploit available single cell analyses on human liver metastasis samples to confirm the validity of their data in patients, and in relation to established early markers of metastasis (eg Emp1, etc).
6. According to the literature, Plexin B2 is strongly expressed in the endothelial cells of the hepatic sinusoids and portal veins (Zielonka et al. Exp Cell Res 2010). Is this cell type involved in the host-derived MET-liver seeding mechanism proposed in this work, or is PlexinB2 activation in hepatocytes specifically a necessary step for liver metastasis in the cancer types investigated?
7. Several conclusions are based on visual representations of data/images throughout the paper. A more thorough quantification of staining and phenotypes should be performed throughout to allow for a more systematic evaluation of the claims – eg: Fig 3 h-i, 4c-d, several panels on Ext Fig 4, Ext Data 5, Ext Data 6 etc. Similarly, authors should perform more detailed statistical analyses – ie: Fig 2A, Ext Fig Data 5, etc
8. The discussion would benefit from a more elaborated conceptual discussion of their findings in relation to the relative contribution of the “seed” versus the “soil” in metastasis formation.

Minor comments:

A clearer description of experimental design details of the screening technology would be appropriate so this information is more accessible to readers: size (micro, macro), number of nodules per mice at the timepoint(s) analyzed; number of replicates, controls, etc,

Spatial transcriptomics: it is not clear the number of different hepatic CRC metastases and patient samples the data was generated from. Also, regarding the NicheNet predictions: what is the number of predicted interactions of tumor cells at the metastatic leading edge with metastasis-distal hepatocytes, as a reference? Without a reference or control comparison, it is hard to interpret whether the numbers and overlaps described are higher than what one would find randomly.

Figure calling should be done in chronological order to facilitate the reading – eg: Fig 3g is called prior to Fig 3s-b etc.

Extended Data Fig. 2: same x-axis for experimental and control panels.

While the narrative is easy to follow, there are minor typos in the text that should be corrected.

Referee #4 (Remarks to the Author):

The authors perform an innovative CRISPR activation screen with a focused sgRNA library in hepatocytes *in vivo*, to identify factors that influence metastasis of cancer cells into the liver. They find known and unknown factors influencing, i.e., enhance or decrease, metastases and they decide to focus on Plexin B2. Analysis of human liver tissues with metastases validates Plexin B2 as a potential target as well. The authors perform *in vitro* experiments to demonstrate that metastasised cancer cells are affected via a Plexin B2 stimulation derived from the hepatocytes. Deletion of Plexin B2 in the hepatocytes in mice almost completely abolishes seeding of cancer cells into the liver. Rac1 signalling is proposed to be induced in the cancer cells, via Plexin B2 signalling involving Semaphorins. Plexin B2 signalling seems to be responsible for a mesenchymal to epithelial phenotype observed in the liver metastases.

The idea of using a secreting fluorophore (sLP-mCherry) that is secreted by the cancer cells (colorectal and pancreatic organoids) and combining it with a CRISPR activation approach *in vivo* in the seeded liver is a highly innovative, unique assay identifying factors that enhance or reduce seeding of metastases in the liver. The approach is very valid and innovative, and the quality of the data presented very high. However, in its current form the paper is not acceptable for publication as more experimental data is required to fully support the conclusions drawn by the authors. The authors use the LSL-dCas9-SPH mice crossed to the Albumin-Cre transgenic animals to tissue specifically induce the CRISPR activation machinery – based on the Suntag technology – in the hepatocytes. The LSL-dCas9-SHP mice have been developed and described for applications in the nervous system (Nature Neuroscience 2018 (PMID: 29335603)). However, it was recently reported on the Jackson webpage that these mice had multiple phenotypic issues (spontaneous urination, etc.) when they were received from the authors of that study. The Jackson lab bred these mice onto a mixed B6 x DBA background and that resulted in a normal phenotype. However, further issues with this model were detected as the original insertion site of the dCas9-SPH described in the original publication could not be verified, but a totally different insertion site had been detected. While I am sure that the authors of the current study are aware of these issues, it is mentioned in the Methods part that the mice were kept on a C57BL6/J background. Could the authors please indicate how many generations their strain was backcrossed and which model in terms of insertion sites the authors received from the Jackson lab? This could lead to some skewing of the experiments as for example the expression of the CRISPR activation machinery could be mosaic, suggesting that only a few liver cells do indeed express sufficient dCas9-SHP resulting in mosaic transcriptional activation of target genes despite the presence of a sgRNA. Conversely, the induction of expression of a certain gene could become too high when combined with a specific sgRNA due to the transgene being in a certain genomic region, which leads to very high expression of the CRISPRa components. In addition, it is not clear, whether the combination of the dCas9-SHP and sgRNA could induce some off-targets. This might be a problem as some sgRNAs could cause unwanted side effects on top of their normal role. In summary the authors need to do more work to characterise the dCas9-SPH model described in their paper.

They should perform RNAseq analysis of Alb-Cre/dCas9-SPH injected with the Adeno NT sgRNA and PlexinB2 sgRNA to identify specific activation of Plexin B2 and to show which other genes have been induced either in the NT ctrl or the PlexinB2 sgRNA used in the study.

I was also wondering whether the authors observe any immune related anti-transplant issues towards the organoids expressing a secreted Cherry protein. Could the authors please transplant the sLP-mCherry transduced organoids in Rag1 vs wt mice and test the seeding in the liver. Will this have any effects?

For the screen the authors decide to use 3 sgRNAs per gene. As described by Heidersbach AJ, Nature Comms 2023, it is very clear that not all sgRNAs will work and that it often requires many more optimally positioned sgRNAs to obtain robust expression of the gene of interest. Can the authors provide more information on the individual sgRNA potency? From Figure 1E (Figure 1G is mentioned in the main text but the Figure 1g doesn't exist) it looks like that Plexin B2 sgRNAs were strongly enriched. However, it is not clear which of the sgRNAs enrich the most. Is it only one sequence or is it distributed across the 3 independent sgRNAs targeting Plexin B2 found across 7 mice. Can the authors provide qPCR data or immune staining as performed in the Extended data to show how each individual sgRNA targeting Plexin B2 performs in the hepatocytes?

In the in vitro experiments I am not quite clear on the experimental setup. In the methods section the authors say: "On the next day, primary or immortalized hepatocytes were transfected with SB100X and a pool of 3 transposon vectors harboring sgRNAs against selected gene targets using Lipofectamine 3000 (Thermo Fisher Scientific). Three wells were independently transfected for each target in an arrayed fashion, and three wells were left untransfected." Does that mean that per well 3 sgRNAs for one gene or 3 sgRNAs targeting 3 genes were used?

In the same experiment colony formation of the organoids was investigated on the AML12 cell line. Obviously dCas9-SHP was lentivirally introduced into this line, but what other than that is different to the primary hepatocyte assays? To exclude any issues of the sgRNA inducing PlexinB2 the authors should use the AML12 cell line and introduce a Plexin B2 expression construct. This should show similar effects on colony formation than the CRISPRa expression.

Since the experimental errors in the in vitro assays are quite large, and the differences just statistically significant, I would like to ask the authors to perform more repeats of the in vitro experiments and give more background on the analysis, i.e., was it done automated or did someone count the colonies. If the latter than it is important to mention how this person was related to the experiments. Did one person do the experiment and count the colonies or was there someone blinded to the experimental conditions and counted the colonies?

The authors use human mutational analysis to correlate their findings in human liver metastases (Figure 2). Targets include metastasis promoting (e.g. Plexin B2) and non-promoting targets (e.g. App). However, very little information is given on the nature of the mutations found on liver metastatic cancers. Could the authors please elaborate a little bit more on this? Where these gene abolishing or enhancing mutations?

The authors describe an increase of PLXNB2 in the liver leads to increased seeding. However, all hepatocytes seemed to express PLXNB2 at a lower level. Is it just the increased expression of PLXNB2 or something else leading to increased seeding. The in vitro experiments with the recombinant PLXNB2 indicate an effect on proliferation of the tumour cells and the authors make the argument in the paper that this is not the case in vivo as the colony size is similar to cells that seeded in the wt

liver. Does that mean that the in vitro experiments do not really reflect the in vivo scenario? Can you formerly rule out a proliferative advantage in vivo? Please show with KI67 or BrDU stains of the tumour cells seeded in the liver that they are indeed non proliferative. Could it be that in wt livers the metastasising cell are just not detectable, because of lower proliferative rates? Have you tested in the NT sgRNA ctrl, whether the hepatocytes surrounding the metastases express any of your targets you identified in your screen, e.g. PLXNB2 or others?

In the manuscript it is suggested that survival of the cancer cells is enhanced through the PlexinB2 signalling. However, no experimental data on this is provided. Could the authors please experimentally address this point further? They could stimulate AKAP organoids with recombinant PlexinB2 in vitro and then measure the expression of pro-survival as well as pro-apoptotic BCL-2 proteins by Western Blot analysis. In addition, they should stain cultures of vehicle vs recombinant Plexin B2 treated AKAP organoids for apoptotic markers, such as AnnexinV/ PI. The authors make the link to Rac1 signalling and show some Actin filament changes (most of the immunofluorescence images are not quantified and the authors should quantify the images in Figure 3 and Extended Figure 4 and 5) and inhibitor studies. However, the Rac1 link should be formally shown by performing a Rac1 activity assay in vitro of organoids stimulated with recombinant Plexin B2.

In the final experiments the authors identify the Semaphorin family as possible interactors of PlexinB2. Since there are multiple members (4) the authors use a shRNA approach to diminish expression of all 4 family members. The knockdown efficiency looks not very strong, but there are clear effects in vivo. The authors could test the effect of this knockdown in vitro by stimulating the shRNA expressing organoids with recombinant Plexin B2 and test the proliferative capacity. They could do this in a competition assay whereby they combine knockdown vs wt organoids in the same well and treat with PlexinB2.

Author Rebuttals to Initial Comments:

We are grateful to the Referees for their detailed evaluation of our data and insightful remarks. Integrating their comments helped us to significantly improve the strength of our findings, or tone down our claims where necessary. We have now addressed all comments and performed multiple experiments that provide additional validation data to all figures and key findings. Briefly, these encompass but are not limited to validation of the role of Plexin B2 in controlling spontaneous metastasis from an orthotopic model of colorectal cancer, dissection of the mechanism by which plexin-semaphorin interactions induce epithelialization of liver metastases, characterization of Plexin B2 expression in the pre-metastatic and metastatic liver and analysis of semaphorin expression in human primary CRC datasets. Below we detail how we have addressed each point (reviewers' comments are in **bold dark blue**, and our responses are in black). For your convenience we have also included the resulting changes in the manuscript in *blue italics*.

Referees' comments:

Referee #1 (Remarks to the Author):

In this study, Borrelli and colleagues develop an in vivo CRISPR-A-based screen that combines transposon liver transduction of guide RNAs with proximity labeling of hepatocytes to identify host factors involved in liver metastasis formation. Using this methodology, they identify PlexinB2 as a liver-expressed molecule that facilitates metastasis formation by colorectal cancer (CRC) cells. Mechanistic studies suggest that PlexinB2 in hepatocytes interacts with class 4 Semaphorins expressed in metastatic cells and the activity of this pathway induces EMT reversion and survival of disseminated tumor cells during liver colonization. This is an interesting and relevant study. I found the screening method elegant and outstanding from a technical perspective. The data indicating that liver PlexinB2 expression levels modulate metastasis burden in models of intrasplenic tumor cell inoculation is also robust. In contrast, results supporting the proposed mechanistic action, i.e., reversion of EMT and survival, is very preliminary and not convincing. I am skeptical about this part. I detail my criticisms below:

1) Recent studies indicate liver metastasis formation upon implantation of CRC organoids in the primary site is due to dissemination of Lgr5 negative cell populations, which in the primary cancer undergo invasion and extravasation (Cañellas-Socias et al. Nature 2022; Fumagalli et al. Cell Stem Cell. 2020). In contrast, more simplistic metastasis models based on portal vein injection of CRC cells (as authors in this study) showed a dependency on Lgr5+ cancer stem cells (de Sousa E Melo. Nature 2017). These observations reveal substantial differences due to technical biases depending on the site of inoculation. It is, therefore crucial that authors show that PlexinB2-Semaphorin interactions are also required for the formation of liver metastases from the primary CRCs.

We thank the reviewer for the comment and have now conducted several experiments to address this point. We established a model of orthotopic CRC based on colonoscopy-guided submucosal injection of organoids¹, and tested the role of hepatocyte-derived Plexin B2 for the seeding of spontaneous liver metastases from primary colorectal tumors. We could validate the seeding promoting effect of Plexin B2 on spontaneous metastases from AKPS tumors. Moreover, to test the requirement of Plexin B2 for spontaneous liver seeding, we generated primary colon tumor with APTAK organoids, which exhibit an elevated metastatic potential², followed by hepatocyte-specific ablation of Plexin B2 in *Plexnb2^{flox/flox}* mice. In the absence of hepatocyte-derived Plexin B2, APTAK primary tumor growth was unaltered, however liver

colonization was abrogated. We therefore think that the observed phenotype is of general relevance (also for CRC metastases with different genetic makeup) and independent of the site of inoculation. The relative section and figures in the revised manuscript are copied below for the reviewer's convenience.

We further assessed the influence of hepatocyte-derived Plexin B2 on liver seeding of spontaneous metastases from colorectal tumors generated by colonoscopy-guided submucosal injection of organoids³ (Extended Data Fig 5a). Strikingly, 86% of the *Plxnb2* OE mice developed spontaneous liver metastases 8 weeks after orthotopic AKPS organoid inoculation, while only 29% of control littermates did (Fig. 2g, Extended Data Fig 5b, c). Conversely, *Plxnb2* deletion significantly decreased incidence and numbers of spontaneous liver metastases of *Apc*^{flax/flax}; *Tp53*^{flax/flax}; *Tgfb2*^{flax/flax}; *Kras*^{G12D}; *Akt1*^{myristoylated} (APTAK) organoids, which exhibit a high metastatic potential² (Fig. 2h, Extended Data Fig 5d-f).

Figure 2 | Hepatocyte-derived Plexin B2 is required for liver colonization. **g**, Incidence of spontaneous liver metastases in *Plxnb2* OE and control mice bearing primary AKPS tumors (Fisher's exact test, n = 7 mice per group). **h**, Incidence of spontaneous liver metastases in *Plxnb2* KO and control mice bearing primary APTAK tumors (n = 7 mice per group).

Extended Data Fig. 5 | Plexin B2 is required for spontaneous metastatic seeding. **a**, Representative micrograph of colonoscopy-guided submucosal injection of CRC organoids. Arrowhead indicates the site of injection (n = 7 mice per group). **b,c**, Representative H&E section and quantification of spontaneous APTAK metastases in *AlbCre*; *dCas9SPH* mice injected with *sgPlxnb2* OE or *sgNT*. **d**, Representative micrographs of primary APTAK tumors (bottom) and liver metastases in *Plxnb2*^{wt/wt} and *Plxnb2*^{flax/flax} mice injected with *AAV8-AlbCre-moxGFP*. **e, f**, Representative H&E section and quantification of spontaneous APTAK metastases in *Plxnb2*^{wt/wt} and *Plxnb2*^{flax/flax} mice injected with *AAV8-AlbCre-moxGFP* (n = 7 mice per group).

2) The data supporting that EMT reversion is the mechanism that explains the requirement for PlexinB2 in hepatocytes is far from convincing. This part should be largely strengthened.

- First, evidence that EMT occurs or is required for colonization in the model system used throughout the study is weak. Authors suggest that some cells in organoids remain in an

EMT-like state, but this is largely speculative. In addition, they inoculate single-cell suspensions via the spleen, which should be able to directly enter the liver parenchyma through the sinusoids without the need for invasion or migration.

- Second, the expression of Zeb1 is cytoplasmic (fig. 4d) and the staining seems not specific. None of the EMT hallmarks is modulated upon PlexinB2 knockdown or overexpression. The geneset enrichment related to epithelial morphogenesis does not directly support EMT reversion. Evidence is solely based on FoxP2 and Id3 regulation, but the relevance of these two genes in regulating the colonic epithelial phenotype is unclear.

To address these questions, authors should characterize by single cell profiling organoids and metastases with gain and loss of PlexinB2-Semaphorin signaling and provide additional evidence for alteration in the EMT state. Experiments of inoculation at the primary site (point 1) may also help clarify this issue.

We agree with the reviewer that our observations in cultured organoids do not directly support the notion that EMT occurs and is reverted in CRC metastasis. As our study is focused on liver colonization, rather than dissemination from the primary tumor, we re-formulated the text and removed any mention of “reversion of EMT”. However, we provide additional experimental evidence that implicates Plexin-semaphorin signaling in promoting epithelialization of liver metastases by means of suppression of mesenchymal traits, a process that can be generally referred to as mesenchymal-to-epithelial transition. Indeed, there is increasing evidence that, while primary epithelial tumors undergo dedifferentiation, secondary tumors in the liver are highly differentiated and epithelial⁴⁻⁷. The new results are listed below accompanied by the relative figure panels for the reviewer’s convenience. We think that cumulatively, these results support the notion that Plexin B2-induced acquisition of epithelial traits is a necessary step for liver colonization, as in absence of Plexin B2 hepatic seeding is suppressed, and the few remaining lesions exhibit mesenchymal traits.

- 1) As suggested by the reviewer, we performed single cell RNA sequencing (scRNASeq) of organoids in culture. AKPS organoids exhibit mutually exclusive expression of EMT and MET signatures (obtained from Gene Ontology, GO) (**Extended Data Fig 8a**). Due to the sparsity of scRNASeq, signatures of multiple genes are more robust than plotting of individual genes, yet it is important to note that EMT is controlled by a dynamic interplay of transcription factors, whose transcripts are challenging to capture by scRNASeq, and that are extensively post-translationally modified. The non-uniform EMT/MET state of AKPS organoids in culture corroborates our stainings showing a mix of E-cadherin^{high}Zeb1^{low} and E-cadherin^{low}Zeb1^{high} cells within the same organoid (**Extended Data Fig 8b**). Of note, we observe a significant upregulation of Zeb1 immunoreactivity in AKPS organoids, as compared to WT organoids, suggesting that transformation induces the acquisition of mesenchymal traits. Indeed McFalin-Figueroa *et al* showed that KRAS signaling drives the exit from an epithelial state⁸. As cell contacts are retained, as evidenced by E-cadherin staining, these data are consistent with a partial EMT state of AKPS organoids prior to inoculation. Indeed, partial EMT cells have been shown to upregulate EMT transcription factors but to retain proliferation, cell-cell contacts, apical-basal polarity and collective migration⁹. Moreover, partial EMT cells have been shown to exhibit a higher metastatic potential than fully mesenchymal cells⁹⁻¹². Whether cells that disseminated from primary CRCs exhibit a similar partial EMT state warrants further investigation, however it is outside of the scope of this study.

Extended Data Fig. 8 | a, Expression density of EMT and MET gene signatures in AKPS organoids profiled by scRNASeq. **b**, Representative fluorescent micrographs and quantification of E-cadherin and Zeb1 immunoreactivity in WT colon and AKPS organoids. Arrow shows E-cadherin^{high}Zeb1^{low} cell, arrowhead indicates E-cadherin^{low}Zeb1^{high}. Two-tailed unpaired T-test.

- 2) We further expanded our characterization of the EMT state of metastases growing in *Plxnb2*-overexpressing, -deficient and WT liver by staining for transcription factors that have been implicated in promoting epithelial traits in tumors by suppressing EMT, namely Elf3 and Grhl2^{13–17}. We found expression of these factors in metastases grown in *Plxnb2*-WT and -overexpressing livers, but absence thereof in the few lesions growing in *Plxnb2*-deficient livers (**Extended Data Fig 8c**). These stainings provide further evidence that, in absence of Plexin B2, metastases do not undergo epithelialization and therefore fail to adapt to the liver environment. Promotion of epithelial traits by Plexin-semaphorin interactions is further consistent with elevated Epcam immunoreactivity in AKPS metastases growing in *Plxnb2*-OE livers (**Extended Data Fig 7j**). These stainings provide further evidence that Plexin B2 levels in hepatocytes are inversely correlated with mesenchymal traits in mets, suggesting that Plexin B2 promotes epithelialization of metastases.

Extended Data Fig. 8 | c, Representative fluorescent micrograph and quantification of AKPS metastases in *Plxnb2* OE, *Plxnb2* KO and control livers. GRHL2, ELF3 and Zeb1 shown in red, F-actin shown in gray, DAPI in blue used as nuclear

counterstain. Scale bar, 50 μ m. Inset 1 shows GRHL2⁺ nuclei in *Plxnb2* OE livers. Inset 2 shows Zeb1⁺ nuclei in *Plxnb2* KO livers. Scale bar, 50 μ m.

Extended Data Fig. 7 | j, Representative fluorescent micrograph and quantification of Epcam immunoreactivity in AKPS metastases in *Plxnb2* OE and control livers. Scale bar, 20 μ m. Dots represent individual metastases in n = 2 mice.

- 3) To characterize the EMT profile of metastases *in vivo*, we performed combined single nucleus RNA and ATAC sequencing data from metastatic livers overexpressing *Plxnb2* or controls. We reasoned that while transcription factor levels might be technically challenging to detect and quantify by mRNA sequencing, their chromatin occupancy would be a more faithful proxy of their activity. We used the transcriptome space to annotate our dataset, and then computed differentially open peaks between tumor cells in *Plxnb2*-OE vs control livers (**Extended Data Fig 7d,e**). With respect to controls, metastases growing in *Plxnb2* OE livers exhibited increased activity of several members of the SP/KLF family of zinc finger transcription factors¹⁸ (**Fig. 3g**). In line with these results, we detected increased nuclear levels of Krüppel-like factor 4 (Klf4) in AKPS metastases growing in *Plxnb2* OE livers, while it was absent from lesions in *Plxnb2* KO livers (**Fig. 3h**). Klf4 is expressed by differentiated cells of the colonic epithelium, but its expression is lost in CRC, where it acts as tumor suppressor by inhibiting epithelial-mesenchymal transition (EMT)^{19–26}. Strikingly, Klf4 is absent in lesions growing in *Plxnb2*-KO livers, and this coincides with retained Zeb1 nuclear expression. We now provide additional stainings that indicate nuclear Zeb1 expression in AKPS metastases growing in *Plxnb2*-KO livers (**Fig. 3h** and **Extended Data Fig 8c**, see above)

Extended Data Fig. 7 | Multiomic characterization of AKPS metastases in *Plxnb2* OE and control livers. d, UMAP of RNA profile of nuclei isolated from *Plxnb2* OE and control livers harboring AKPS metastases. Colors indicate cell types. **e**, Marker gene expression of distinct cell types in dataset.

Figure 3 | Plexin B2 induces epithelialization of liver metastases. **f**, UMAP of chromatin profile of epithelial cells in AKPS metastases in *Plxnb2* OE and control livers profiled by snATACseq. **g**, Enriched TF binding motifs in differentially open peaks. **h**, **i**, Representative fluorescent micrograph and quantification of AKPS metastases in *Plxnb2* OE, *Plxnb2* KO and control livers. Klf4 shown in red, Zeb1 shown in red, E-cadherin in gray, DAPI in blue used as nuclear counterstain. Scale bar, 50 μ m. Arrows and inset shows Klf4⁺ or Zeb1⁺ nuclei. Asterisk indicates background. Scale bar, 10 μ m. **h**, **i** Ordinary one-way ANOVA, comparing each treatment group to control. Tukey's correction for multiple testing. Dots represent individual nuclei in $n = 3$ mice. Barplots indicate mean \pm SD.

- 4) We treated two-dimensional cultures of AKPS organoids with a small-molecule inhibitor of KLF4 (WX2-43), which specifically abrogates the interaction between PRMT5 and KLF4, thereby enhancing KLF4 degradation²⁷. Treatment induced acquisition of a mesenchymal morphology, decreased colony size, F-actin cytoskeleton and proliferation (**Extended Data Fig 7k,l**). These results strongly suggest that Klf4 is required to maintain suppression of mesenchymal programs in AKPS organoids.

Extended Data Fig. 7 | k,l Representative brightfield and fluorescent micrograph and quantification of two-dimensional AKPS cultures treated with the Klf4 inhibitor WX2-43²⁷. F-actin shown in gray, EdU in red, DAPI in blue. Scale bar, 10 μ m. **a,c,h,i,j,l** Two-tailed unpaired T-test, **b,g** Ordinary one-way ANOVA, comparing each treatment group to vehicle control. Tukey's correction for multiple testing. **a,b,g,h,i,l** Dots represent individual wells.

- 5) We assessed the epithelial state of AKPS metastases lacking class IV semaphorins and found that they lose Klf4 and retain Zeb1 expression, indicating that Plexin B2 on hepatocytes signals

through the semaphorin/Klf4 axis to suppress mesenchymal traits (for details see response to point 4).

Cumulatively, the new evidence provided strongly indicates that Plexin-semaphorin signaling induces epithelialization of CRC metastases, and that this is a rate-limiting step for liver colonization.

3) The proposed model is difficult to reconcile with the data shown. PlexinB2 is a membrane-bound protein expressed in hepatocytes. Authors propose that PlexinB2-Semaphorin interactions between disseminated cancer cells and hepatocytes at the time of metastatic colonization are necessary for MET. But why the acquisition of epithelial features is sustained in large metastases of *Plxnb2* deficient livers where hepatocytes no longer interact with tumor cells because of TME recruitment? Even if Semaphorin are shed and act as soluble ligands, how would cancer cells be instructed by hepatocytes that are not interacting directly with cancer cells, and through which mechanism MET is enforced in these metastases? can authors rule out that decreased desmoplastic reactions that characterize metastasis growing in livers overexpressing PlexinB2 indirectly affect the organization of epithelial cells, including changes in Epcam levels?

Our data suggest that in the absence of Plexin B2, metastatic colonization is almost completely abolished, and that the few lesions remaining retain mesenchymal traits. Conversely, upon Plexin B2 overexpression we observed increased colonization and enhanced epithelialization of metastases. This led us to speculate that Plexin B2 pushes AKPS cells towards a more epithelial state, increasing their ability to adapt to and colonize the liver environment. We proposed that re-epithelialization of cancer cells is necessary for liver colonization, which was not shown for colorectal cancer so far. We now provide additional evidence to support this hypothesis (see above), and show that genetic ablation of semaphorins in AKPS metastases (see response to point 4) fully phenocopies loss of Plexin B2 on hepatocytes, supporting the notion that Plexin-semaphorin interactions are required to suppress mesenchymal traits in metastasis.

However, we agree that our data do not fully explain how epithelial traits are retained in metastases after seeding, and that the observed changes in the metastatic microenvironment in *Plxnb2*-overexpressing livers raise the possibility that the stromal or endothelial compartment affect the phenotype of AKPS metastases. To address this point, we conducted thorough analysis of the microenvironmental alternations in pre-metastatic and metastatic livers with altered Plexin B2 levels, as well as several *in vitro* and *in vivo* experiments to disentangle the effect of manipulating Plexin B2 levels from that of the TME. The new data are presented below.

- 1) We transcriptionally profiled livers overexpressing *Plxnb2* and controls before inoculation of AKPS organoids, and found upregulation of gene sets involved in proliferation, and downregulation of genes related to xenobiotic metabolism. However, we found no evidence of a significant change in cellular composition when applying cell-type decomposition with the Enrichr platform (using Tabula Muris as a reference, **Extended Data Fig. 6d-f**). This corroborated our previous findings indicating unaltered CD146 and α -SMA immunoreactivity upon *Plxnb2* overexpression (**Extended Data Fig. 6c**).

Extended Data Fig. 6 | AKPS metastases in *Plxnb2* OE livers have an altered stromal and endothelial compartment. **c**, Representative fluorescent micrographs and quantification of *Plxnb2* OE and control livers showing nuclear DAPI stain (blue), and α -SMA (green) and CD146 (red) immunoreactivity. Scale bar, 100 μm . **d,e**, DEGs and GSEA in *Plxnb2* OE vs. control livers (n = 3). **f**, Cell type enrichment analysis in bulk RNASeq of *Plxnb2* OE vs. control livers (n = 3).

- 2) We characterized and quantified in more depth the metastatic microenvironment by means of multiplexed immunofluorescence (Lunaphorb Comet) and single nucleus RNA sequencing of *Plxnb2*-OE and WT livers harboring AKPS metastases. We found a significant increase in epithelial areas (E-cadherin) in the tumor, at the expense of the stromal compartment (α -SMA) (Fig 3j). As shown before, the stromal compartment lacked expression of POSTN1, a marker of fibroblast activation. Furthermore, AKPS metastases in *Plxnb2*-OE livers exhibited an extensive CD146⁺ vascular network (Extended Data Fig. 6a, b). Interestingly, lesions in *Plxnb2* OE livers also exhibited strong downregulation of genes involved in immune recognition such as *Cd74*, *B2m* and *H2-D1*, which coincided with diminished CD4⁺ T cell infiltration (Fig. 3d, Extended Data Fig. 7f,i).

Figure 2 | j, Representative fluorescent micrograph and quantification of AKPS metastases in *Plxnb2* OE and control livers. α -SMA shown in magenta, E-cadherin shown in green, DAPI used as nuclear counterstain. Scale bar, 100 μm . Barplots indicate mean area density \pm SD. Dots represent individual metastases from n=2 mice, two-tailed unpaired t-test.

Figure 3 | c, UMAP of RNA profile of epithelial cells of AKPS metastases in *Plxn2 OE* and control livers profiled by snRNAseq. **d**, Gene set enrichment analysis of DEGs in epithelial cells of AKPS metastases in *Plxn2 OE* vs control livers.

Extended Data Fig. 7 | Multiomic characterization of AKPS metastases in *Plxn2 OE* and control livers. f, Differential gene expression in tumor cell nuclei isolated from *Plxn2 OE* vs control livers. **i**, Representative fluorescent micrograph and quantification of AKPS metastases in *Plxn2 OE* and control livers showing nuclear DAPI stain (gray), and E-cadherin (yellow), CD4 (cyan, T cells) and CD68 (magenta, macrophages) immunoreactivity. Scale bar, 20 μm . Dots represent individual metastases in $n = 2$ mice.

3) We transcriptionally profiled AKPS organoids treated with rmPlexin B2 *in vitro*. Treatment with rmPlxn2 induced a compositional shift in AKPS organoids from a cell population characterized by high levels of oxidative phosphorylation, to a more proliferative population with elevated MAPK and JNK pathway (Fig 3a). These results are in line with an induction of epithelial traits, as mesenchymal cells are often associated with low proliferative and respiratory states^{28,29}. Moreover, rPlexin B2 increased frequencies of EdU⁺ cells in AKPS organoids and PDOs (Extended Data Fig. 7a,b), and lesions in *Plxn2 OE* livers further harbored higher density of Ki67⁺ epithelial cells and lower cleaved-caspase 3 levels, indicating a proliferative advantage *in vitro* and *in vivo* (Fig. 3b, Extended Data Fig. 7c).

Figure 3 | Plexin B2 induces epithelialization of liver metastases. a, UMAP of rmPlexin B2-treated AKPS organoids and controls profiled by scRNASeq. Colors indicate distinct transcriptional clusters. Significantly enriched gene ontology terms and cluster proportions per sample are shown. **b**, Representative fluorescent micrograph and quantification of AKPS metastases in *Plxn2 OE* and control livers. E-cadherin shown in green, Ki67 shown in magenta, DAPI used as nuclear counterstain. Scale bar, 100 μm .

Extended Data Fig. 7 | a, Edu⁺/Edu⁻ cells in two-dimensional AKPS organoid cultures treated with rmPlexin B2. **b**, Left, Edu⁺/Edu⁻ cells in two-dimensional PDO cultures treated with rhPlexin B2. Right, representative fluorescent micrograph of Edu⁺ PDO colonies (right). Scale bar, 10 μ m. **c**, Representative fluorescent micrograph and quantification of AKPS metastases in *Plxn2* OE and control livers showing nuclear DAPI stain (cyan), and E-cadherin (gray) and cleaved caspase-3 (yellow) immunoreactivity. Dots represent individual metastases in n = 2 mice.

- 4) We injected AKPS organoids treated *ex vivo* with recombinant Plexin B2 in naive livers, and observed an increased grafting rate compared to untreated organoids. Interestingly, pretreatment with a Rac1 inhibitor abolished this effect, supporting the notion that Plexin B2 binds to class IV semaphorins on tumor cells as these have been reported to activate reverse signaling via Rac1³⁰⁻³².

Figure 4 | Plexin B2 binds semaphorins on tumor cells. a, Metastatic foci per liver section 2 weeks post intrasplenic injection of AKPS organoids treated *ex vivo* with rmPlexinB2 and/or the Rac1 inhibitor NSC323766. Dots represent individual mice (n=3).

- 5) Finally, we ablated *Plxn2* in hepatocytes 5 days after intrasplenic inoculation of AKPS^{Luciferase;zsGreen} organoids, and monitored tumor growth by IVIS. Interestingly, liver colonization was delayed but not impaired by *Plxn2* ablation after tumor establishment, suggesting that the presence of Plexin B2 is not required to sustain tumor growth.

Extended Data Fig. 5 | Plexin B2 is required for spontaneous metastatic seeding. l, BLI signal over time in *Plxn2* KO and control mice injected with AAV8-*AlbCre-moxGFP* (AAV) 5 days post intrasplenic injection of AKPS organoids and (n =

3). Two-tailed unpaired T-test. **m**, Representative H&E section of AKPS metastases in livers of *Plxnb2* KO and control mice injected with AAV8-*AlbCre-moxGFP* (AAV) 5 days post intrasplenic injection.

Together with our *in vitro* assays and the observed grafting impairment of AKPS^{Sema4KD} and AKPS^{Sema4KO} organoids in naive livers (see point 4 below), these results strictly implicate Plexin-semaphorin interactions in promoting early seeding events occurring upon DTC extravasation in the liver, prior to the establishment of a growth promoting niche. We hypothesize that Plexin-semaphorin interactions increase the rate of successful seeding events by inducing the acquisition of epithelial traits in extravasated DTCs, which in turn promotes proliferation and their adaptation to the liver environment. It is being increasingly recognized how distinct tumor states (genetic, epigenetic and transcriptional) dictate the composition of their microenvironments³³. Changes in the metastatic microenvironment of metastases grown in *Plxnb2*-OE vs WT livers are therefore likely a consequence of an altered tumor state, rather than a cause. In turn, the microenvironment might then support and reinforce an epithelial state of established metastases when Plexin-semaphorin interactions are no longer required.

4) The manuscript will also gain relevance if the authors could provide insights into the mechanism by which Semaphorin signaling triggers EMT reversion in AKPS organoids.

We thank the reviewer for the comment and now provide new experimental evidence implicating the transcription factor Klf4 as mediator of Plexin B2-induced epithelialization downstream of semaphorins. Our multiomic analysis of metastases in *Plxnb2* overexpressing livers indicated a significant enrichment of motifs bound by Klf4, which has recently been implicated in suppressing EMT²³, which we validated on the protein level by immunofluorescence (see above). To test whether signaling via semaphorins is required for Klf4 induction, we generated quadruple KO AKPS organoids (AKPS^{Sema4KO}) lacking *semaphorin 4A, C, D and G* with enhanced enAsCas12a technology. Quadruple editing was validated by NGS and immunofluorescence in multiple clonal lines (see point 7). While AKPS^{Sema4KO} organoids show unaltered proliferation *in vitro*, they exhibit significant grafting impairment *in vivo*, as they fail to form primary tumors and metastases when inoculated orthotopically.

Extended Data Fig. 10 | Class IV semaphorins on tumor cells are required for metastatic seeding. **f**, Edu⁺/Edu⁻ cells in two-dimensional AKPS^{Sema4KO} and AKPS^{Sema4WT} organoids. **g**, Representative images of livers (top) and primary colon tumors (bottom) in mice orthotopically injected with AKPS^{Sema4KO} or AKPS^{Sema4WT} organoids. **h**, Incidence of spontaneous liver metastases in mice orthotopically injected with AKPS^{Sema4KO} or AKPS^{Sema4WT} organoids (n = 8).

To assay liver colonization, we injected AKPS^{Sema4KO} organoids intrasplenically, and found significantly decreased metastatic burden compared to control organoids (**Fig. 4e,f**). Strikingly, Klf4 immunoreactivity is lost in AKPS^{Sema4KO} metastases, which further exhibit diminished Epcam and E-cadherin expression and high Zeb1 levels (**Fig. 4g**). Loss of semaphorins in AKPS metastases thus phenocopies loss of Plexin B2 on hepatocytes, supporting the notion that Plexin-semaphorin interactions are required for metastatic seeding (**Fig. 4h**).

Figure 4 | Plexin B2 binds semaphorins on tumor cells. **e**, Metastatic foci per liver section 2 weeks post intrasplenic injection of AKPS^{Sema^{WT}} or AKPS^{Sema^{4KO}} organoids. Dots represent individual mice (n=3). **f**, Representative H&E staining of AKPS^{Sema^{WT}} or AKPS^{Sema^{4KO}} metastases. Scale bar, 100 μ m. **g**, Representative fluorescent micrograph and quantification of Epcam (in gray) and Klf4 (in red) immunoreactivity in AKPS^{Sema^{WT}} and AKPS^{Sema^{4KO}} metastases. Scale bar, 20 μ m. Arrow and inset show Klf4⁺ nuclei. Scale bar, 10 μ m. **h**, Representative fluorescent micrograph and quantification of E-cadherin (in gray) and Zeb1 (in red) immunoreactivity in AKPS^{Sema^{WT}} and AKPS^{Sema^{4KO}} metastases. Scale bar, 20 μ m. Inset 1 shows E-cadherin⁺Zeb1⁺ nuclei. Inset 2 shows E-cadherin⁺Zeb1⁻ nuclei. Scale bar, 10 μ m. **i**, Plexin B2 on hepatocytes binds class IV semaphorins on DTCs, inducing Klf4 upregulation and Zeb1 downregulation, which results in epithelialization and outgrowth of metastases. a,b,c Dots indicate individual wells. Ordinary one-way ANOVA, comparing each treatment group to control. Tukey's correction for multiple testing. d,e,g,h Two-tailed unpaired t-test. Barplots indicate mean \pm SD.

We further identified a dependency on Rac1 signaling both for the in vitro seeding promoting effects of rhPlexin B2 on PDOs, and the in vivo seeding promoting effect of rmPlexin B2 on AKPS organoids (Fig 4a,c). Moreover, we show that both treatment with recombinant PlexinB2 and with a Klf4 inhibitors alters the cytoskeletal organization of tumor cells in culture.

Figure 4 | Plexin B2 binds semaphorins on tumor cells. **a**, Metastatic foci per liver section 2 weeks post intrasplenic injection of AKPS organoids treated ex vivo with rmPlexinB2 and/or the Rac1 inhibitor NSC323766. Dots represent individual mice (n=3). **c**, CFU per well of PDOs cultured with 2 μ g/mL rhPlexin B2 and NSC 23766.

Extended Data Fig. 7 | g, Representative fluorescent micrograph and quantification of F-actin in two-dimensional PDO cultures treated with rhPlexin B2. Scale bar, 10 µm. **h**, F-actin intensity in KPC cultures treated with rmPlexin B2. **k, l** Representative brightfield and fluorescent micrograph and quantification of two-dimensional AKPS cultures treated with the Klf4 inhibitor WX2-43. F-actin shown in gray, EdU in red, DAPI in blue. Scale bar, 10 µm. a,c,h,i,j,l Two-tailed unpaired T-test, b,g Ordinary one-way ANOVA, comparing each treatment group to vehicle control. Tukey's correction for multiple testing. a,b,g,h,l, Dots represent individual wells.

Previous studies have shown regulatory relationships between Rac1 signaling and Klf4 activation^{34–36}, however a fine mechanistic dissection of the molecular pathways that act downstream of semaphorins to induce Klf4-mediated transcription in AKPS metastases is still outstanding. While the redundant functions of class IV semaphorins complicate experimental studies, work to reveal the molecular mediators of reverse signaling by semaphorins is ongoing, and will be the focus of future studies.

5) The data on cancer cell death versus proliferation triggered by PlexinB2-Semaphorin signaling on DTCs is neither clear. How is it connected to the MET reversion process? According to the authors, organoids are in an EMT state yet proliferate and survive. To address these issues, authors must investigate the kinetics of metastasis (using bioluminescence and/or histology) and assess the numbers of proliferative and apoptotic cells during liver colonization.

We agree with the reviewer that mesenchymal states are generally associated with low proliferation. However, due to the presence of cell-cell contacts we hypothesize that AKPS organoids are in a hybrid or mixed EMT state, which is characterized by upregulation of EMT TFs together with maintained proliferation⁹. We hypothesize that Plexin-semaphorin signaling suppresses mesenchymal programs in AKPS metastases, inducing epithelialization and promoting proliferation. In line with this hypothesis, single-cell transcriptional profiling indicates that *in vitro* treatment of AKPS organoids with rmPlexin B2 induced a compositional shift towards a more proliferative cell population (**Fig 3a**). Moreover, rPlexin B2 increased frequencies of EdU⁺ cells in AKPS organoids and PDOs (**Extended Data Fig. 7a,b**). Lesions in *Plexnb2* OE livers further harbored higher density of Ki67⁺ epithelial cells and lower cleaved-caspase 3 levels, indicating a proliferative advantage *in vivo* (**Fig. 3b, Extended Data Fig. 7c**).

Figure 3 | Plexin B2 induces epithelialization of liver metastases. **a**, UMAP of rmPlexin B2-treated AKPS organoids and controls profiled by scRNASeq. Colors indicate distinct transcriptional clusters. Significantly enriched gene ontology terms and cluster proportions per sample are shown. **b**, Representative fluorescent micrograph and quantification of AKPS metastases in *Plxn2* OE and control livers. E-cadherin shown in green, Ki67 shown in magenta, DAPI used as nuclear counterstain. Scale bar, 100 μ m.

Extended Data Fig. 7 | a, Edu⁺/Edu⁻ cells in two-dimensional AKPS organoid cultures treated with rmPlexin B2. **b**, Left, Edu⁺/Edu⁻ cells in two-dimensional PDO cultures treated with rhPlexin B2. Right, representative fluorescent micrograph of Edu⁺ PDO colonies (right). Scale bar, 10 μ m. **c**, Representative fluorescent micrograph and quantification of AKPS metastases in *Plxn2* OE and control livers showing nuclear DAPI stain (cyan), and E-cadherin (gray) and cleaved caspase-3 (yellow) immunoreactivity. Dots represent individual metastases in $n = 2$ mice.

We further monitored AKPS metastases in *Plxn2*-KO livers by lentivirally introducing the pLVX-fireflyLuc-IRES-zsGreen1 vector in AKPS organoids and monitoring tumor growth in vivo by bioluminescent imaging (BLI). While Plexin B2 deletion prior to tumor inoculation significantly reduced BLI signal and tumor burden, liver colonization was delayed, but not impaired, when *Plxn2* was ablated 5 days after tumor inoculation, suggesting that the presence of Plexin B2 is not required to sustain tumor proliferation.

Figure 2 | Hepatocyte-derived Plexin B2 is required for liver colonization. **f**, BLI signal over time in *Plxn2*^{flx/flx} or *Plxn2*^{wt/wt} mice injected with AAV8-*AlbCre-moxGFP* and subsequently intrasplenically injected with AKPS^{Luciferase;zsGreen} organoids (Fisher's exact test, $n = 3$).

Extended Data Fig. 5 | Plexin B2 is required for spontaneous metastatic seeding. **i**, BLI signal over time in *Plxn2* KO and control mice injected with AAV8-*AlbCre-moxGFP* (AAV) 5 days post intrasplenic injection of AKPS organoids and ($n = 3$). Two-tailed unpaired T-test. **m**, Representative H&E section of AKPS metastases in livers of *Plxn2* KO and control mice injected with AAV8-*AlbCre-moxGFP* (AAV) 5 days post intrasplenic injection.

6) A more detailed analysis of PlexinB2 expression in the liver and interactions with Semaphorin+ tumor cells during organ colonization and outgrowth is also needed. Are PlexinB2-expressing hepatocytes localized in proximity to the portal vein and liver sinusoids? Do they contact Semaphorin+ tumor cells?

We analyzed Plexin and Semaphorin expression by multiplexed FISH and protein stainings. We copy below the relative result section and figures.

At steady state, Plexin B2 is widely expressed by hepatocytes - with higher expression in portal areas (Extended Data Fig 5g, h) - as well as endothelial cells of the hepatic sinusoids and portal veins³⁷.

Extended Data Fig. 5 | g, Multiplexed *in situ* hybridization (Molecular Cartography) of *Plxnb2* as well as portal and central hepatocyte markers in WT mouse liver. **h**, Quantification of *Plxnb2* mRNA abundance in portal and central areas. Two-tailed unpaired Wilcoxon test.

Interestingly, while Plexin B2 immunoreactivity was higher in peritumoral hepatocytes (Extended Data Fig. 5i,j), Plxnb2 expression was unaltered in livers of mice bearing AKPS colon tumors, suggesting that Plxnb2 is not upregulated by primary tumor secreted factors nor systemic effects, but rather due to a local response to the presence of metastases (Extended Data Fig. 5k).

Extended Data Fig. 5 | i, Representative fluorescent micrograph of Plexin B2 expression in AKPS liver metastasis. Scale bar, 100 μ m. **j**, Plexin B2 staining intensity in peritumoral and distal areas. Dots represent individual metastases in n=3 mice. Two-tailed unpaired T-test. **k**, Differential gene expression analysis in livers of primary colon tumor-bearing vs naive mice. n = 3.

AKPS metastases express multiple semaphorins reported to interact with Plexin B2, namely Sema4a, Sema4c, Sema4d, Sema4g (Extended Data Fig. 9a), and Sema4A⁺ tumor cells contact hepatocytes at the metastasis leading edge (Extended Data Fig. 9b).

Extended Data Fig. 9 | Class IV semaphorin expression in murine and human CRC. **a**, Multiplexed *in situ* hybridization (Molecular Cartography) and quantification of *Sema4a*, *4c*, *4d*, *4g* and *Plexnb2* transcripts in AKPS metastases. **b**, Representative fluorescent micrograph of *Sema4A*⁺ tumor cells (red) in contact with GFP⁺ hepatocytes (green). DAPI in blue as nuclear counterstain.

6) Many assumptions are based on the interpretation of individual stainings, but no quantifications are provided. This is particularly relevant in Figure 4d and extended data in fig 5.

Quantifications are now provided for all stainings.

7) Authors show that combined shRNA-mediated knockdown of four semaphorin-4 members reduces metastatic capacities of AKPS organoids. However:

- Downregulation of each individual semaphorin at RNA levels only ranges between 20%-50%, which makes me suspicious that the lower metastatic capacity of the KD organoids could be due to off-target effects.
- Proper controls (shRNA targeting of 4 neutral sequences and additional shRNA targeting semaphorins) are not included.

Authors should provide appropriate controls for this experiment and characterize changes in MET/epithelial organization of liver metastasis upon Semaphorin downregulation. Including additional experiments of semaphoring knock-down or knock-out by other genetic means or activity inhibition *in vivo* (e.g., using soluble PlexinB2 recombinant protein inoculation) would make this part more convincing.

To address this point and achieve a complete depletion of semaphorins, we generated quadruple KO AKPS organoids (AKPS^{qKO}) lacking *Semaphorin 4A, C, D and G* with enAsCas12a technology, which enables highly efficient multiplexed genome editing. As Cas12a was integrated stably in organoids via lentiviral technology, we used organoids expressing only Cas12a as controls. Quadruple editing was validated by NGS and immunofluorescence in multiple clonal lines.

Extended Data Fig. 10 | Class IV semaphorins on tumor cells are required for metastatic seeding. **e**, Representative fluorescent micrograph of Sema4A, C, D and G immunoreactivity in AKPS^{Sema4KO} and AKPS^{Sema4WT} organoids. Scale bar, 20 μ m.

Sema4a	WT		AGCCAAGGGAACCAACAGTGAATGCTCTCTGGCTGCTCTTCAAGGCAGGTTGGCGGGACCAAGGAGCTCAGCAGCTGTGTCCTCTCTCTCACGGACATTGAGCGAGTCTTTAAAGGGAAGTACAAGGAGCTGAACAAGGAGACCTCCCG	PREPQMSSSLAVFRQVGGTRSSAVCAFLSLTDIERVFKGKYKELNKETS
	Sema4KO clone 3	23% of reads	AGCCAAGGGAACCAACAGTGAATGCTCTCTGGCTGCTCTTCAAGGCAGGTTGGCGGGACCAAGGAGCTCAGCAGCTTTAAAGGGAAGTACAAGGAGCTGAACAAGGAGACCTCCCGCTGAGCCACTTACC	PREPQMSSSLAVFRQVGGTRSSAVFKGKYKELNKETSRTWTTYRGSEVSPX
		77% of reads	AGCCAAGGGAACCAACAGTGAATGCTCTCTGGCTGCTCTTCAAGGCAGGTTGGCGGGACCAAGGAGCTCAGCAGCTGTGTCCTCTCTCTGAGCTGAAACAAGGAGACCTCCCGCTGAGCCACTTACC	PREPQMSSSLAVFRQVGGTRSSAVCAFLSLSTRRPPAGPLTGAQRSARGX
Sema4c	WT		TTCCCTGTTCCTGGTATGGTGGGAGCAAGTAGGCACCCCAACCTGACTTCCAAGTCTCTTTAACAGGGGCGATATGGACCTGTCTGAGTGTGAGTACCAGTTGGAACAGATCCAGCAAGTGTGGAGGTCCTACAAGGAGTA	PLFPGMVGASRHPTLPRSFNFRGDMDSLAVCEYQLEIQVFEQPYKEX
	Sema4KO clone 3	37% of reads	TTCCCTGTTCCTGGTATGGTGGGAGCAAGTAGGCACCCCAACCTGACTTCCAAGTCTCTTTGTGAGTACCAGTTGGAACAGATCCAGCAAGTGTGGAGGTCCTACAAGGAGTA	PLFPGMVGASRHPTLPRSFVYSTWNRSSKCLRVPTRSTVSKRSPGAPX
		63% of reads	TTCCCTGTTCCTGGTATGGTGGGAGCAAGTAGGCACCCCAACCTGACTTCCAAGTCTCTTTAACAGGGGCGATATGGTCTGAGTGTGAGTACCAGTTGGAACAGATCCAGCAAGTGTGGAGGTCCTACAAGGAGTACAGTGTG	PLFPGMVGASRHPTLPRSFNFRGDMVCSLVPVGTDPASVGLQGVQX
Sema4d	WT		AAGGGTCTTTGATGGTGCCTGTTAATGGGGTGTTCCTCTGAGCAGCGAATGCCTAACTACATTGAGTACTACAGCCCAAGCAGCACTTCCCTCTATGTGTGGGACCAATGCGTTCAGCCCACTGTGACCACTGTAAGA	RVLWVPVNGVFPQLTECLNIRVLRQLPSSLSYVCGTNAFQPTCDHLVR
	Sema4KO clone 3	54% of reads	AAGGGTCTTTGATGGTGCCTGTTAATGGGGTGTTCCTCTGAGCAGCGAATGCCTAACTACAGTACTACAGCCCAAGCAGCACTTCCCTCTATGTGTGGGACCAATGCGTTCAGCCCACTGTGACCACTGTAAGA	RVLWVPVNGVFPQLTECLNEYSHAALPSCMCGPMRSPVPTTWDHX
		46% of reads	AAGGGTCTTTGATGGTGCCTGTTAATGGGGTGTTCCTCTGAGCAGCGAATGCCTAACTACAGCCCAAGCAGCACTTCCCTCTATGTGTGGGACCAATGCGTTCAGCCCACTGTGACCACTGTAAGA	RVLWVPVNGVFPQLTECLNIRVLRQLPSSLSYVCGTNAFQPTCDHLVR
Sema4g	WT		TGATAAGCTAGACTAATGGAGGTGGCTGTCTCTGGGTTGATGAGTCTTCACTGCCTCCCTCTGCTGTAGACCGAGTGTTCACCAACAGTGGGATTTCTACAGCGGCTCAATGCCACCCTCTTATGATGTGGGACCAACGCTT	DKLDWRWLVSLMSSITASLCLTECFNHRVFLRQLNATHYACGTHAX
	Sema4KO clone 3	22% of reads	TGATAAGCTAGACTAATGGAGGTGGCTGTCTCTGGGTTGATGAGTCTTCACTGCCTCCCTCTGCTGTAGACCGAGTGTTCACCAACAGTGGGATTTCTACAGCGGCTCAATGCCACCCTCTTATGATGTGGGACCAACGCTT	DKLDWRWLVSLMSSITASLCLTECFNHRVFLRPLPAPLCSYCEACPL
		78% of reads	TGATAAGCTAGACTAATGGAGGTGGCTGTCTCTGGGTTGATGAGTCTTCACTGCCTCCCTCTGCTGTAGACCGAGTGTTCACCAACAGTGGGATTTCTACAGCGGCTCAATGCCACCCTCTTATGATGTGGGACCAACGCTT	DKLDWRWLVSLMSSITASLCLTECFNHRVFLRPLPAPLCSYCE

Supplementary Table 2 | Indel analysis of AKPS^{Sema4KO} (clonal line 3) showing out of frame mutations in all 4 targeted semaphorin genes. Green rows show WT base and amino acid sequence. Mutated parts shown in red.

While AKPS^{qKO} organoids show unaltered proliferation in vitro, as compared to control organoids, they exhibit significant impairment in colon and liver colonization (see response to point 4). Strikingly, Klf4 is lost in AKPS^{qKO} metastases, which further low Epcam and Ecadherin expression and high Zeb1. Loss of semaphorins in AKPS metastases phenocopies loss of Plexin B2 on hepatocytes (see response to point 4).

8) The relevance of the author's findings for human CRC is only supported by a few Kaplan-Meier plots based on bulk transcriptomic data. Given the wealth of scRNAseq datasets of CRC samples, it would be interesting to assess whether Semaphorins are expressed in particular tumor cell states that have been associated with metastasis formation (Lgr5 negative cells, HCRs, etc). Plexin B2 and Semaphorin expression analyses by ISH in human samples would also strengthen the study.

We thank the reviewer for the comment and we now conducted a deeper analysis of semaphorin gene mutational status and expression in CRC patients.

SEMA4A, C, D and G are detected in human colon and CRC, and, in two published scRNASeq datasets of human CRC³⁸. Expression of the semaphorin genes is significantly upregulated in high-relapse (HR) epithelial cells³⁹ and in intrinsic consensus molecular subtype 3 (iCMS3) cells⁴⁰, but not in Lgr5^{high} cells, and it coincides with a MET signature (Extended Data Fig. 9c-e). Semaphorin expression is further unaltered in liver metastases, as compared to matched primary tumors in two independent scRNASeq datasets of metastatic CRC^{41,42} (Extended Data Fig. 9f). Copy number variation (CNV) analysis in the COAD dataset⁴³ further revealed that SEMA4A, SEMA4C and SEMA4D are commonly found amplified, while SEMA4G is often deleted in CRC patients (Extended Data Fig. 9g). Moreover, in a large cohort of CRC patients, increased expression of SEMA4A, SEMA4C and SEMA4D, but not SEMA4G, is associated with reduced recurrence-free survival (Extended Data Fig. 9h).

Extended Data Fig. 9 | c, Immunohistochemistry for SEMA4A, C, D, and G in healthy and human colon obtained from the Human Protein Atlas (<https://www.proteinatlas.org/>). **d**, Left, Averaged expression of *Sema4a*, *4c*, *4d*, *4g* in scRNASeq datasets of human primary CRC (only epithelial cells). Cells grouped as high-relapse cells (HRCs), Lgr5 cells or others. Right, averaged expression of *Sema4a*, *4c*, *4d*, *4g* projected on epithelial cells of KUL dataset together with Wnt ON signature or MET signature. **e**, Averaged expression of *Sema4a*, *4c*, *4d*, *4g* in scRNASeq datasets of human primary CRC (only epithelial cells). Cells grouped as iCMS2 or iCMS3. **f**, Averaged expression of *Sema4a*, *4c*, *4d*, *4g* in epithelial cells from CRC liver metastasis or matched primaries in two independent datasets. **g**, CNV analysis *SEMA4A*, *SEMA4C*, *SEMA4D* and *SEMA4G* in COAD dataset. $n = 290$ patients. **h**, Kaplan-Meier analysis (<http://kmpplot.com>) of the recurrence free survival (RFS) of CRC patients stratified by *SEMA4A*, *SEMA4C*, *SEMA4D* and *SEMA4G* expression. $n = 1211$ patients. **d,e,f** Two-tailed unpaired Wilcoxon test.

The upregulation of class IV semaphorins by HRCs, a distinct cell state found in primary CRC that were recently implicated in metastatic dissemination and recurrence¹⁴, is in line with the proposed role of plexin-semaphorin in mediating liver colonization. Moreover, plexin-semaphorin interactions were recently identified as mediating formation of homotypic and heterotypic clusters of circulating cancer cells⁹⁴. In light of the colon grafting impairment of semaphorin-deficient organoids, the role of plexin-semaphorin signaling in primary tumor formation, invasion and dissemination warrants further investigation and will be the subjects of future studies of our lab. However, we believe that this is outside of the scope of this manuscript that focuses on host-organ derived constraints to colonization.

Referee #3 (Remarks to the Author):

The study by Costanza Borrelli and colleagues seeks to explore the host determinants that modulate the seeding and outgrowth of metastatic cells in the liver. To address this, the authors have developed a novel and elegant *in vivo* approach to systematically interrogate hepatocyte intrinsic factors that promote the establishment of metastases. The screening approach used is based on a CRISPR-activation system that achieves specific gene activation in hepatocytes. The focused approach undertaken is reasonable and targets a wide range of factors (ligands, receptors, and relevant pathways) for their role in metastasis. Authors posit that metastatic outgrowth is contingent on interactions between hepatocytes and disseminating tumor cells, which they infer using a clever *in vivo* niche-labeling system that enables to distinguish between (niche) perturbations that are proximal and distal to metastases.

To prioritize candidates for validation, authors perform spatial transcriptomics of liver CRC mets and narrow down the list to 22 factors involved in tumor-hepatocyte interaction, and further filter the list according to mutated factors in liver mets. They then pursue orthogonal *in vitro* validation of sgRNAs of top-ranked hits on hepatocytes co-cultured with dissociated CRC organoids. Overexpression of PlexinB2 by sg-activation or treatment with recombinant protein, consistently promotes metastatic proliferation of organoids through interaction with class 4 semaphorins. This is validated *in vivo* with the AKPS model and KPC cells.

Collectively, the authors have developed an interesting approach that may be leveraged to investigate the biological relevance of host factors in metastatic colonization. This reviewer congratulates the authors on pioneering such a methodological advance which will certainly enable further studies. However, the mechanistic insights related to the biology of metastatic seeding, human relevance, and generalizability of their findings are underdeveloped. I hope the following comments will help authors improve their manuscript.

Major comments:

1. Given the novelty and broad-use potential of the *in vivo* screening system, additional details should be included. Specifically, authors should validate the “inferred proximity” measurement (as now inferred from flow cytometric analysis of dissociated livers) by providing histological/imaging data that directly shows a differential spatial distribution (proximal vs distal from grafted metastasis) of hepatocytes transduced with seeding-promoting vs seeding-suppressing gRNAs (top hits). To do so, authors could sub-clone gRNAs targeting “seeding-promoting” host factors into a modified version of the initial transposon vector in which the GFP reporter is substituted by a different fluorophore (eg BFP). The prediction would be that BFP+ hepatocytes (expressing “seeding promoting” hits: *Plxnb2*, *Nrp2*, *Ncam1*, *Nenf*, *Egf*, *Psen1*, *Gpc3*) would be more proximal to metastases than GFP+ hepatocytes (expressing “seeding-suppressive” hits: *Saa*, *Ltb*, *Serpina1a*, *App*, *Fn1*, *Cp*) in *AlbCre*;*dCas9*-SPH but not *NonCre* mice. The % of BFP/GFP double-positive hepatocytes will also be informative to support a low MOI in the pooled screening system.

We thank the reviewer for the suggestion, and agree that a “visualization” of the screening technology would be beneficial for the potential users. However, it is technically challenging to obtain an imaging-based quantification of the differential spatial distribution of seeding-suppressing and seeding-promoting perturbations, as outlined below.

Following the reviewer’s suggestion, we have cloned individual sgRNAs against *Plxnb2*, *Psen1* and *Nrp2* into transposon vectors expressing GFP, and *Saa1*, *Ltb* and *App* into transposon vectors expressing

BFP, and injected them into *AlbCre;dCas9SPH* mice, followed by intrasplenic injection of AKPS^{sLPmCherry} organoids. 2 weeks later we analyzed endogenous fluorescence, however, given the sparsity of targeted hepatocytes (0.5%), we found it extremely challenging to find a sectioning plane that captures targeted hepatocytes in the proximity of a metastasis. This is exemplified below, as two sectioning planes from the same metastasis have 3 or 0 metastasis-neighboring perturbations, respectively. Moreover, the BFP signal was extremely difficult to distinguish from background liver autofluorescence.

Consecutive liver sections of mice injected with a pool of seeding promoting sgRNAs (marked by GFP), and a pool of seeding-suppressing sgRNAs (marked by BFP), and subsequently subjected to intrasplenic injection of AKPS organoids.

As hepatocyte-tumor interactions happen in the 3D space, and imaging is only in 2D, we believe that, at the chosen transposon concentration, a FACS-based readout is more appropriate for our screen, while an imaging-based readout is not high-throughput enough. Indeed, we chose the minimal concentration of transposon necessary to achieve a 500X coverage for a library of 997 sgRNAs, which corresponds to 0.5 Mio targeted hepatocytes. In the adult murine liver, which contains 100-150 Mio hepatocytes⁴⁴, this can be achieved with a 0.5% transfection rate (0.5-0.75 Mio targeted hepatocytes). While such a low transfection rate ensures $MOI < 1$, it also makes it difficult to capture interactions by imaging.

Regarding the question of the MOI, we now include experimental evidence of $MOI < 1$. During the establishment of the method, we conducted an experiment to formally test both the stability of the transposon insertion, and to assess whether the concentration of transposon needed to achieve a library coverage of 500X per mouse was also ensuring $MOI < 1$. To do so, we generated transposon vectors encoding differently tagged versions of the fumarylacetoacetate hydrolase (*Fah*) enzyme (PT4-CMV-*Fah*-T2A-GFP and PT4-CMV-*Fah*-myc), and injected them in mice deficient for the essential enzyme fumarylacetoacetate hydrolase (*Fah*), which can only be maintained viable by the continuous administration of nitisinone (NTBC). We quantified myc⁺ and GFP⁺ hepatocytes upon co-injection of SB100X and an equimolar mix of transposon vectors, and could confirm that the dosage was resulting in $MOI < 1$.

Left, number of single or double positive hepatocytes per field of view (FOV). Results pooled from 2 mice (6 FOVs per mouse quantified). Right, Representative fluorescent micrograph of a murine liver upon hydrodynamic tail vein co-injection of SB100X (15 μg) and transposon plasmids (PT4-CMV-Fah-T2A-GFP in green and PT4-CMV-Fah-myc in magenta, 150 μg).

NTBC withdrawal induced liver repopulation by Fah-competent hepatocytes, and sustained hepatic function for up to 2 months. In the absence of SB100X, transient episomal expression of PT4-CMV-Fah-T2A-GFP was insufficient to rescue Fah^{-/-} mice. We concluded that HTVI of SB100X and transposon plasmids can therefore be used for dose-dependent integration and stable functional expression of exogenous DNA. Indeed, contrary to Cas9-mediated editing, which relies on a single genomic event, CRISPR-a requires stable expression of sgRNAs to maintain gene activation.

Mouse weight in percent of initial weight upon injection of PT4-CMV-Fah-T2A-GFP and nitisinone (NTBC) withdrawal. Each line represents a mouse. Asterisks indicate humane endpoints.

While these results contributed to the establishment and refinement of the methods, we did not include them in the manuscript for space limitations. We now added the quantification of double and single positive hepatocytes indicating MOI < 1 in **Extended Data Fig 1e**.

2. Authors should assess the generalizability of their findings in additional lines and primary tumors of origin, and validate the role of Plexin B2 as a gatekeeper of liver metastasis in models beyond intrasplenic injection (eg. GEMMs or at least orthotopic transplantation-based) that better recapitulate the natural course of progression from an existing primary tumor.

We thank the reviewer for the comment and have now conducted several experiments to address this point. We include new results showing that *Plxnb2*-overexpression in hepatocytes increases the grafting of a syngeneic melanoma line injected intrasplenically.

Figure 2 | Hepatocyte-derived Plexin B2 is required for liver colonization. **a**, Schematics of experimental workflow. AAV8-U6-sgPlxn2OE-EF1a-EGFP injection induces dCas9-SPH-mediated overexpression of *Plxn2* in hepatocytes and is followed by intrasplenic injection of AKPS organoids. **b-d**, Metastatic foci per liver section 2 weeks after injection of AKPS organoids, KPC or D4M-3A cells in *Plxn2* OE and control mice. Average of 2 non-consecutive sections per mouse. Results pooled from two independent experiments.

We thank the reviewer for the comment and have now conducted several experiments to address this point. We established a model of orthotopic CRC based on colonoscopy-guided submucosal injection of organoids¹, and tested the role of hepatocyte-derived Plexin B2 for the seeding of spontaneous liver metastases from primary colorectal tumors. We could validate the seeding promoting effect of Plexin B2 on spontaneous metastases from AKPS tumors. Moreover, to test the requirement of Plexin B2 for spontaneous liver seeding, we generated primary colon tumor with APTAK organoids, which exhibit an elevated metastatic potential², followed by hepatocyte-specific ablation of Plexin B2 in *Plxn2*^{fllox/fllox} mice. In the absence of hepatocyte-derived Plexin B2, APTAK primary tumor growth was unaltered, however liver colonization was abrogated. We therefore think that the observed phenotype is of general relevance (also for CRC metastases with different genetic makeup as well as other cancers that frequently metastasize to the liver) and independent of the site of inoculation. The relative section and figures in the revised manuscript are copied below for the reviewer's convenience.

*We further assessed the influence of hepatocyte-derived Plexin B2 on liver seeding of spontaneous metastases from colorectal tumors generated by colonoscopy-guided submucosal injection of organoids³ (Extended Data Fig 5a). Strikingly, 86% of the Plxn2 OE mice developed spontaneous liver metastases 8 weeks after orthotopic AKPS organoid inoculation, while only 29% of control littermates did (Fig. 2g, Extended Data Fig 5b, c). Conversely, Plxn2 deletion significantly decreased incidence and numbers of spontaneous liver metastases of *Apc*^{fllox/fllox}; *Tp53*^{fllox/fllox}; *Tgfr2*^{fllox/fllox}; *Kras*^{G12D}; *Akt1*^{myristoylated} (APTAK) organoids, which exhibit a high metastatic potential² (Fig. 2h, Extended Data Fig 5d-f).*

Figure 2 | Hepatocyte-derived Plexin B2 is required for liver colonization. **g**, Incidence of spontaneous liver metastases in *Plxn2* OE and control mice bearing primary AKPS tumors (Fisher's exact test, $n = 7$ mice per group). **h**, Incidence of spontaneous liver metastases in *Plxn2* KO and control mice bearing primary APTAK tumors ($n = 7$ mice per group).

Extended Data Fig. 5 | Plexin B2 is required for spontaneous metastatic seeding. **a**, Representative micrograph of colonoscopy-guided submucosal injection of CRC organoids. Arrowhead indicates the site of injection (n = 7 mice per group). **b,c**, Representative H&E section and quantification of spontaneous APTAK metastases in *AlbCre; dCas9SPH* mice injected with *sgPlxnB2 OE* or *sgNT*. **d**, Representative micrographs of primary APTAK tumors (bottom) and liver metastases in *PlxnB2^{wt/wt}* and *PlxnB2^{flox/flox}* mice injected with *AAV8-AlbCre-moxGFP*. **e, f**, Representative H&E section and quantification of spontaneous APTAK metastases in *PlxnB2^{wt/wt}* and *PlxnB2^{flox/flox}* mice injected with *AAV8-AlbCre-moxGFP* (n = 7 mice per group).

3. The mechanism of *PlxnB2* upregulation is unclear and should be explored further. This is important to support the physiological relevance of their approach. Specifically, authors should show the basal expression of *PlxnB2* in unperturbed hepatocytes and test whether *PlxnB2* is upregulated in response to tumor-secreted factors (prior and after seeding?), by direct tumor-hepatocyte interactions, and/or via other host/liver-related traits linked with metastasis burden and/or susceptibility, to clarify whether this is a tumor-induced or host-induced mechanism.

We thank the reviewer for the comment and now address this point experimentally. We found peritumoral upregulation of Plexin B2 protein levels in murine livers harboring AKPS metastases by immunofluorescence. Interestingly, *PlxnB2* upregulation was only occurring in hepatocytes around the lesions. This indicates a local response to the presence of the tumor. Indeed, *PlxnB2* was not found upregulated in livers of mice harboring primary AKPS colorectal tumors, as compared to sham controls. This indicates that *PlxnB2* is not upregulated in response to tumor secreted factors and systemic effects. The observed local upregulation of Plexin B2 led us to speculate that it might be involved in a regenerative response. Indeed, *PlxnB2* KO mice exhibit impaired liver vascular regeneration after partial hepatectomy⁴⁵. Moreover, Plexin-semaphorin signaling was implicated in epithelial repair⁴⁶. However, in depth characterization of the role of Plexin B2 in liver regeneration is outstanding and will be the focus of further studies.

We copy below the relative result section and figures.

*At steady state, Plexin B2 is widely expressed by hepatocytes - with higher expression in portal areas (Extended Data Fig 5g, h) - as well as endothelial cells of the hepatic sinusoids and portal veins³⁷. Interestingly, while Plexin B2 immunoreactivity was higher in peritumoral hepatocytes (Extended Data Fig. 5i,j), *PlxnB2* expression was unaltered in livers of mice bearing AKPS colon tumors, suggesting that *PlxnB2* is not upregulated by primary tumor secreted factors nor systemic effects, but rather due to a local response to the presence of metastases (Extended Data Fig. 5k). Interestingly, ablation of *PlxnB2* 5 days after intrasplenic AKPS organoid injection delayed but did not prevent metastasis formation, indicating*

that the presence of Plexin B2 on peritumoral hepatocytes is only required for metastatic seeding, but not to sustain growth (Extended Data Fig. 5l,m).

Extended Data Fig. 5 | g, Multiplexed *in situ* hybridization (Molecular Cartography) of *Plxnb2* as well as portal and central hepatocyte markers in WT mouse liver. **h**, Quantification of *Plxnb2* mRNA abundance in portal and central areas. Two-tailed unpaired Wilcoxon test. **i**, Representative fluorescent micrograph of Plexin B2 expression in AKPS liver metastasis. Scale bar, 100 μm . **j**, Plexin B2 staining intensity in peritumoral and distal areas. Dots represent individual metastases in $n=3$ mice. Two-tailed unpaired T-test. **k**, Differential gene expression analysis in livers of primary colon tumor-bearing vs naive mice. $n = 3$.

4. The proposed effects by which *Plxnb2* OE induces mesenchymal-to-epithelial transition (and its blockade has an inhibitory effect) relies on illustrative panels of established markers that should include more extensive immunolabeling and quantification, and be accompanied by OE/KD and epistasis experiments to support the claims that *Plxnb2* modulates metastasis via regulation of EMT/MET states, and that this is mediated by *Rac1* signaling and actin cytoskeleton remodeling. It would also be relevant to assess whether the *Plxnb2*-Sema axis is required to maintain the growth of established metastases after epithelial traits have been acquired. Expanding the mechanistic depth would enhance the significance of the study.

We conducted several experiments to deepen our mechanistic understanding of Plexin-semaphorin signaling in metastases. These experiments revealed that, downstream of semaphorins, the transcription factor Klf4 suppresses mesenchymal traits and promotes reepithelialization of metastases. Indeed, there is increasing evidence that, while primary epithelial tumors undergo dedifferentiation, secondary tumors in the liver are highly differentiated and epithelial^{4,5}. The new results are listed below accompanied by the relative figure panels for the reviewer's convenience. We think that cumulatively, these results support the notion that Plexin B2-induced re-acquisition of epithelial traits is necessary for liver colonization, as in absence of Plexin B2 hepatic seeding is suppressed, and the few remaining lesions exhibit mesenchymal traits.

- 1) Single cell RNA sequencing (scRNASeq) revealed that AKPS organoids exhibit mutually exclusive expression of EMT and MET signatures (obtained from Gene Ontology, GO) (**Extended Data Fig 8a**). Due to the sparsity of scRNASeq, signatures of multiple genes are more robust than plotting of individual genes, yet it is important to note that EMT is controlled by a dynamic interplay of transcription factors, whose transcripts are challenging to capture by scRNASeq, and that are extensively post-translationally modified. The non-uniform EMT/MET state of AKPS organoids in culture corroborates our stainings showing a mix of E-cadherin^{high}Zeb1^{low} and E-cadherin^{low}Zeb1^{high} cells within the same organoid (**Extended Data Fig 8b**). Of note, we observe a significant upregulation of Zeb1 immunoreactivity in AKPS organoids, as compared to WT organoids, suggesting that transformation induces the acquisition of mesenchymal traits. Indeed McFalín-Figueroa *et al* showed that KRAS signaling drives the exit from an epithelial state⁸. As cell contacts are retained, as evidenced by E-cadherin staining, these data are consistent with a partial EMT state of AKPS organoids prior to inoculation. Indeed, partial EMT cells have been shown to upregulate EMT transcription factors but to retain proliferation, cell-cell contacts, apical-basal polarity and collective migration⁹. Moreover, partial EMT cells have been shown to exhibit a higher metastatic potential than fully mesenchymal cells^{9–12}. Whether cells that disseminated from primary CRCs exhibit a similar partial EMT state warrants further investigation, however it is outside of the scope of this study.

Extended Data Fig. 8 | **a**, Expression density of EMT and MET gene signatures in AKPS organoids profiled by scRNASeq. **b**, Representative fluorescent micrographs and quantification of E-cadherin and Zeb1 immunoreactivity in WT colon and AKPS organoids. Arrow shows E-cadherin^{high}Zeb1^{low} cell, arrowhead indicates E-cadherin^{low}Zeb1^{high}. Two-tailed unpaired T-test.

- 2) We further expanded our characterization of the EMT state of metastases growing in *Plxnb2*-overexpressing, -deficient and WT liver by staining for transcription factors that have been implicated in promoting epithelial traits in tumors by suppressing EMT, namely Elf3 and Grhl2¹³⁻¹⁷. We found expression of these factors in metastases grown in *Plxnb2*-WT and -overexpressing livers, but absence thereof in the few lesions growing in *Plxnb2*-deficient livers (**Extended Data Fig 8c**). These stainings provide further evidence that, in absence of Plexin B2, metastases do not undergo epithelialization and therefore fail to adapt to the liver environment. Promotion of epithelial traits by Plexin-semaphorin interactions is further consistent with elevated Epcam immunoreactivity in AKPS metastases growing in *Plxnb2*-OE livers (**Extended Data Fig 7j**). These stainings provide further evidence that Plexin B2 levels in hepatocytes are inversely correlated with mesenchymal traits in mets, suggesting that Plexin B2 promotes epithelialization of metastases.

Extended Data Fig. 8 | c, Representative fluorescent micrograph and quantification of AKPS metastases in *Plxnb2* OE, *Plxnb2* KO and control livers. GRHL2, ELF3 and Zeb1 shown in red, F-actin shown in gray, DAPI in blue used as nuclear counterstain. Scale bar, 50 μ m. Inset 1 shows GRHL2⁺ nuclei in *Plxnb2* OE livers. Inset 2 shows Zeb1⁺ nuclei in *Plxnb2* KO livers. Scale bar, 50 μ m.

Extended Data Fig. 7 | j, Representative fluorescent micrograph and quantification of Epcam immunoreactivity in AKPS metastases in *Plxnb2* OE and control livers. Scale bar, 20 μ m. Dots represent individual metastases in $n = 2$ mice.

- 3) To characterize the EMT profile of metastases *in vivo*, we performed combined single nucleus RNA and ATAC sequencing data from metastatic livers overexpressing *Plxnb2* or controls. We reasoned that while transcription factor levels might be technically challenging to detect and quantify by mRNA sequencing, their chromatin occupancy would be a more faithful proxy of their activity. We used the transcriptome space to annotate our dataset, and then computed differentially open peaks between tumor cells in *Plxnb2*-OE vs control livers (**Extended Data Fig 7d,e**). With respect to

controls, metastases growing in *Plxnb2* OE livers exhibited increased activity of several members of the SP/KLF family of zinc finger transcription factors¹⁸ (Fig. 3g). In line with these results, we detected increased nuclear levels of Krüppel-like factor 4 (Klf4) in AKPS metastases growing in *Plxnb2* OE livers, while it was absent from lesions in *Plxnb2* KO livers (Fig. 3h). Klf4 is expressed by differentiated cells of the colonic epithelium, but its expression is lost in CRC, where it acts as tumor suppressor by inhibiting epithelial-mesenchymal transition (EMT)^{19–26}. Strikingly, Klf4 is absent in lesions growing in *Plxnb2*-KO livers, and this coincides with retained Zeb1 nuclear expression. We now provide additional stainings that indicate nuclear Zeb1 expression in AKPS metastases growing in *Plxnb2*-KO livers (Fig. 3h and Extended Data Fig 8c, see above)

Extended Data Fig. 7 | Multiomic characterization of AKPS metastases in *Plxnb2* OE and control livers. d. UMAP of RNA profile of nuclei isolated from *Plxnb2* OE and control livers harboring AKPS metastases. Colors indicate cell types. **e.** Marker gene expression of distinct cell types in dataset.

Figure 3 | Plexin B2 induces epithelialization of liver metastases. f. UMAP of chromatin profile of epithelial cells in AKPS metastases in *Plxnb2* OE and control livers profiled by snATACseq. **g.** Enriched TF binding motifs in differentially open peaks. **h, i.** Representative fluorescent micrograph and quantification of AKPS metastases in *Plxnb2* OE, *Plxnb2* KO and control livers. Klf4 shown in red, E-cadherin in gray, DAPI in blue used as nuclear counterstain. Scale bar, 50 µm. Arrows and inset shows Klf4⁺ or Zeb1⁺ nuclei. Asterisk indicates background. Scale bar, 10 µm. **h, i** Ordinary one-way ANOVA, comparing each treatment group to control. Tukey's correction for multiple testing. Dots represent individual nuclei in $n = 3$ mice. Barplots indicate mean \pm SD.

- 4) We treated two-dimensional cultures of AKPS organoids with a small-molecule inhibitor of KLF4 (WX-43), which specifically abrogates the interaction between PRMT5 and KLF4, thereby enhancing KLF4 degradation²⁷. Treatment induced acquisition of a mesenchymal morphology, decreased colony size, F-actin cytoskeleton and proliferation (Extended Data Fig 7k,l). These

results strongly suggest that Klf4 is required to maintain suppression of mesenchymal programs in AKPS organoids.

Extended Data Fig. 7 | k,l Representative brightfield and fluorescent micrograph and quantification of two-dimensional AKPS cultures treated with the Klf4 inhibitor WX2-43²⁷. F-actin shown in gray, EdU in red, DAPI in blue. Scale bar, 10 μ m. a,c,h,i,j,l Two-tailed unpaired T-test, b,g Ordinary one-way ANOVA, comparing each treatment group to vehicle control. Tukey's correction for multiple testing. a,b,g,h,i,l Dots represent individual wells.

- 5) To test whether signaling via semaphorins is required for MET induction, we generated quadruple KO AKPS organoids (AKPS^{qKO}) lacking *semaphorin 4A, C, D and G* with enhanced enAsCas12a technology. Quadruple editing was validated by NGS and immunofluorescence in multiple clonal lines.

Extended Data Fig. 10 | Class IV semaphorins on tumor cells are required for metastatic seeding. e, Representative fluorescent micrograph of Sema4A, C, D and G immunoreactivity in AKPS^{Sema4KO} and AKPS^{Sema4WT} organoids. Scale bar, 20 μ m.

Sema4a	WT	AGCCAAGGGAACCAAGTGAATGCTCTCTGCTGCTCTTCAAGGACAGTTGGCGGACCAAGGAGCTCAGCAGTCTGTCCTCTCTCTCACGGACATTGAGCGAGTCTTTAAAGGGAAGTACAAGGAGCTGAAACAAGGAGACCTCCCG	PREPQMSSLAVFRQVGGTRSSAVCAFLSDIERVFKGKYKELNKETSR
	Sema4KO clone 3	23% of reads: AGCCAAGGGAACCAAGTGAATGCTCTCTGCTGCTCTTCAAGGACAGTTGGCGGACCAAGGAGCTCAGCAGTCTTAAAGGGAAGTACAAGGAGCTGAAACAAGGAGACCTCCCGCTGGACCACTACCAGGGCTCAGAGGTCAAGCCCGAG 77% of reads: AGCCAAGGGAACCAAGTGAATGCTCTCTGCTGCTCTTCAAGGACAGTTGGCGGACCAAGGAGCTCAGCAGTCTGTGCCCTCTCTGAGCTGAGCAAGGAGACCTCCCGCTGGACCACTACCAGGGCTCAGAGGTCAAGCCCGAGGCGA	PREPQMSSLAVFRQVGGTRSSAVFKGKYKELNKETSRWITTYRGSEVSPX PREPQMSSLAVFRQVGGTRSSAVCAFLSSTRRPPAGPLTGAQRSARCX
Sema4c	WT	TTCTCTGTCCCTGGTATGGTGGGAGCAAGTAGGCACCCAACTGACTTCCAAGTCTCTTTAACAGGGGCGATATGGACCTGCTCGAGTTGTGAGTACCAGTTGGAACAGATCCAGCAAGTGTGGAGGTCCTCAAGGAGTA	PLFPGMVGASRHPITLPRSFNRGDMDSAVCEYQLEIQVFEFGPYKEX
	Sema4KO clone 3	37% of reads: TTCTCTGTCCCTGGTATGGTGGGAGCAAGTAGGCACCCAACTGACTTCCAAGTCTCTTTGTGAGTACCAGTTGGAACAGATCCAGCAAGTGTGGAGGTCCTCAAGGAGTA 63% of reads: TTCTCTGTCCCTGGTATGGTGGGAGCAAGTAGGCACCCAACTGACTTCCAAGTCTCTTTAACAGGGGCGATATGGTCTGAGTGTGAGTACCAGTTGGAACAGATCCAGCAAGTGTGGAGGTCCTCAAGGAGTACAGTG	PLFPGMVGASRHPITLPRSFVYTSWNRSSKCLRVPTRSTVSKPFRSGPAX PLFPGMVGASRHPITLPRSFNRGDMVCSLVPVGTDPASVGSLSGQVQX
Sema4d	WT	AAGGGTCTTTGATGGTGCCTGTAATGGGGTGTTCCTTCCAGACGGAATGCCTAACTACATTGAGTACTACAGCCACTAAGCAGCACTTCCCTCTATGTGTGGGACCAATGGCTCCAGCCCACTGTGACCACTGGTAAGA	RVLWVPVINGVFLPQTECLNYIRVLQPLSSTLSYCGTNAFQPTCDHLVR
	Sema4KO clone 3	54% of reads: AAGGGTCTTTGATGGTGCCTGTAATGGGGTGTTCCTTCCAGACGGAATGCCTAACTACAGTACTACAGCCACTAAGCAGCACTTCCCTCTATGTGTGGGACCAATGGCTCCAGCCCACTGTGACCACTGGTAAGCACCACAA 46% of reads: AAGGGTCTTTGATGGTGCCTGTAATGGGGTGTTCCTTCCAGACGGAATGCCTAACTACTACAGCCACTAAGCAGCACTTCCCTCTATGTGTGGGACCAATGGCTCCAGCCCACTGTGACCACTGGTAAGCACCACATA	RVLWVPVINGVFLPQTECLNEYSHAALPSCMGVPMRSPVPTTWDHX RVLWVPVINGVFLPQTECLNYSHAALPSCMGVPMRSPVPTTWDHNI
Sema4g	WT	TGATAAGCTAGACTAATGGAGTGGCTTGTCTCTGCGTGTGATGAGTCTACTGCTCCCTCTGCTGTAGACCGAGTGTTCACACAGCTGCGATTTCTACAGCGCTCAATGCCACCACTTCTATGATGGGACCAAGCCCT	DKLDWRWLVSGLMSSITASLCLTECFNH/RFLQRLNATHFYACGTHAX
	Sema4KO clone 3	22% of reads: TGATAAGCTAGACTAATGGAGTGGCTTGTCTCTGCGTGTGATGAGTCTACTGCTCCCTCTGCTGTAGACCGAGTGTTCACACAGCTGCGATTTCTACAGCGCTCAATGCCACCACTTCTATGATGGGACCAAGCCCT 78% of reads: TGATAAGCTAGACTAATGGAGTGGCTTGTCTCTGCGTGTGATGAGTCTACTGCTCCCTCTGCTGTAGACCGAGTGTTCACACAGCTGCGATTTCTATGATGGGACCAAGCCCTTCCAGCCCTCTGCTGAGCTATTGTGAG	DKLDWRWLVSGLMSSITASLCLTECFNH/RFLPAPLCSYCEYACPL DKLDWRWLVSGLMSSITASLCLTECFNH/RFLCMWDRLPAPLCSYCE

Supplementary Table 2 | Indel analysis of AKPS^{Sema4KO} (clonal line 3) showing out of frame mutations in all 4 targeted semaphorin genes. Green rows show WT base and amino acid sequence. Mutated parts shown in red.

While AKPS^{KO} organoids show unaltered proliferation *in vitro*, as compared to control organoids, they exhibit significant grafting impairment *in vivo*, as they fail to form primary tumors and metastases when inoculated orthotopically.

Extended Data Fig. 10 | Class IV semaphorins on tumor cells are required for metastatic seeding. **f**, Edu⁺/Edu⁻ cells in two-dimensional AKPS^{Sema4KO} and AKPS^{Sema4WT} organoids. **g**, Representative images of livers (top) and primary colon tumors (bottom) in mice orthotopically injected with AKPS^{Sema4KO} or AKPS^{Sema4WT} organoids. **h**, Incidence of spontaneous liver metastases in mice orthotopically injected with AKPS^{Sema4KO} or AKPS^{Sema4WT} organoids (n = 8).

To assay liver colonization, we injected AKPS^{Sema4KO} organoids intrasplenically, and found significantly decreased metastatic burden compared to control organoids (**Fig. 4e,f**). Strikingly, Klf4 immunoreactivity is lost in AKPS^{Sema4KO} metastases, which further exhibit diminished Epcam and E-cadherin expression and high Zeb1 levels (**Fig. 4g**). Loss of semaphorins in AKPS metastases thus phenocopies loss of Plexin B2 on hepatocytes, supporting the notion that Plexin-semaphorin interactions are required for metastatic seeding (**Fig. 4h**).

Figure 4 | Plexin B2 binds semaphorins on tumor cells. **e**, Metastatic foci per liver section 2 weeks post intrasplenic injection of AKPS^{SemaWT} or AKPS^{Sema4KO} organoids. Dots represent individual mice (n=3). **f**, Representative H&E staining of AKPS^{SemaWT} or AKPS^{Sema4KO} metastases. Scale bar, 100 μ m. **g**, Representative fluorescent micrograph and quantification of Epcam (in gray) and Klf4 (in red) immunoreactivity in AKPS^{SemaWT} and AKPS^{Sema4KO} metastases. Scale bar, 20 μ m. Arrow and inset show Klf4⁺ nuclei. Scale bar, 10 μ m. **h**, Representative fluorescent micrograph and quantification of E-cadherin (in gray) and Zeb1 (in red) immunoreactivity in AKPS^{SemaWT} and AKPS^{Sema4KO} metastases. Scale bar, 20 μ m. Inset 1 shows E-cadherin⁺Zeb1⁺ nuclei. Inset 2 shows E-cadherin⁺Zeb1⁻ nuclei. Scale bar, 10 μ m. **i**, Plexin B2 on hepatocytes binds class IV semaphorins on DTCs, inducing Klf4 upregulation and Zeb1 downregulation, which results in epithelialization and outgrowth of metastases. a,b,c Dots indicate individual wells. Ordinary one-way ANOVA, comparing each treatment group to control. Tukey's correction for multiple testing. d,e,g,h Two-tailed unpaired t-test. Barplots indicate mean \pm SD.

6) We further identified a dependency on Rac1 signaling both for the in vitro seeding promoting effects of rhPlexin B2 on PDOs, and the in vivo seeding promoting effect of rmPlexin B2 on AKPS organoids.

Figure 4 | Plexin B2 binds semaphorins on tumor cells. **a**, Metastatic foci per liver section 2 weeks post intrasplenic injection of AKPS organoids treated *ex vivo* with rmPlexinB2 and/or the Rac1 inhibitor NSC323766. Dots represent individual mice (n=3). **c**, CFU per well of PDOs cultured with 2 μ g/mL rhPlexin B2 and NSC 23766.

7) Finally, to test whether Plexin B2 is required to sustain metastatic tumor growth after seeding, we conditionally ablated *Plxnb2* 5 days after intrasplenic AKPS inoculation and monitored tumor growth by IVIS. Interestingly, liver colonization was delayed but not impaired by *Plxnb2* ablation after tumor establishment, suggesting that the presence of Plexin B2 is not required to sustain tumor growth.

Extended Data Fig. 5 | Plexin B2 is required for spontaneous metastatic seeding. **I**, BLI signal over time in *Plxn2* KO and control mice injected with AAV8-*AlbCre-moxGFP* (AAV) 5 days post intrasplenic injection of AKPS organoids and (n = 3). Two-tailed unpaired T-test. **m**, Representative H&E section of AKPS metastases in livers of *Plxn2* KO and control mice injected with AAV8-*AlbCre-moxGFP* (AAV) 5 days post intrasplenic injection.

Cumulatively, the new evidence provided strongly indicates that Plexin-semaphorin signaling induces epithelialization of CRC metastases by inducing Klf4 and suppressing Zeb1, and that this is a rate-limiting step for liver seeding and colonization.

5. The human relevance of the findings should be strengthened by providing additional insights on the regulation of *Plxn2* levels and/or signaling in patients with differential liver metastasis burden, and/or assessing whether *Plxn2* levels associate with patient prognosis or metastasis relapse in human datasets – to complement the data in Ext Data Fig 6. Authors could exploit available single cell analyses on human liver metastasis samples to confirm the validity of their data in patients, and in relation to established early markers of metastasis (eg *Emp1*, etc).

We thank the reviewer for the comment and we now conducted a deeper analysis of semaphorin gene mutational status and expression in CRC patients.

*SEMA4A, C, D and G are detected in human colon and CRC, and, in two published scRNASeq datasets of human CRC³⁸. Expression of the semaphorin genes is significantly upregulated in high-relapse (HR) epithelial cells³⁹ and in intrinsic consensus molecular subtype 3 (iCMS3) cells⁴⁰, but not in *Lgr5^{high}* cells, and it coincides with a *MET* signature (Extended Data Fig. 9c-e). Semaphorin expression is further unaltered in liver metastases, as compared to matched primary tumors in two independent scRNASeq datasets of metastatic CRC^{41,42} (Extended Data Fig. 9f). Copy number variation (CNV) analysis in the COAD dataset⁴³ further revealed that *SEMA4A*, *SEMA4C* and *SEMA4D* are commonly found amplified, while *SEMA4G* is often deleted in CRC patients (Extended Data Fig. 9g). Moreover, in a large cohort of CRC patients, increased expression of *SEMA4A*, *SEMA4C* and *SEMA4D*, but not *SEMA4G*, is associated with reduced recurrence-free survival (Extended Data Fig. 9h).*

Extended Data Fig. 9 | c, Immunohistochemistry for SEMA4A, C, D, and G in healthy and human colon obtained from the Human Protein Atlas (<https://www.proteinatlas.org/>). **d**, Left, Averaged expression of *Sema4a*, *4c*, *4d*, *4g* in scRNASeq datasets of human primary CRC (only epithelial cells). Cells grouped as high-relapse cells (HRCs), Lgr5 cells or others. Right, averaged expression of *Sema4a*, *4c*, *4d*, *4g* projected on epithelial cells of KUL dataset together with Wnt ON signature or MET signature. **e**, Averaged expression of *Sema4a*, *4c*, *4d*, *4g* in scRNASeq datasets of human primary CRC (only epithelial cells). Cells grouped as iCMS2 or iCMS3. **f**, Averaged expression of *Sema4a*, *4c*, *4d*, *4g* in epithelial cells from CRC liver metastasis or matched primaries in two independent datasets. **g**, CNV analysis *SEMA4A*, *SEMA4C*, *SEMA4D* and *SEMA4G* in COAD dataset. $n = 290$ patients. **h**, Kaplan-Meier analysis (<http://kmpplot.com>) of the recurrence free survival (RFS) of CRC patients stratified by *SEMA4A*, *SEMA4C*, *SEMA4D* and *SEMA4G* expression. $n = 1211$ patients. **d,e,f** Two-tailed unpaired Wilcoxon test.

The upregulation of class IV semaphorins by HRCs, a distinct cell state found in primary CRC that were recently implicated in metastatic dissemination and recurrence¹⁴, is in line with the proposed role of plexin-semaphorin in mediating liver colonization. Moreover, plexin-semaphorin interactions were recently identified as mediating formation of homotypic and heterotypic clusters of circulating cancer cells⁹⁴. In light of the colon grafting impairment of semaphorin-deficient organoids, the role of plexin-semaphorin signaling in primary tumor formation, invasion and dissemination warrants further investigation and will be the subjects of future studies of our lab. However, we believe that this is outside of the scope of this manuscript that focuses on host-organ derived constraints to colonization.

6. According to the literature, Plexin B2 is strongly expressed in the endothelial cells of the hepatic sinusoids and portal veins (Zielonka et al. Exp Cell Res 2010). Is this cell type involved in the host-derived MET-liver seeding mechanism proposed in this work, or is PlexinB2 activation in hepatocytes specifically a necessary step for liver metastasis in the cancer types investigated?

We thank the reviewer for the comment, however, according to our data, PlexinB2 on endothelial cells is not involved in metastasis seeding. Indeed, in both in *Albumin-Cre;Cas9* or *dCas9-SPH* mice injected with AAV8 harboring sgRNAs targeting *Plxnb2*, and in *Plxnb2^{fl/fl}* animals injected with AAV8 harboring *Albumin-Cre*, *Plxnb2* overexpression or deletion is restricted to hepatocytes. We have now specified this in the text:

*At steady state, Plexin B2 is widely expressed by hepatocytes - with higher expression in portal areas (Extended Data Fig 5g, h) - as well as endothelial cells of the hepatic sinusoids and portal veins³⁷. However, our experimental design induces hepatocyte-specific *Plxnb2* OE or KO (Extended Data Fig. 4g) and thus implicates DTCs interactions with these cells, and not endothelial cells, as being required for metastatic outgrowth.*

We show in Extended Data Fig 4g Plexin B2 levels in *Plxnb2^{fl/fl}* animals 14 days after injection with AAV8 harboring *Albumin-Cre*, which exhibit complete loss of Plexin B2 in hepatocytes but still retain Plexin B2 around liver veins and sinusoids.

Extended Data Fig. 4 | g, Representative fluorescent micrographs of Plexin B2 immunoreactivity in *Plxnb2^{wt/wt}* and *Plxnb2^{lox/lox}* mice injected with *AAV8-AlbCre-moxGFP*. DAPI used as a nuclear counterstain. Scale bar, 200 μ m. Inset shows residual Plexin B2 on endothelial cells lining the central vein. Scale bar, 100 μ m.

7. Several conclusions are based on visual representations of data/images throughout the paper. A more thorough quantification of staining and phenotypes should be performed throughout to allow for a more systematic evaluation of the claims – eg: Fig 3 h-i, 4c-d, several panels on Ext Fig 4, Ext Data 5, Ext Data 6 etc. Similarly, authors should perform more detailed statistical analyses – ie: Fig 2A, Ext Fig Data 5, etc

Quantifications are now provided for all stainings.

8. The discussion would benefit from a more elaborated conceptual discussion of their findings in relation to the relative contribution of the “seed” versus the “soil” in metastasis formation.

We now included the following paragraph in the discussion:

The data presented herein identify hepatocyte-derived Plexin B2 as required for liver colonization, and its upregulation sufficient to increase metastatic burden in multiple tumor models. Interestingly, Plxnb2 expression is unaltered in the liver of colon tumor-bearing mice, indicating that it is not upregulated by systemic effects inducing the pre-metastatic niche⁴⁷. While Plexin B2 levels are elevated in peritumoral hepatocytes, suggesting a local, possibly regenerative, response, its expression on hepatocytes is not required to sustain metastatic growth. Together with the observed grafting impairment of AKPS^{Sema4KD} and AKPS^{Sema4KO} organoids, these results strictly implicate Plexin-semaphorin interactions in promoting early seeding events occurring upon DTC extravasation in the liver. Interestingly, class IV semaphorins are upregulated by HRCs, a distinct cell state found in primary CRC that were recently implicated in metastatic dissemination and recurrence³⁹. Moreover, plexin-semaphorin interactions were recently identified as mediating formation of homotypic and heterotypic clusters of circulating cancer cells⁴⁸. Hence, the role of plexin-semaphorin signaling in primary tumor formation, invasion and dissemination warrants further investigation.

Minor comments:

A clearer description of experimental design details of the screening technology would be appropriate so this information is more accessible to readers: size (micro, macro), number of nodules per mice at the timepoint(s) analyzed; number of replicates, controls, etc,

We thank the reviewer for the remark and now included more information in the Methods section, as detailed below:

Library cloning: sgRNA sequences for dCas9-SPH-mediated overexpression were retrieved from the Caprano library⁴⁹ and obtained as oPool from Integrated DNA Technologies with Gecko flanking sequences⁵⁰. The 3 sgRNA per target were chosen from Caprano Set A, for a total of 897 sgRNAs targeting 299 genes. 100 safe-targeting sgRNAs⁵¹ were also included.

Library delivery to mouse liver and tumor inoculation: the sgRNA transposon plasmid libraries (150 µg) and SB100X-encoding plasmids (15 µg) were co-injected hydrodynamically into AlbCre;dCas9-SPH mice or littermate controls lacking AlbCre. One week later, dissociated AKPS^{SLP-mCherry} organoids were inoculated by intrasplenic injection, followed by splenectomy. Metastases were allowed to grow for 2 weeks, during which several (10-100) small metastases (0.5-1.5 mm) formed. In total, 7 AlbCre;dCas9-SPH mice and 5 nonCre littermate controls were used for 3 independent screening experiments.

Spatial transcriptomics: it is not clear the number of different hepatic CRC metastases and patient samples the data was generated from. Also, regarding the NicheNet predictions: what is the number of predicted interactions of tumor cells at the metastatic leading edge with metastasis-distal hepatocytes, as a reference? Without a reference or control comparison, it is hard to interpret whether the numbers and overlaps described are higher than what one would find randomly.

We thank the reviewer for the remark and now specify in the Main text and in the Method sections that Visium data was generated from one sample of human CRC liver met (the only we could obtain with a clear and extensive tumor border). We generated two technical replicates from the same sample to increase the number of spots analyzed.

We calculated the number of possible LR interactions of tumor cells with metastasis-distal hepatocytes (503), and found that it is quite similar to the number of predicted interactions with proximal hepatocytes (533). Our NicheNet analysis, however, prioritizes LR at the tumor border by calculating their potential to regulate the differentially expressed genes between metastasis-core vs. center spots (ligand activity analysis). Of the 533 possible LRs between metastasis edge and proximal hepatocytes (which are used as background genes for the regulatory score calculation), 109 are predicted to be active, and of those 22 are among the top-scoring hits of our screen (top and bottom decile). We now clarified this in the text and method section, visualized the predicted LR interactions as circos plot in **Fig 1g**, and added the differential gene expression and ligand activity score to **Extended Data 2e-g**.

Extended Data Fig. 2 | f, Differentially expressed genes between metastasis-edge and metastasis-core spots. Two-tailed unpaired Wilcoxon test. **g**, Top 20 active hepatocyte ligands and their regulatory potential on DEGs at metastatic edge.

Figure 1 | Screening tumor-hepatocyte interactions in a mosaic liver. g, LR interaction analysis from ST data of a human hepatic CRC metastasis. 22 putative seeding-regulating factors expressed by metastasis-proximal hepatocytes (orange) are predicted to interact with LRs on tumor cells at the metastatic leading edge (turquoise).

Figure calling should be done in chronological order to facilitate the reading – eg: Fig 3g is called prior to Fig 3s-b etc.

We amended this in the revised manuscript.

Extended Data Fig. 2: same x-axis for experimental and control panels.

We amended this in the revised manuscript.

While the narrative is easy to follow, there are minor typos in the text that should be corrected.

We apologized for the typos and carefully proofread the revised manuscript.

Referee #4 (Remarks to the Author):

The authors perform an innovative CRISPR activation screen with a focused sgRNA library in hepatocytes *in vivo*, to identify factors that influence metastasis of cancer cells into the liver. They find known and unknown factors influencing, i.e., enhance or decrease, metastases and they decide to focus on Plexin B2. Analysis of human liver tissues with metastases validates Plexin B2 as a potential target as well. The authors perform *in vitro* experiments to demonstrate that metastasised cancer cells are affected via a Plexin B2 stimulation derived from the hepatocytes. Deletion of Plexin B2 in the hepatocytes in mice almost completely abolishes seeding of cancer cells into the liver. Rac1 signalling is proposed to be induced in the cancer cells, via Plexin B2 signalling involving Semaphorins. Plexin B2 signalling seems to be responsible for a mesenchymal to epithelial phenotype observed in the liver metastases.

The idea of using a secreting fluorophore (sLP-mCherry) that is secreted by the cancer cells (colorectal and pancreatic organoids) and combining it with a CRISPR activation approach *in vivo* in the seeded liver is a highly innovative, unique assay identifying factors that enhance or reduce seeding of metastases in the liver. The approach is very valid and innovative, and the quality of the data presented very high. However, in its current form the paper is not acceptable for publication as more experimental data is required to fully support the conclusions drawn by the authors.

The authors use the LSL-dCas9-SPH mice crossed to the Albumin-Cre transgenic animals to tissue specifically induce the CRISPR activation machinery – based on the Suntag technology – in the hepatocytes. The LSL-dCas9-SHP mice have been developed and described for applications in the nervous system (Nature Neuroscience 2018 (PMID: 29335603). However, it was recently reported on the Jackson webpage that these mice had multiple phenotypic issues (spontaneous urination, etc.) when they were received from the authors of that study. The Jackson lab bred these mice onto a mixed B6 x DBA background and that resulted in a normal phenotype. However, further issues with this model were detected as the original insertion site of the dCas9-SPH described in the original publication could not be verified, but a totally different insertion site had been detected. While I am sure that the authors of the current study are aware of these issues, it is mentioned in the Methods part that the mice were kept on a C57BL6/J background. Could the authors please indicate how many generations their strain was backcrossed and which model in terms of insertion sites the authors received from the Jackson lab? This could lead to some skewing of the experiments as for example the expression of the CRISPR activation machinery could be mosaic, suggesting that only a few liver cells do indeed express sufficient dCas9-SHP resulting in mosaic transcriptional activation of target genes despite the presence of a sgRNA. Conversely, the induction of expression of a certain gene could become too high when combined with a specific sgRNA due to the transgene being in a certain genomic region, which leads to very high expression of the CRISPRa components. In addition, it is not clear, whether the combination of the dCas9-SHP and sgRNA could induce some off-targets. This might be a problem as some sgRNAs could cause unwanted side effects on top of their normal role. In summary the authors need to do more work to characterise the dCas9-SPH model described in their paper.

We thank the reviewer for the remark and confirm that we have observed minor phenotypic issues in LSL-dCas9-SHP mice, such as spontaneous urination. We obtained the mice from Jackson in 2021, so according to the Jackson website “It cannot be determined which insertion(s) researchers may have received from JAX before 2022”. These issues were solved when mice were outcrossed to B6J and crossed to the AlbCre strain, which occurred for over 10 generations.

Upon injection of *AAV8-sgPlxn2-EGFP*, we detected uniform upregulation in all mice analyzed, both on the RNA and on the protein level (**Extended Data Fig. 4a,b,c,d,e**). We also observed very strong

ectopic upregulation of glutamine synthetase upon injection of *sgGlul* (**Extended Data Fig. 1a, b**). These results strongly indicate uniform expression of and strong functionality of dCas9SPH. We have now used these mice repeatedly in other projects, including with different Cre-drivers, and are confident of wide SPH expression. While it is still possible that SPH expression is mosaic, reducing the number of hepatocytes that actually overexpress *Plxnb2* (or other targets) when harboring a sgRNA, we believe that this would not impact our conclusions. Indeed, this would indicate that even at lower actual coverage (both SPH and sgRNA present in a cell), our perturbation has significant effects. Most importantly, we orthogonally validated the effect of hepatocyte-derived *Plxnb2* on CRC metastasis *in vitro* and *in vivo* models that are independent of the CRISPR-a system, strengthening the notion that the observed phenotype is not due to off-target effects.

Extended Data Fig. 4 | Overexpression of *Plxnb2* in murine hepatocytes after AAV-mediated sgRNA transduction. a, Percentage of GFP⁺ hepatocytes as assessed by immunofluorescence in *AlbCre;dCas9-SPH* mice injected with AAV8-U6-*sgPlxnb2OE-EF1a-EGFP* (*sgPlxnb2 OE*, n=5) or AAV8-U6-*sgNT-EF1a-EGFP* (*sgNT*, n=7). Scale bar, 20 μ m. **b,** Representative micrograph of DAPI staining (green) and *Plxnb2* mRNA (white) as detected by smFISH. Arrowheads indicate single mRNA molecules. **c,** Quantification of *Plxnb2* mRNA molecules per cell. Two-tailed unpaired t-test. **d,e,f,** Representative fluorescent micrographs and quantification of Plexin B2 immunoreactivity in *AlbCre;dCas9-SPH* mice injected with AAV8-U6-*sgPlxnb2OE-EF1a-EGFP* (*sgPlxnb2 OE*) or AAV8-U6-*sgNT-EF1a-EGFP* (*sgNT*, n=7), and *AlbCre;Cas9* mice injected with AAV8-U6-*sgPlxnb2KO-EF1a-EGFP* (*sgPlxnb2 KO*). DAPI used as a nuclear counterstain. Scale bar, 20 μ m. Two-tailed unpaired t-test.

Extended Data Fig. 1 | a, Glutamine synthetase immunoreactivity in *AlbCre; dCas9-SPH* mice upon co-injection of SB100X and *PT4-U6-sgGlul* or *PT4-U6-sgNT*. Scale bar, 100 μ m. CV, central vein. **b**, Glul gene expression of whole liver extracts, as assessed by qRT-PCR. Two-tailed unpaired t-test. n = 2. Log2FC normalized to *Gapdh*.

They should perform RNAseq analysis of *Alb-Cre/dCas9-SPH* injected with the Adeno NT sgRNA and PlexinB2 sgRNA to identify specific activation of Plexin B2 and to show which other genes have been induced either in the NT ctrl or the PlexinB2 sgRNA used in the study.

We have conducted bulk RNA seq analysis of livers overexpressing *Plxnb2* and controls, and found upregulation of gene sets involved in proliferation, and downregulation of genes in metabolism. While the role of Plexin B2 in hepatocyte homeostasis and function is not well studied and will be the focus of further investigation, we have not observed phenotypic abnormalities in mice with *Plxnb2* overexpression or deletion. Furthermore, we found unchanged cellular composition, as predicted by deconvolution of bulk liver profiles using Tabula Muris⁵² as a reference (**Extended Data Fig. 6d-f**). Together with our in vitro assays, these results indicate that hepatocyte-derived Plexin B2 has a direct effect on the grafting of DTCs.

Extended Data Fig. 6 | d,e, DEGs and GSEA in *Plxnb2* OE vs. control livers (n = 3). **f**, Cell type enrichment analysis in bulk RNASeq of *Plxnb2* OE vs. control livers (n = 3).

I was also wondering whether the authors observe any immune related anti-transplant issues towards the organoids expressing a secreted Cherry protein. Could the authors please transplant the sLP-mCherry transduced organoids in Rag1 vs wt mice and test the seeding in the liver. Will this have any effects?

We thank the reviewer for this remark. We have not observed any immune related transplantation issues related to the sLP-mCherry protein expression. We performed the suggested experiment for another unpublished project in the lab, but using the PDAC cell line (*KPC^{sLPmCherry}*) and could not detect differences in grafting in B6 vs NXG mice.

Similarly, the amount of metastatic foci induced by injection of AKPS organoids with or without sLPmCherry expression was comparable. This is in line with the initial characterization by the Malanchi lab in Rag1 mice that shows no adaptive immunogenicity against sLP-mCherry (Ombrato et al, 2019, Extended Data Fig 2d-f), copied below for the reviewer's convenience.

[REDACTED]

d-f, CD45+ cell frequency on live cells in distal lung, Cherry-niche and naïve lungs (collected from mice which were not injected) by FACS: (d) Balb/c mice injected with Labelling-4T1 cells (n=5 per group); (e) Balb/c mice injected with Labelling-HC11 cells (n=4); (f) Rag1ko mice injected with Labelling-4T1 cells (n=10). Statistical analysis by paired two-tailed t-test. Data are represented as mean \pm SEM.

For the screen the authors decide to use 3 sgRNAs per gene. As described by Heidersbach AJ, Nature Comms 2023, it is very clear that not all sgRNAs will work and that it often requires many more optimally positioned sgRNAs to obtain robust expression of the gene of interest. Can the authors provide more information on the individual sgRNA potency? From Figure 1E (Figure 1G is mentioned in the main text but the Figure 1g doesn't exist) it looks like that Plexin B2 sgRNAs were strongly enriched. However, it is not clear which of the sgRNAs enrich the most. Is it only one sequence or is it distributed across the 3 independent sgRNAs targeting Plexin B2 found across 7 mice. Can the authors provide qPCR data or immune staining as performed in the Extended data to show how each individual sgRNA targeting Plexin B2 performs in the hepatocytes?

We thank the reviewer for the remark and agree that different sgRNAs may have different potencies. We extracted the per-sgRNA enrichment and found that mostly sgRNA1 and 2 were enriched in proximal hepatocytes across 3 library batches.

Proximal to distal enrichment (log₂FC) of individual sgRNAs targeting *Plxnb2* across 3 screening experiments and a total of 7 mice.

To test their effect on *Plxnb2* transcriptional activation, we transfected individual sgRNAs in immortalized hepatocytes (AML12) stably expressing dCas9-SPH and quantified *Plxnb2* expression by qRT-PCR. We believe that even if *in vitro*, this experiment provides the information regarding potency of individual sgRNAs and further complies with the 3R principle (specifically, reducing the amount of *in vivo* experiments by replacing them with *in vitro* assays). Our results indicate that sgRNA 2 and 3 are more potent than sgRNA 1, and achieve *in vitro* a 2-fold upregulation of *Plxnb2*, which is similar to our findings *in vivo*.

Plxnb2 expression levels in AML12-SPH cells transfected with individual sgRNAs targeting *Plxnb2* compared to non-targeting, as quantified by qRT-PCR. Barplots indicate the mean and standard deviation of two independent experiments. For every experiment, results from three technical replicates (3 transfected wells per condition) were averaged.

In the *in vitro* experiments I am not quite clear on the experimental setup. In the methods section the authors say: “On the next day, primary or immortalized hepatocytes were transfected with SB100X and a pool of 3 transposon vectors harboring sgRNAs against selected gene targets using Lipofectamine 3000 (Thermo Fisher Scientific). Three wells were independently transfected for each target in an arrayed fashion, and three wells were left untransfected.” Does that mean that per well 3 sgRNAs for one gene or 3 sgRNAs targeting 3 genes were used?

We have amended the Method section to improve clarity.

On the next day, primary AlbCre;dCas9-SPH hepatocytes or AML12-SPH were transfected with SB100X and transposon vectors harboring sgRNAs against selected gene targets using Lipofectamine 3000 (Thermo Fisher Scientific). For every target, three sgRNAs were independently cloned and amplified into transposon vectors, and then pooled prior to transfection. Three wells were transfected for each target, and three wells were left untransfected.

In the same experiment colony formation of the organoids was investigated on the AML12 cell line. Obviously dCas9-SHP was lentivirally introduced into this line, but what other than that is different to the primary hepatocyte assays? To exclude any issues of the sgRNA inducing PlexinB2 the

authors should use the AML12 cell line and introduce a Plexin B2 expression construct. This should show similar effects on colony formation than the CRISPRa expression.

We thank the reviewer for the remark and agree that *Plexnb2* overexpression in primary *AlbCre;dCas9-SPH* hepatocytes and AML12-SPH relies on the same mechanism. We have therefore transduced AML12 cells with a lentiviral vector harboring the coding sequence of VSV-G-tagged Plexin B2. Efficient expression and membrane localization was assessed by VSV-G immunofluorescence and flow cytometric analysis (Extended Data Fig 3f,g). We observed a similar effect on AKPS seeding and now include this in Fig. 2k.

Figure 1 | **k**, CFU per well of AKPS^{sLpMCherry} organoids after co-culture with AML12 cells overexpressing *Plexnb2*. Two-tailed unpaired t-test.

Extended Data Fig. 3 | *In vitro* screen and Plexin B2 overexpression. **f**, Representative fluorescent micrograph of AML12 transduced with pLVX-Plexnb2-VSVG or pLVX-EV stained with anti-VSVG (red) and anti-E-cadherin (green). **g**, Plexin B2-PE mean fluorescent intensity (MFI) in VSVG⁺ and VSVG⁻ AML12 cells, as assessed by flow cytometry. Two-tailed unpaired Wilcoxon test.

Since the experimental errors in the *in vitro* assays are quite large, and the differences just statistically significant, I would like to ask the authors to perform more repeats of the *in vitro* experiments and give more background on the analysis, i.e., was it done automated or did someone count the colonies. If the latter than it is important to mention how this person was related to the experiments. Did one person do the experiment and count the colonies or was there someone blinded to the experimental conditions and counted the colonies?

We agree with the reviewer that the effect sizes in the *in vitro* experiments are not extremely strong. However, they are statistically significant ($P < 0.05$), which indicates sufficient statistical power. We further believe that multiple orthogonal and independent experiments showing the same effect *in vitro* as well as *in vivo* are sufficient to support our conclusions. Nonetheless, we included more replicates and gave the

results of our in vitro results to an independent researcher, which was blinded to the conditions, for scoring. Colonies were quantified both manually and automatically by thresholding on the DAPI or mCherry area.

The authors use human mutational analysis to correlate their findings in human liver metastases (Figure 2). Targets include metastasis promoting (e.g. Plexin B2) and non-promoting targets (e.g. App). However, very little information is given on the nature of the mutations found on liver metastatic cancers. Could the authors please elaborate a little bit more on this? Where these gene abolishing or enhancing mutations?

We thank the reviewer for the comment and include now a more in depth analysis of the mutations in the Nguyen dataset, however in this dataset only CNV status (deleted or amplified) is reported, and the rest of the genes are referred to as “mutated”. We found that 21 LRs associated with hepatic metastases were predicted to interact with top-scoring hits of our screen, such as *Plxnb2*, *Psen1*, *App*, and *Ncam1* (Fig. 1h). Interestingly, deleted LRs in tumor cells were predicted to interact with seeding-suppressing factors, while amplified LRs mainly with promoting factors. This might suggest a selection of these interactions.

Figure 1 | h, Predicted interactions between seeding regulating factors and LRs frequently mutated in hepatic metastases. Copy number variation (CNV) status shown in orange (amplified), yellow (mutation) or blue (deleted).

As semaphorin genes were not assayed in the Nguyen dataset, we conducted a separate CNV analysis in the COAD dataset, which revealed that *SEMA4A*, *SEMA4C* and *SEMA4D* are commonly found amplified in CRC patients, while *SEMA4G* is often lost. This is in line with our previous analysis that correlated increased expression of *SEMA4A*, *SEMA4C* and *SEMA4D*, but not *SEMA4G*, with reduced recurrence-free survival in a large cohort of CRC patients.

Extended Data Fig. 9 | g, CNV analysis *SEMA4A*, *SEMA4C*, *SEMA4D* and *SEMA4G* in COAD dataset. n = 290 patients. **h**, Kaplan-Meier analysis (<http://kmpplot.com>) of the recurrence free survival (RFS) of CRC patients stratified by *SEMA4A*, *SEMA4C*, *SEMA4D* and *SEMA4G* expression. n = 1211 patients. d,e,f Two-tailed unpaired Wilcoxon test.

The authors describe an increase of *PLXNB2* in the liver leads to increased seeding. However, all hepatocytes seemed to express *PLXNB2* at a lower level. Is it just the increased expression of *PLXNB2* or something else leading to increased seeding. The in vitro experiments with the recombinant *PLXNB2* indicate an effect on proliferation of the tumour cells and the authors make the argument in the paper that this is not the case in vivo as the colony size is similar to cells that seeded in the wt liver. Does that mean that the in vitro experiments do not really reflect the in vivo scenario? Can you formerly rule out a proliferative advantage in vivo? Please show with KI67 or BrDU stains of the tumour cells seeded in the liver that they are indeed non proliferative. Could it be that in wt livers the metastasising cell are just not detectable, because of lower proliferative rates? Have you tested in the NT sgRNA ctrl, whether the hepatocytes surrounding the metastases express any of your targets you identified in your screen, e.g. *PLXNB2* or others?

Plxnb2 is normally expressed by both murine and human hepatocytes. Indeed, it was included in our library - together with 190 other targets of our screen - as an LR gene expressed by quiescent murine hepatocytes according to⁵³. Therefore, all of our targets are expected to be expressed by hepatocytes surrounding metastases. Notably, we found upregulation of *Plxnb2* protein levels in murine livers harboring AKPS metastases by immunofluorescence. Interestingly, *Plxnb2* upregulation was only occurring in hepatocytes around the lesions. This indicates a local response to the presence of the tumor. Indeed, *Plxnb2* was not found upregulated in livers of mice harboring primary AKPS colorectal tumors, as compared to sham controls. This indicates that *Plxnb2* is not upregulated in response to tumor secreted factors and systemic effects. The observed local upregulation of Plexin B2 led us to speculate that it might be involved in a regenerative response. Indeed, *Plxnb2* KO mice exhibit impaired liver vascular regeneration after partial hepatectomy⁴⁵. Moreover, Plexin-semaphorin signaling was implicated in epithelial repair⁴⁶. However, in depth characterization of the role of Plexin B2 in liver regeneration is outstanding and will be the focus of further studies. We copy below the relative result section and figures.

At steady state, Plexin B2 is widely expressed by hepatocytes - with higher expression in portal areas (Extended Data Fig 5g, h) - as well as endothelial cells of the hepatic sinusoids and portal veins³⁷. Interestingly, while Plexin B2 immunoreactivity was higher in peritumoral hepatocytes (Extended Data Fig. 5i,j), Plxnb2 expression was unaltered in livers of mice bearing AKPS colon tumors, suggesting that Plxnb2 is not upregulated by primary tumor secreted factors nor systemic effects, but rather due to a local response to the presence of metastases (Extended Data Fig. 5k). Interestingly, ablation of Plxnb2 5 days after intrasplenic AKPS organoid injection delayed but did not prevent metastasis formation, indicating that the presence of Plexin B2 on peritumoral hepatocytes is only required for metastatic seeding, but not to sustain growth (Extended Data Fig. 5l,m).

Extended Data Fig. 5 | g, Multiplexed *in situ* hybridization (Molecular Cartography) of *Plxnb2* as well as portal and central hepatocyte markers in WT mouse liver. **h**, Quantification of *Plxnb2* mRNA abundance in portal and central areas. Two-tailed unpaired Wilcoxon test. **i**, Representative fluorescent micrograph of Plexin B2 expression in AKPS liver metastasis. Scale bar, 100 μm . **j**, Plexin B2 staining intensity in peritumoral and distal areas. Dots represent individual metastases in $n=3$ mice. Two-tailed unpaired T-test. **k**, Differential gene expression analysis in livers of primary colon tumor-bearing vs naive mice. $n = 3$.

We also agree with the reviewer that the effect of Plexin B2 on proliferation remained unclear in the submitted manuscript, but we now provide new experimental data suggesting that Plexin B2 promotes proliferation both *in vitro* and *in vivo*. Single-cell transcriptional profiling further revealed that, in AKPS organoids, 2 hours treatment with rmPlexin B2 was sufficient to induce a compositional shift towards a more proliferative cell population (Fig 3a). In line with an induction of proliferation, rPlexin B2 increased frequencies of EdU⁺ cells in AKPS organoids and PDOs (Extended Data Fig. 7a,b). Lesions in *Plxnb2* OE livers further harbored higher density of Ki67⁺ epithelial cells and lower cleaved-caspase 3 levels, indicating a proliferative advantage *in vivo* (Fig. 3b, Extended Data Fig. 7c).

Figure 3 | Plexin B2 induces epithelialization of liver metastases. **a**, UMAP of rmPlexin B2-treated AKPS organoids and controls profiled by scRNASeq. Colors indicate distinct transcriptional clusters. Significantly enriched gene ontology terms and cluster proportions per sample are shown. **b**, Representative fluorescent micrograph and quantification of AKPS metastases in *Plexin2* OE and control livers. E-cadherin shown in green, Ki67 shown in magenta, DAPI used as nuclear counterstain. Scale bar, 100 μ m.

Extended Data Fig. 7 | a, Edu⁺/Edu⁻ cells in two-dimensional AKPS organoid cultures treated with rmPlexin B2. **b**, Left, Edu⁺/Edu⁻ cells in two-dimensional PDO cultures treated with rhPlexin B2. Right, representative fluorescent micrograph of Edu⁺ PDO colonies (right). Scale bar, 10 μ m. **c**, Representative fluorescent micrograph and quantification of AKPS metastases in *Plexin2* OE and control livers showing nuclear DAPI stain (cyan), and E-cadherin (gray) and cleaved caspase-3 (yellow) immunoreactivity. Dots represent individual metastases in n = 2 mice.

We further monitored AKPS metastases in *Plexin2*-KO livers by bioluminescence by lentivirally introducing the pLVX-fireflyLuc-IRES-zsGreen1 vector in AKPS organoids and monitoring tumor growth in vivo by IVIS. While Plexin B2 deletion prior to tumor inoculation significantly reduced BLI signal and tumor burden, liver colonization was delayed but not impaired when *Plexin2* was ablated 5 days after tumor inoculation, suggesting that the presence of Plexin B2 is not required to sustain tumor growth.

Figure 2 | Hepatocyte-derived Plexin B2 is required for liver colonization. **f**, BLI signal over time in *Plexin2*^{lox/lox} or *Plexin2*^{wt/wt} mice injected with AAV8-*A1bCre-moxGFP* and subsequently intrasplenically injected with AKPS^{Luciferase;zsGreen} organoids (Fisher's exact test, n = 3).

Extended Data Fig. 5 | Plexin B2 is required for spontaneous metastatic seeding. **i**, BLI signal over time in *Plexin2* KO and control mice injected with AAV8-*A1bCre-moxGFP* (AAV) 5 days post intrasplenic injection of AKPS organoids and (n = 3). Two-tailed unpaired

T-test. **m**, Representative H&E section of AKPS metastases in livers of *Plxnb2* KO and control mice injected with *AAV8-AlbCre-moxGFP* (AAV) 5 days post intrasplenic injection.

In the manuscript it is suggested that survival of the cancer cells is enhanced through the PlexinB2 signalling. However, no experimental data on this is provided. Could the authors please experimentally address this point further? They could stimulate AKAP organoids with recombinant PlexinB2 *in vitro* and then measure the expression of pro-survival as well as pro-apoptotic BCL-2 proteins by Western Blot analysis. In addition, they should stain cultures of vehicle vs recombinant Plexin B2 treated AKAP organoids for apoptotic markers, such as AnnexinV/ PI.

Single cell profiling of organoids revealed that rPlxnb2 treatment significantly decreases expression of genes involved in apoptosis in AKPS organoids (FDR = 3.620902e-11). We agree with the reviewer that the study of the effect of Plexin B2 on the survival of cancer cells requires a deeper investigation on the protein level, however we preferred to study this *in vivo* as the treatment *in vitro* involves organoid dissociation and detachment from Matrigel, which naturally lead to anoikis and therefore would likely lead to confounding effects. We therefore assessed Ki67 and cleaved caspase 3 levels in AKPS metastases growing in *Plxnb2*-overexpressing and control livers (see above). Nevertheless, as survival was not formally addressed, we removed any statement about Plexin's effect on survival from the text and we now only discuss its seeding-promoting effects.

The authors make the link to Rac1 signalling and show some Actin filament changes (most of the immunofluorescence images are not quantified and the authors should quantify the images in Figure 3 and Extended Figure 4 and 5) and inhibitor studies. However, the Rac1 link should be formally shown by performing a Rac1 activity assay *in vitro* of organoids stimulated with recombinant Plexin B2.

We thank the reviewer for the comment and we have now quantified all actin images, including AKPS metastases in *Plxnb2*-overexpressing livers (Fig 3e), PDOs and KPC cells treated *in vitro* with recombinant Plexin B2 (Extended Data Fig. 7g,h) and AKPS organoids treated with Klf4 inhibitor (Extended Data Fig. 7k,l).

Figure 3 | e, Representative fluorescent micrograph and quantification of F-actin in AKPS metastases in *Plxnb2* OE and control livers. Scale bar, 20 μ m.

Extended Data Fig. 7 | g, Representative fluorescent micrograph and quantification of F-actin in two-dimensional PDO cultures treated with rhPlexin B2. Scale bar, 10 μ m. **h**, F-actin intensity in KPC cultures treated with rmPlexin B2. **k, l** Representative brightfield and fluorescent micrograph and quantification of two-dimensional AKPS cultures treated with the Klf4 inhibitor WX2-43. F-actin shown in gray, EdU in red, DAPI in blue. Scale bar, 10 μ m. **a, c, h, i, j, l** Two-tailed unpaired T-test, **b, g** Ordinary one-way ANOVA, comparing each treatment group to vehicle control. Tukey's correction for multiple testing. **a, b, g, h, l**, Dots represent individual wells.

We also tried to show formal Rac1 activation in AKPS organoids following rPlexin B2 treatment. Unfortunately this assay repeatedly failed technically, we speculate due to cell loss during processing and Rac1 protein levels below detection limits. However, we identified a dependency on Rac1 signaling both for the *in vitro* seeding promoting effects of rhPlexin B2 on PDOs, and the *in vivo* seeding promoting effect of rmPlexin B2 on AKPS organoids.

Figure 4 | Plexin B2 binds semaphorins on tumor cells. **a**, Metastatic foci per liver section 2 weeks post intrasplenic injection of AKPS organoids treated *ex vivo* with rmPlexinB2 and/or the Rac1 inhibitor NSC323766. Dots represent individual mice (n=3). **c**, CFU per well of PDOs cultured with 2 μ g/mL rhPlexin B2 and NSC 23766.

In the final experiments the authors identify the Semaphorin family as possible interactors of PlexinB2. Since there are multiple members (4) the authors use a shRNA approach to diminish expression of all 4 family members. The knockdown efficiency looks not very strong, but there are clear effects *in vivo*. The authors could test the effect of this knockdown *in vitro* by stimulating the shRNA expressing organoids with recombinant Plexin B2 and test the proliferative capacity. They could do this in a competition assay whereby they combine knockdown vs wt organoids in the same well and treat with PlexinB2.

To address this point and achieve a complete depletion of semaphorins, we generated quadruple KO AKPS organoids (AKPS^{qKO}) lacking *Semaphorin 4A, C, D and G* with enhanced Cas12a technology, which enables highly efficient multiplexed genome editing. Quadruple KO was validated by NGS and immunofluorescence in multiple clonal lines. As Cas12a was integrated stably in organoids via lentiviral technology, we used organoids expressing only Cas12a as controls. Quadruple editing was validated by NGS and immunofluorescence in multiple clonal lines (**Extended Data Fig 10e, Supplementary Table S2**). While AKPS^{qKO} organoids show unaltered proliferation *in vitro*, as compared to control organoids, they exhibit significant grafting impairment *in vivo*, as they fail to form primary tumors and metastases when inoculated orthotopically (**Extended Data Fig 10f-h**). To assay liver colonization, we injected AKPS^{Sema4KO} organoids intrasplenically, and found significantly decreased metastatic burden compared to control organoids (**Fig. 4e,f**). Strikingly, Klf4 immunoreactivity is lost in AKPS^{Sema4KO} metastases, which further exhibit diminished Epcam and E-cadherin expression and high Zeb1 levels (**Fig. 4g**). Loss of semaphorins in AKPS metastases thus phenocopies loss of Plexin B2 on hepatocytes, supporting the notion that Plexin-semaphorin interactions are required for metastatic seeding (**Fig. 4h**).

Extended Data Fig. 10 | Class IV semaphorins on tumor cells are required for metastatic seeding. e, Representative fluorescent micrograph of Sema4A, C, D and G immunoreactivity in AKPS^{Sema4KO} and AKPS^{Sema4WT} organoids. Scale bar, 20 μ m.

Gene	Genotype	Sequence	
		DNA	Protein
Sema4a	WT	AGCCAAGGGAACCAAGTGAATGTCTTCTCTGGCTGTCTTCAGGCAAGTTGCGGGACCAAGAGCTCAGCAGTCTGTGCTTCTCTCACGGACATTGAGCGAGTCTTAAAGGGAAGTACAAGGAGCTGAACAAGGAGACCTCCCG	PREPQMSSSLAVFRQVGGTRSSAVCAFSLTDIERVFKGKYKELNKETSR
	Sema4KO clone 3	23% of reads TAAAGGGAAAGTACAAGGAGCTGAACAAGGAGACCTCCCGCTGAGCCACTACCGGGCTCAGAGGTCAGCCCGAG	PREPQMSSSLAVFRQVGGTRSSAVFKGKYKELNKETSRWITYRGSEVSPX
		77% of reads AGCCAAGGGAACCAAGTGAATGTCTTCTCTGGCTGTCTTCAGGCAAGTTGCGGGACCAAGAGCTCAGCAGTCTGTGCTTCTCTCTGAGCTGAACAAGGAGACCTCCCGCTGAGCCACTACCGGGCTCAGAGGTCAGCCCGAGGCCA	PREPQMSSSLAVFRQVGGTRSSAVCAFSLSTRRPPAGPLTGAQRSARGX
Sema4c	WT	TTCTCTGTCCCTGGTATGGTGGGAGCAAGTAGGCACCCAACTGACTTCAAAGTCTCTTTAAACAGGGGGCGATATGGACCTGTCTGCAGTTTGTGAGTACCAGTTGGAACAGATCCAGCAAGTGTGGAGGTCCTTACAAGGAGTA	PLFPGMVGASRHPPLRPSFFNRGMDLMSAVCEYQLEIQVQVFEQPYKEX
	Sema4KO clone 3	37% of reads TTCTCTGTCCCTGGTATGGTGGGAGCAAGTAGGCACCCAACTGACTTCAAAGTCTCTTTGTGAGTACCAGTTGGAAACAGATCCAGCAAGTGTGGAGGTCCTTACAAGGAGTACAGTACAGCCAGCCGCTAT	PLFPGMVGASRHPPLRPSFFVSTWNRSSKCLRVPTRSTVSKPRSPGPAK
		63% of reads TTCTCTGTCCCTGGTATGGTGGGAGCAAGTAGGCACCCAACTGACTTCAAAGTCTCTTTAAACAGGGGGCGATATGGTCTCTGCAGTTTGTGAGTACCAGTTGGAACAGATCCAGCAAGTGTGGAGGTCCTTACAAGGAGTACAGTGTG	PLFPGMVGASRHPPLRPSFFNRGDMVCSLVPVGTDPASVGLQGVQX
Sema4d	WT	AAGSGTCTTTGATGGTGGCTGTTAATGGGGTGTTCCTTCCAGACGGAATGCCTAACTACATTGGAAGTACTACAGCCACTAAGCAGCACTTCCCTCTATGTGTGGGACCAATGCCAGTGTTCAGGCCACCTGTGACCACTGGTAAGA	RVLWVPVNGVFPLQTECLNRYRVLQPLSSTLSYVCGTNAFQPTCDHLVR
	Sema4KO clone 3	54% of reads AAGSGTCTTTGATGGTGGCTGTTAATGGGGTGTTCCTTCCAGACGGAATGCCTAACTACAGTACTACAGCCACTAAGCAGCACTTCCCTCTATGTGTGGGACCAATGCCAGTGTTCAGGCCACCTGTGACCACTGGTAAGCACAACATA	RVLWVPVNGVFPLQTECLNEYSHAALPSCMVGPMRSPVPTTWDHX
		46% of reads AAGSGTCTTTGATGGTGGCTGTTAATGGGGTGTTCCTTCCAGACGGAATGCCTAACTACAGTACTACAGCCACTAAGCAGCACTTCCCTCTATGTGTGGGACCAATGCCAGTGTTCAGGCCACCTGTGACCACTGGTAAGCACAACATA	RVLWVPVNGVFPLQTECLNYSHAALPSCMVGPMRSPVPTTWDHNI
Sema4g	WT	TGATAAGCTAGACTAATGGAGGTGGCTTGTCTCTGGTGTGATGAGTCTATCACTGCCTCCCTCTGCTGTAGACCGAGTGTTCACACCAGTGCATTTCTACAGCGGCTCAATGCCACCCTCTATGATGTGGGACCAAGCCCTT	DKLDWRWLVSGLMSSITASLCLTECFNH/RFLQRLNATHFYACGTHAX
	Sema4KO clone 3	22% of reads TGATAAGCTAGACTAATGGAGGTGGCTTGTCTCTGGTGTGATGAGTCTATCACTGCCTCCCTCTGCTGTAGACCGAGTGTTCACACCAGTGCATTTCTCACGCCCTCCAGCCCTCTGTGACGACTATTGTGAGTACGCTGCCCCCTTG	DKLDWRWLVSGLMSSITASLCLTECFNH/RFLPRLPAPLCSYCEYACPL
		78% of reads TGATAAGCTAGACTAATGGAGGTGGCTTGTCTCTGGTGTGATGAGTCTATCACTGCCTCCCTCTGCTGTAGACCGAGTGTTCACACCAGTGCATTTCTATGCATGTGGGACCAAGCCCTCCAGCCCTCTGTGACGACTATTGTGAG	DKLDWRWLVSGLMSSITASLCLTECFNH/RFLCMWDPRLPAPLCSYCE

Supplementary Table 2 | Indel analysis of AKPS^{Sema4KO} (clonal line 3) showing out of frame mutations in all 4 targeted semaphorin genes. Green rows show WT base and amino acid sequence. Mutated parts shown in red.

Extended Data Fig. 10 | Class IV semaphorins on tumor cells are required for metastatic seeding. **f**, Edu⁺/Edu⁻ cells in two-dimensional AKPS^{Sema4KO} and AKPS^{Sema4WT} organoids. **g**, Representative images of livers (top) and primary colon tumors (bottom) in mice orthotopically injected with AKPS^{Sema4KO} or AKPS^{Sema4WT} organoids. **h**, Incidence of spontaneous liver metastases in mice orthotopically injected with AKPS^{Sema4KO} or AKPS^{Sema4WT} organoids (n = 8).

Figure 4 | Plexin B2 binds semaphorins on tumor cells. **e**, Metastatic foci per liver section 2 weeks post intrasplenic injection of AKPS^{Sema4WT} or AKPS^{Sema4KO} organoids. Dots represent individual mice (n=3). **f**, Representative H&E staining of AKPS^{Sema4WT} or AKPS^{Sema4KO} metastases. Scale bar, 100 μm . **g**, Representative fluorescent micrograph and quantification of Epcam (in gray) and Klf4 (in red) immunoreactivity in AKPS^{Sema4WT} and AKPS^{Sema4KO} metastases. Scale bar, 20 μm . Arrow and inset show Klf4⁺ nuclei. Scale bar, 10 μm . **h**, Representative fluorescent micrograph and quantification of E-cadherin (in gray) and Zeb1 (in red) immunoreactivity in AKPS^{Sema4WT} and AKPS^{Sema4KO} metastases. Scale bar, 20 μm . Inset 1 shows E-cadherin⁺Zeb1⁺ nuclei. Inset 2 shows E-cadherin⁺Zeb1⁻ nuclei. Scale bar, 10 μm . **i**, Plexin B2 on hepatocytes binds class IV semaphorins on DTCs, inducing Klf4 upregulation and Zeb1 downregulation, which results in epithelialization and outgrowth of metastases. a,b,c Dots indicate individual wells. Ordinary one-way ANOVA, comparing each treatment group to control. Tukey's correction for multiple testing. d,e,g,h Two-tailed unpaired t-test. Barplots indicate mean \pm SD.

References

1. Roper, J. *et al.* Colonoscopy-based colorectal cancer modeling in mice with CRISPR–Cas9 genome editing and organoid transplantation. *Nat. Protoc.* **13**, 217–234 (2018).
2. Varga, J. *et al.* AKT-dependent NOTCH3 activation drives tumor progression in a model of mesenchymal colorectal cancer. *J. Exp. Med.* **217**, (2020).
3. Zigmund, E. *et al.* Utilization of murine colonoscopy for orthotopic implantation of colorectal cancer. *PLoS One* **6**, e28858 (2011).
4. Reichert, M. *et al.* Regulation of Epithelial Plasticity Determines Metastatic Organotropism in Pancreatic Cancer. *Dev. Cell* **45**, 696–711.e8 (2018).
5. Sanghvi, N. *et al.* Charting the transcriptomic landscape of primary and metastatic cancers in relation to their origin and target normal tissues. *bioRxiv* (2023) doi:10.1101/2023.10.30.564810.
6. Jolly, M. K., Ware, K. E., Gilja, S., Somarelli, J. A. & Levine, H. EMT and MET: necessary or permissive for metastasis? *Mol. Oncol.* **11**, 755–769 (2017).
7. Christiansen, J. J. & Rajasekaran, A. K. Reassessing epithelial to mesenchymal transition as a prerequisite for carcinoma invasion and metastasis. *Cancer Res.* **66**, 8319–8326 (2006).
8. McFaline-Figueroa, J. L. *et al.* A pooled single-cell genetic screen identifies regulatory checkpoints in the continuum of the epithelial-to-mesenchymal transition. *Nature Genetics* vol. 51 1389–1398 Preprint at <https://doi.org/10.1038/s41588-019-0489-5> (2019).
9. Jolly, M. K. *et al.* Implications of the Hybrid Epithelial/Mesenchymal Phenotype in Metastasis. *Front. Oncol.* **5**, 155 (2015).
10. Sinha, D., Saha, P., Samanta, A. & Bishayee, A. Emerging Concepts of Hybrid Epithelial-to-Mesenchymal Transition in Cancer Progression. *Biomolecules* **10**, (2020).
11. Zhang, Y. *et al.* Genome-wide CRISPR screen identifies PRC2 and KMT2D-COMPASS as regulators of distinct EMT trajectories that contribute differentially to metastasis. *Nat. Cell Biol.* **24**, 554–564 (2022).
12. Akhmetkaliyev, A., Alibrahim, N., Shafiee, D. & Tulchinsky, E. EMT/MET plasticity in cancer and Go-or-Grow decisions in quiescence: the two sides of the same coin? *Mol. Cancer* **22**, 90 (2023).
13. Subbalakshmi, A. R. *et al.* The ELF3 transcription factor is associated with an epithelial phenotype and represses epithelial-mesenchymal transition. *J. Biol. Eng.* **17**, 17 (2023).
14. Sengez, B. *et al.* The Transcription Factor Elf3 Is Essential for a Successful Mesenchymal to Epithelial Transition. *Cells* **8**, (2019).
15. Liu, D. *et al.* ELF3 is an antagonist of oncogenic-signalling-induced expression of EMT-TF ZEB1. *Cancer Biol. Ther.* **20**, 90–100 (2019).
16. Yang, Z., Wu, D., Chen, Y., Min, Z. & Quan, Y. GRHL2 inhibits colorectal cancer progression and metastasis via oppressing epithelial-mesenchymal transition. *Cancer Biol. Ther.* **20**, 1195–1205 (2019).
17. Xiang, J., Fu, X., Ran, W. & Wang, Z. Grhl2 reduces invasion and migration through inhibition of TGFβ-induced EMT in gastric cancer. *Oncogenesis* **6**, e284 (2017).
18. Presnell, J. S., Schnitzler, C. E. & Browne, W. E. KLF/SP Transcription Factor Family Evolution: Expansion, Diversification, and Innovation in Eukaryotes. *Genome Biol. Evol.* **7**, 2289–2309 (2015).
19. Xu, J. *et al.* Dynamic down-regulation of Krüppel-like factor 4 in colorectal adenoma-carcinoma sequence. *J. Cancer Res. Clin. Oncol.* **134**, 891–898 (2008).
20. Lee, H.-Y. *et al.* High KLF4 level in normal tissue predicts poor survival in colorectal cancer patients. *World J. Surg. Oncol.* **12**, 232 (2014).
21. Zhao, W. *et al.* Identification of Krüppel-like factor 4 as a potential tumor suppressor gene in colorectal cancer. *Oncogene* **23**, 395–402 (2004).
22. Choi, B. J. *et al.* Altered expression of the KLF4 in colorectal cancers. *Pathol. Res. Pract.* **202**, 585–589 (2006).
23. Subbalakshmi, A. R. *et al.* KLF4 Induces Mesenchymal-Epithelial Transition (MET) by Suppressing Multiple EMT-Inducing Transcription Factors. *Cancers* **13**, (2021).
24. Yori, J. L., Johnson, E., Zhou, G., Jain, M. K. & Keri, R. A. Krüppel-like Factor 4 Inhibits Epithelial-to-Mesenchymal Transition through Regulation of E-cadherin Gene Expression*. *J. Biol. Chem.* **285**, 16854–16863 (2010).
25. Fujimoto, S. *et al.* KLF4 prevents epithelial to mesenchymal transition in human corneal epithelial

- cells via endogenous TGF- β 2 suppression. *Regen Ther* **11**, 249–257 (2019).
26. Agbo, K. C. *et al.* Loss of the Krüppel-like factor 4 tumor suppressor is associated with epithelial-mesenchymal transition in colorectal cancer. *J Cancer Metastasis Treat* **5**, (2019).
 27. Zhou, Z. *et al.* A novel small-molecule antagonizes PRMT5-mediated KLF4 methylation for targeted therapy. *EBioMedicine* **44**, 98–111 (2019).
 28. Sun, H. *et al.* Metabolic switch and epithelial-mesenchymal transition cooperate to regulate pluripotency. *EMBO J.* **39**, e102961 (2020).
 29. Deshmukh, A. P. *et al.* Identification of EMT signaling cross-talk and gene regulatory networks by single-cell RNA sequencing. *Proc. Natl. Acad. Sci. U. S. A.* **118**, (2021).
 30. Zhou, H. *et al.* Recruitment of Tiam1 to Semaphorin 4D Activates Rac and Enhances Proliferation, Invasion, and Metastasis in Oral Squamous Cell Carcinoma. *Neoplasia* **19**, 65–74 (2017).
 31. Sun, T. *et al.* A reverse signaling pathway downstream of Sema4A controls cell migration via Scrib. *J. Cell Biol.* **216**, 199–215 (2017).
 32. Gurrapu, S. *et al.* Reverse signaling by semaphorin 4C elicits SMAD1/5- and ID1/3-dependent invasive reprogramming in cancer cells. *Science Signaling* vol. 12 Preprint at <https://doi.org/10.1126/scisignal.aav2041> (2019).
 33. Dhainaut, M. *et al.* Spatial CRISPR genomics identifies regulators of the tumor microenvironment. *Cell* **185**, 1223–1239.e20 (2022).
 34. Wan, J. *et al.* Methylated cis-regulatory elements mediate KLF4-dependent gene transactivation and cell migration. *Elife* **6**, (2017).
 35. Ghaleb, A. M. & Yang, V. W. Krüppel-like factor 4 (KLF4): What we currently know. *Gene* **611**, 27–37 (2017).
 36. Rivero, M., Montagnani, V. & Stecca, B. KLF4 is regulated by RAS/RAF/MEK/ERK signaling through E2F1 and promotes melanoma cell growth. *Oncogene* **36**, 3322–3333 (2017).
 37. Zielonka, M., Xia, J., Friedel, R. H., Offermanns, S. & Worzfeld, T. A systematic expression analysis implicates Plexin-B2 and its ligand Sema4C in the regulation of the vascular and endocrine system. *Exp. Cell Res.* **316**, 2477–2486 (2010).
 38. Lee, H. O., Hong, Y., Etilioglu, H. E., Cho, Y. B. & Pomella, V. Lineage-dependent gene expression programs influence the immune landscape of colorectal cancer. *Nature* (2020).
 39. Cañellas-Socias, A. *et al.* Metastatic recurrence in colorectal cancer arises from residual EMP1+ cells. *Nature* **611**, 603–613 (2022).
 40. Joanito, I. *et al.* Single-cell and bulk transcriptome sequencing identifies two epithelial tumor cell states and refines the consensus molecular classification of colorectal cancer. *Nat. Genet.* **54**, 963–975 (2022).
 41. Wang, F. *et al.* Single-cell and spatial transcriptome analysis reveals the cellular heterogeneity of liver metastatic colorectal cancer. *Sci Adv* **9**, eadf5464 (2023).
 42. Che, L.-H. *et al.* A single-cell atlas of liver metastases of colorectal cancer reveals reprogramming of the tumor microenvironment in response to preoperative chemotherapy. *Cell Discov* **7**, 80 (2021).
 43. Cancer Genome Atlas Network. Comprehensive molecular characterization of human colon and rectal cancer. *Nature* **487**, 330–337 (2012).
 44. Sohlenius-Sternbeck, A.-K. Determination of the hepatocellularity number for human, dog, rabbit, rat and mouse livers from protein concentration measurements. *Toxicol. In Vitro* **20**, 1582–1586 (2006).
 45. Worzfeld, T. *et al.* Genetic dissection of plexin signaling in vivo. *Proc. Natl. Acad. Sci. U. S. A.* **111**, 2194–2199 (2014).
 46. Xia, J. *et al.* Semaphorin-Plexin Signaling Controls Mitotic Spindle Orientation during Epithelial Morphogenesis and Repair. *Dev. Cell* **33**, 299–313 (2015).
 47. Peinado, H. *et al.* Pre-metastatic niches: organ-specific homes for metastases. *Nat. Rev. Cancer* **17**, 302–317 (2017).
 48. Schuster, E. *et al.* Computational ranking-assisted identification of Plexin-B2 in homotypic and heterotypic clustering of circulating tumor cells in breast cancer metastasis. *bioRxiv* (2023) doi:10.1101/2023.04.10.536233.
 49. Sanson, K. R. *et al.* Optimized libraries for CRISPR-Cas9 genetic screens with multiple modalities. *Nat. Commun.* **9**, 5416 (2018).
 50. Jung, J. *et al.* Genome-scale CRISPR-Cas9 knockout and transcriptional activation screening. *Nat. Protoc.* **12**, 828–863 (2017).

51. Morgens, D. W. et al. Genome-scale measurement of off-target activity using Cas9 toxicity in high-throughput screens. *Nat. Commun.* **8**, 15178 (2017).
52. Tabula Muris Consortium *et al.* Single-cell transcriptomics of 20 mouse organs creates a Tabula Muris. *Nature* **562**, 367–372 (2018).
53. Chembazhi, U. V., Bangru, S., Hernaez, M. & Kalsotra, A. Cellular plasticity balances the metabolic and proliferation dynamics of a regenerating liver. *Genome Res.* **31**, 576–591 (2021).

Reviewer Reports on the First Revision:

Referee #1 (Remarks to the Author):

The authors have done an excellent job addressing all of my criticisms. As a result, this revised version of the manuscript has significantly improved. In my opinion, this is an important study that merits publication in Nature.

Eduard Batlle.

Referee #3 (Remarks to the Author):

The authors have significantly expanded the study's physiological relevance and mechanistic aspects, which conclusively demonstrates that liver-to-tumour signaling enables metastatic colonization in the liver via the Plexin B2-semaphorin axis. The incorporation of spontaneous metastasis models, together with additional omics and functional assays, also strengthens their initial conclusions. Some specific mechanistic aspects and statements would benefit from further analyses and/or text clarifications, which this Reviewer considers could be achieved via a more integrated analysis (and interpretation) of much of the data already included (particularly omics), to tie everything together more clearly.

Specifically, it remains unclear whether the three major molecular changes/mechanisms the authors suggest underlie the pro-metastatic effects of the Plexin B2 (namely, Klf4 upregulation/activity; Zeb1 downregulation, and Rac1 signalling) are functionally interconnected, and which of their downstream phenotypic effects (epithelialization, proliferation, both?) are critical for the successful outgrowth of seeded CRC cells. Authors conclude that the seeding-promoting effects of the Plexin B2-Semaphorin axis are mediated by Klf4 upregulation (and Zeb1 downregulation). To support this, the authors first identify KLF binding sites as enriched in scATAC-peaks that are differentially accessible between AKPS metastases in Plxn2 OE and control livers. They further show that, while Klf4 expression is lost during malignant transformation, it is re-activated in successful metastasis (from AKPS organoids in Plxn2 OE livers; and control livers?). To assess functionality, they treat 2D AKPS cultures with an inhibitor WX2-43 and show that Klf4 inhibition is sufficient to induce a mesenchymal-like phenotype in these cells *ex vivo*, with parallel changes in the actin cytoskeleton and proliferation (reduced?) in this setting. While these data support the conclusion that Klf4 is functionally relevant for sustaining epithelial-like states in CRC, the experiment does not specifically probe Klf4 function in the context of Plexin B2 action. Thus, the final manuscript should more clearly relate the Klf4-related data (scATAC KLF-motifs *in vivo*, Klf4 inhibitor effects *in vitro*) with the Plexin B2 pro-seeding phenotypes (and other associated changes mentioned - Zeb1, Rac1, etc) by addressing and/or clarifying specific questions/aspects indicated below.

Overall, this Reviewer considers that the updated manuscript's conceptual and technical novelty merits publication in Nature, with the clarifications mentioned below and a more integrated analysis of the mechanistic data, so that the final text and figures describe the major findings in a more accurate and cohesive manner.

1. Regarding the effects of Klf4 inhibition in CRC cells (and their functional relevance): Overall, it is not very clear whether (and how) the different mechanisms mentioned (Zeb1 downregulation, Klf4 upregulation, Rac1 signaling) are inter-connected (eg: Klf4 acting upstream of Zeb1?) or whether these are parallel/independent pathways downstream of Plexin B2 signaling. Is Zeb1 upregulated upon Klf4 inhibition? Are Grhl2 and Elf3 (suggested in the text as mediators of (Klf4-driven?) epithelialization phenotype) downregulated? Does Klf4 inhibitor phenocopy the effects of Rac1 inhibition in rescuing rmPlexin B2 metastasis-seeding effects? This last point is relevant to support their conclusion that “the seeding-promoting effects of Plexin B2 on tumor cells [are mediated] by upregulating Klf4 and downregulating Zeb1”, as stated in the text (again, please clarify if the downregulation of Zeb1 is a consequence of Klf4 upregulation, or a parallel mechanism?). As presented now, it can't be ruled out that the observed expression of Klf4 (and gain in accessibility at KLF-motif+ loci) in Plexin B2/semaphorin-competent epithelialized metastasis is a consequence, rather than the causal mechanism, of a successful metastasis seeding process.

2. Along these lines, do authors interpret the epithelialization and proliferation phenotypes to be coupled and/or mechanistically linked, and equally relevant for Plexin B2's effects as a gatekeeper of liver metastasis? The data suggest Klf4 is important for the epithelialization phenotype (but not proliferation?) and Rac1 signaling underlies the proliferation (CFU; or also epithelialization?) of CRC cells? The text should be more precise in describing whether and how the different phenotypes and pathways linked to Plexin B2 signaling are interconnected (or not).

3. Regarding the upregulation and/or activation of Klf4 in tumor cells by Plexin B2-semaphorin signaling: Can authors please clarify if Klf4 expression, or only its predicted transcriptional activity (i.e. expression of genes associated with KLF-motif+ ATAC peaks and/or sensitive to Klf4 inhibition), is induced in the rhPlexin B2-treated organoids that gain epithelial morphogenesis, proliferation, and ultimately, CFU capacity? More broadly, authors should better integrate scRNA and scATAC dimensions of the multi-omics data (as now are mostly analyzed independently) and clarify how well these changes are recapitulated in the in vitro experiments used to dissect mechanism (eg what's the status of Klf4, Zeb1, Grhl2/Elf3 expression in rhPlexin B2- or Rac1/Klf4 inhibitor-treated cells changing epithelial/EMT/actin/proliferation traits?).

4. In the last section of the study, the authors show how semaphorins are significantly upregulated in CRC tumor cell states with increased metastasis-seeding capacity (HRC and iCMS3 cells), further supporting the role of Plexin B2 – semaphoring signaling in liver metastasis, and providing a mechanism to explain the differential capacity of distinct tumor cell subpopulations for successfully forming liver metastasis. This is a highly relevant addition of the updated manuscript, yet is not sufficiently well-linked with the rest of the elements studied (i.e. Plexin-OE-induced chromatin changes correlating with epithelialization/metastasis outgrowth, Klf4 as a putative mediator of such process, etc). To better connect the dots: can authors map HRC and iCMS3 signatures in the scRNA/ATAC-seq data in which a Plexin B2-induced enhanced KLF4 activity was inferred (Figure 3c, Figure 3f)? and, conversely, are Klf4 levels and/or its transcriptional output specifically enriched in these HRC and iCMS3 cells having increased Semaphorin levels and metastasis-seeding capacity?

Additional minor/related points:

- Fig. 1h & Ext Data Fig. 3a-related: Given that metastasis-seeding capacity in CRC (and other cancer types investigated in this work; PDAC, melanoma) are not exclusively determined by genetic/CNV changes, authors should also highlight seeding-regulating factors that are transcriptionally dysregulated (upregulated/downregulated) in hepatic metastasis (and/or in specific subpopulations with increased metastasis-seeding potential, eg HRC). As of now, the study appears to consider relevant only seeding-regulating factors that are mutated (amplified/deleted; Fig. 1h), making this analysis somehow disconnected from the new/expanded mechanistic data linking Plxn2-mediated effects to transcriptional and chromatin changes of CRC cells. I suggest expanding (or separately analyzing) transcriptionally dysregulated RLs, to also capture non-genetic mechanisms influencing the effects of the seeding-regulating factors identified in their screen.

- Fig. 2c-d related: it would be convenient to indicate in methods ("Intrasplenic injection of tumor cells followed by splenectomy" section) whether the PDAC cells inoculated were derived from 2D cell lines or 3D organoids, as 3D vs 2D cell line-derived single-cell suspensions may not behave similarly. Also, note a reference to the melanoma cell line is currently missing in this method's section.

- Fig. 3a / Ext data 8-related: rmPlexinB2 effects in organoids. (i) Please provide UMAP colored by conditions to facilitate the reader's understanding of which cells came from which rmPlexin B2-treated vs untreated APKS cells. (ii) Please map the expression of KLF4 (if detected) and of the gene program linked to gained KLF-motif+ ATAC-peaks (from Fig. 3 c-g) in the rmPlexin B2-treated vs untreated APKS cells (ideally also in the Klf4 inhibitor experiment). Also, are the epithelialization genes (eg Elf3, Grhl2) and/or HRC signatures induced upon rmPlexinB2 treatment, or is the major effect proliferation (CFU, Edu+, Ki67, etc) in this ex vivo setting? (iii) Can you clarify if Zeb1 is suppressed?

- Fig. 3c-f-related: scRNA/ATAC: (i) Distinct subpopulations of cells are observed: Can authors please describe which one(s) specifically show increased accessibility at KLF-motif containing loci? All, or is there a dominant one? (ii) Related to the above point: are any of these subpopulations (scRNA and/or ATAC) reminiscent of HRC or other metastasis CRC-relevant cell states (eg EMT/MET)? (signatures could be mapped here); (iii) As above, does the gain in accessibility at KLF-moti+ loci correlate with an increased expression of Klf4 (and/or other TF whose motifs are differentially enriched in ATAC peaks of compared conditions), or not necessarily?

- Extended Data Fig. 7j-related: Can authors please clarify if Klf4 expression is also gained in metastasis forming in regular livers (or only in the PlexinB2-OE setting)? If so, this would further support their conclusion that epithelialization (via Klf4?) is generally required for liver metastasis.

Referee #4 (Remarks to the Author):

The authors have addressed my comments. I have a few points which in my opinion need to be answered before publication of this manuscript.

1, My first question related to the reliability of the mouse model. While I think that it is sufficient for this study, I would like the authors to add a sentence into the manuscript that this model was reported to have some issues and it is important - as for any other study of course - to do the best possible control experiments.

To address the reported issues of the mouse model I asked a few questions, which the authors addressed with data that was already in the paper. However, I would like to raise a few issues with these experiments.

a, they perform mRNA quantification in Extended Figure 4c with a statistical Two-tailed unpaired t test. The graph however only shows 1 -2 data points per group. To perform a statistical analysis the authors need at least 3 independent data points. Either take away the statistical test or analyse more data points.

b, the staining intensity (how was this measured) in e and f has a 10 fold different x axis labeling (1000 - 4000 in e and 100 - 300 in f). Why is this? Again statistics in f with 2 mice used for the sgNT.

c, It is argued that the mice respond with similar amounts of target induction and Glu1 is used as an example. However, the fluctuation of Glu1 induction in Extended Figure 1b is between 2 to 8 fold - clearly a difference. Again sgNT is again only 2 mice and stats are analysed. Please change.

These experiments clearly suggest that expression is variable, which could be independent of the mouse model, but my concern on this model remains and should be discussed as mentioned above. Too many people use animal models which don't do what they are supposed to be doing and it is important to inform the scientific community.

2, The authors perform bulk RNA seq on liver cells transduced with the sgPlxnB2sgRNA and compare it to the NT ctrl. I couldn't find any info on the experiment. Did the authors just inject their routine cocktail of sgPlxnB2-OE cells at a 0.5% transduction rate? If so, did they sort the cells before isolating RNA for the RNAseq experiment? If the whole liver population was used it would be surprising to see the differences. Nevertheless, why was PlxnB2 not found? Or it is there and I couldn't see it in their DE blot. Could they please amend the methods and also demonstrate PlxnB2 upregulation in the RNAseq data?

3, Immunogenicity of slpmCherry: The authors argue that they have used another pancreas Organoid model and looked at seeding in the liver. Firstly, the blots shown in the rebuttal are not explained well and I assume that the left blot shows B6 injection and the right blot injection of slpmCherry organoids into B6 and NXG mice. It looks ok, but again n=2 only per group and I would have preferred to see the seeding of AKAP organoids into B6 and NXG mice as this is what is used in the paper. It could also be that many of the cells were killed by immunecells after transplantation. Has this been checked? Again the immunogenicity aspect should at least be discussed.

4, PlexinB2 induction, sgRNA efficiency:

I assume the authors wanted to write that sgRNA 2 and 3 are mostly enriched in the proximal to distal analysis as only these two sgRNAs show a 2 fold upregulation in vitro. I would have preferred to see these experiments in vivo as it is an important control. That would have been an additional 12 mice (3/ group). However, even in vitro n numbers are only 2 and the authors could have gone through the effort of at least 3 repeats since they haven't done an in vivo experiment.

5, The apoptosis angle is left out, although the authors still say that PlexinB2 enhances survival and show reduced Caspase3 activity. They also say that the reason for the effect in PlexinB2 livers however is through higher proliferation as shown by increased KI67. How do they explain then the first observation that the metastatic sides are similar in size between wt and Plexin B2 overexpressing cells in the liver? They also say that proliferation is up as shown by increased KI67 and lower Caspase 3 activity. Caspase 3 has nothing to do with proliferation. Please correct.

Referee #4 (Remarks on code availability):

I just had a look at the code but this is not my expertise.

Author Rebuttals to First Revision:

Referees' comments:

Referee #1 (Remarks to the Author):

The authors have done an excellent job addressing all of my criticisms. As a result, this revised version of the manuscript has significantly improved. In my opinion, this is an important study that merits publication in Nature.

Eduard Batlle.

Referee #3 (Remarks to the Author):

The authors have significantly expanded the study's physiological relevance and mechanistic aspects, which conclusively demonstrates that liver-to-tumour signaling enables metastatic colonization in the liver via the Plexin B2-semaphorin axis. The incorporation of spontaneous metastasis models, together with additional omics and functional assays, also strengthens their initial conclusions. Some specific mechanistic aspects and statements would benefit from further analyses and/or text clarifications, which this Reviewer considers could be achieved via a more integrated analysis (and interpretation) of much of the data already included (particularly omics), to tie everything together more clearly.

Specifically, it remains unclear whether the three major molecular changes/mechanisms the authors suggest underlie the pro-metastatic effects of the Plexin B2 (namely, Klf4 upregulation/activity; Zeb1 downregulation, and Rac1 signalling) are functionally interconnected, and which of their downstream phenotypic effects (epithelialization, proliferation, both?) are critical for the successful outgrowth of seeded CRC cells. Authors conclude that the seeding-promoting effects of the Plexin B2-Semaphorin axis are mediated by Klf4 upregulation (and Zeb1 downregulation). To support this, the authors first identify KLF binding sites as enriched in scATAC-peaks that are differentially accessible between AKPS metastases in Plxnb2 OE and control livers. They further show that, while Klf4 expression is lost during malignant transformation, it is re-activated in successful metastasis (from AKPS organoids in Plxnb2 OE livers; and control livers?). To assess functionality, they treat 2D AKPS cultures with an inhibitor WX2-43 and show that Klf4 inhibition is sufficient to induce a mesenchymal-like phenotype in these cells ex vivo, with parallel changes in the actin cytoskeleton and proliferation (reduced?) in this setting. While these data support the conclusion that Klf4 is functionally relevant for sustaining epithelial-like states in CRC, the experiment does not specifically probe Klf4 function in the context of Plexin B2 action. Thus, the final manuscript should more clearly relate the Klf4-related data (scATAC KLF-motifs in vivo, Klf4 inhibitor effects in vitro) with the Plexin B2 pro-seeding phenotypes (and other associated changes mentioned - Zeb1, Rac1, etc) by addressing and/or clarifying specific questions/aspects indicated below.

Overall, this Reviewer considers that the updated manuscript's conceptual and technical novelty merits publication in Nature, with the clarifications mentioned below and a more integrated analysis of the mechanistic data, so that the final text and figures describe the major findings in a more accurate and cohesive manner.

1. Regarding the effects of Klf4 inhibition in CRC cells (and their functional relevance): Overall, it is not very clear whether (and how) the different mechanisms mentioned (Zeb1 downregulation, Klf4 upregulation, Rac1 signaling) are inter-connected (eg: Klf4 acting upstream of Zeb1?) or whether these are parallel/independent pathways downstream of Plexin B2 signaling. Is Zeb1 upregulated upon Klf4 inhibition? Are Grhl2 and Elf3 (suggested in the text as mediators of (Klf4-driven?) epithelization phenotype) downregulated? Does Klf4 inhibitor phenocopy the effects of Rac1 inhibition in rescuing rmPlexin B2 metastasis-seeding effects? This last point is relevant to support their conclusion that “the seeding-promoting effects of Plexin B2 on tumor cells [are mediated] by upregulating Klf4 and downregulating Zeb1”, as stated in the text (again, please clarify if the downregulation of Zeb1 is a consequence of Klf4 upregulation, or a parallel mechanism?). As presented now, it can't be ruled out that the observed expression of Klf4 (and gain in accessibility at KLF-motif+ loci) in Plexin B2/semaphorin-competent epithelialized metastasis is a consequence, rather than the causal mechanism, of a successful metastasis seeding process.

We thank the reviewer for the remark and agree that our *in vivo* and *in vitro* observations are not extensively linked in the manuscript. As we believe that *in vivo* results are more physiologically relevant, we did not attempt to validate those *in vitro*. Indeed, it is not clear to what extent the morphological, epigenetic and transcriptional changes observed two weeks post inoculation *in vivo* are recapitulated by a short treatment with rmPlexin B2 in two-dimensional organoid cultures. The role of Klf4 during CRC tumor progression, invasion and metastasis, as well as the signaling mechanism downstream of semaphorins leading to Klf4 activation, will be the focus of future investigation in our lab. For the present study, as requested by the reviewer, we have performed a CFU assay subjecting two-dimensional cultures of AKPS organoids to treatment with Klf4 inhibitor and/or rPlxnB2. These results indicate that rPlxnB2 treatment can rescue the reduced CFU seeding ability caused by Klf4 inhibition.

Conversely, treatment of two-dimensional AKPS cultures with the Klf4 inhibitor WX2-43¹ induced a mesenchymal-like phenotype, altering colony morphology, size and actin cytoskeleton, and also reduced the ratio of Ki67⁺ cells proliferation (Extended Data Fig. 7k,l), indicating that Klf4 suppresses mesenchymal traits and promotes proliferation in AKPS organoids, as previously shown for intestinal epithelial cells² and in breast cancer cells³. Klf4 inhibition further reduced CFU of two-dimensional AKPS cultures, which could be rescued by co-treatment with rmPlexin B2 (Extended Data Fig. 7o).

Extended Data Fig 7| o, CFU per well of AKPS organoids treated with 10 uM WX2-43 (Klf4 inhibitor) or 10 uM WX2-43 in combination with 2ug/mL rmPlexin B2.

Upon Klf4 inhibition, we further observed an increase in nuclear Zeb1 in two-dimensional cultures of AKPS organoids. Conversely Zeb1 levels were decreased by treatment with rm Plexin B2, which could also rescue levels in combination with Klf4 inhibition.

Extended Data Fig 8 | c, Zeb1 nuclear intensity in two-dimensional AKPS cultures treated with rmPlexin B2 and/or with Klf4 inhibitor WX2-43. Barplots indicate mean \pm SD. Dots represent nuclear values averaged for individual wells. Ordinary one-way ANOVA, comparing each treatment group to vehicle control.

Independent studies have shown that Klf4 suppresses multiple EMT-Inducing Transcription Factors, including Zeb1⁴. Wang et al further showed that KLF4 knockdown in PDAC cells enhanced ZEB1 expression and gemcitabine resistance while KLF4 overexpression induced the opposite effect⁵. Conversely, the EMT transcription factor Snail was shown to suppress KLF4 in colorectal cancer⁶, suggesting that these master regulators of cell state engage in extensive cross-regulation.

Elf3 and Ghr12 nuclear levels do not significantly vary upon Klf4 inhibition and/or rPlxnb2 treatment in vitro. In general, we observe an elevated epithelialization of AKPS metastases in response to *Plxnb2* overexpression, which is indicated by increased immunoreactivity of markers of epithelial identity such as Klf4, Elf3 and Ghr12. Our multiomic data mechanistically link *Plxnb2* overexpression to Klf4 activation in the nucleus, and absence of Klf4 immunoreactivity in *Sema*^{4KO} metastases epistatically places Klf4 downstream of plexin-semaphorin signaling. We don't have any functional data mechanistically linking the other EMT/MET transcription factors analyzed in our study, namely Elf3, Ghr12, to the observed phenotype. We do not claim these factors drive the phenotype, but rather use them as markers to describe the observed cellular state of AKPS metastases upon perturbing Plexin B2 on hepatocytes: epithelial upon *Plxnb2* overexpression (Klf4, Elf3, and Ghr12 up and Zeb1 down) and mesenchymal upon *Plxnb2* KO or class IV semaphorins KO (Klf4 down, Zeb1 up). We have amended the text as follows:

Further indicating loss of epithelial traits, these lesions lack the expression of Elf3 and Grhl2, two transcription factors that preserve epithelial identity by suppressing EMT⁷⁻¹² (**Extended Data Fig. 8c**).

Cumulatively, our results show that Plexin B2 is a host organ-derived factor required for metastatic seeding in the liver. Plexin B2 promotes epithelialization of metastases by inducing Klf4 in a semaphorin IV-dependent manner.

2. Along these lines, do authors interpret the epithelialization and proliferation phenotypes to be coupled and/or mechanistically linked, and equally relevant for Plexin B2's effects as a gatekeeper of liver metastasis? The data suggest Klf4 is important for the epithelialization phenotype (but not proliferation?) and Rac1 signaling underlies the proliferation (CFU; or also epithelialization?) of CRC cells? The text should be more precise in describing whether and how the different phenotypes and pathways linked to Plexin B2 signaling are interconnected (or not).

In Extended Data Fig. 7n, we show that treatment of AKPS organoids with Klf4 inhibitor significantly decreases proliferation, as assessed by Ki67 staining. Similarly, in *Plxnb2* OE livers, an increase in Klf4⁺ cells coincides with an increase in Ki67⁺ cells. In line with these results, Klf4 has been shown to regulate proliferation of intestinal epithelial cells² and in breast cancer cells³. We now specified this in the text:

Conversely, treatment of two-dimensional AKPS cultures with the Klf4 inhibitor WX2-43¹ induced a mesenchymal-like phenotype, altering colony morphology, size and actin cytoskeleton, and also reduced the ratio of Ki67⁺ cells proliferation (**Extended Data Fig. 7k,l**), indicating that Klf4 suppresses mesenchymal traits and promotes proliferation in AKPS organoids, as previously shown for intestinal epithelial cells² and in breast cancer cells³.

Generally, the link between EMT and proliferation is well studied, and our results corroborate previous studies showing that epithelialization (or MET) induces proliferation^{12,13}.

We have also amended the Discussion as follows:

Our results corroborate these findings, implicating reverse signaling from semaphorins in mediating the seeding-promoting effects of Plexin B2 on tumor cells by upregulating Klf4 and promoting proliferation. This and other studies have ascribed to Klf4 a role as driver of epithelial proliferation^{63,71,102} and Zeb1 antagonist⁶². Yet, a fine mechanistic dissection of the sequence and interconnection of the molecular events downstream of the Plexin B2-semaphorin interaction, including the role of Rac1 signaling and cytoskeletal remodeling, remains to be achieved.

3. Regarding the upregulation and/or activation of Klf4 in tumor cells by Plexin B2-semaphorin signaling: Can authors please clarify if Klf4 expression, or only its predicted transcriptional activity (i.e. expression of genes associated with KLF-motif+ ATAC peaks and/or sensitive to Klf4 inhibition), is induced in the rhPlexin B2-treated organoids that gain epithelial morphogenesis, proliferation, and ultimately, CFU capacity? More broadly, authors should better integrate scRNA and scATAC dimensions of the multi-omics data (as now are mostly analyzed independently) and clarify how well these changes are recapitulated in the in vitro experiments used to dissect mechanism (eg what's the status of Klf4, Zeb1, Grhl2/Elf3 expression in rhPlexin B2- or Rac1/Klf4 inhibitor-treated cells changing epithelial/EMT/actin/proliferation traits?).

We thank the reviewer for the remark and now added a dotplot in Extended Data Fig 7j showing in increase of both Klf4 expression and expression of predicted Klf4 target genes (obtained from the CHEA Transcription Factor Binding Site Profiles¹⁴) in rmPlexin B2 treated AKPS organoids as well as mets growing in *Plxnb2* OE livers.

In line with these results, we detected increased nuclear levels of Krüppel-like factor 4 (Klf4) in AKPS metastases growing in *Plxnb2* OE livers, while it was absent from lesions in *Plxnb2* KO livers (**Fig. 3h**). Moreover, expression of *Klf4* as well as its predicted target genes was increased in metastases growing in *Plxnb2* OE livers, as well as in AKPS organoids treated with rmPlexin B2 (**Extended Data Fig. 7j**).

j, Expression of *Klf4* and *Klf4* targets in AKPS organoids treated with rmPlexin B2, and of AKPS metastases grown in *Plxnb2* OE livers.

Please see above for data addressing the status of *Zeb1*, *Grhl2* and *Elf3* expression upon rmPlexin B2 treatment and *Klf4* inhibition of AKPS organoids.

4. In the last section of the study, the authors show how semaphorins are significantly upregulated in CRC tumor cell states with increased metastasis-seeding capacity (HRC and iCMS3 cells), further supporting the role of Plexin B2 – semaphoring signaling in liver metastasis, and providing a mechanism to explain the differential capacity of distinct tumor cell subpopulations for successfully forming liver metastasis. This is a highly relevant addition of the updated manuscript, yet is not sufficiently well-linked with the rest of the elements studied (i.e. Plexin-OE-induced chromatin changes correlating with epithelialization/metastasis outgrowth, *Klf4* as a putative mediator of such process, etc). To better connect the dots: can authors map HRC and iCMS3 signatures in the scRNA/ATAC-seq data in which a Plexin B2-induced enhanced *KLF4* activity was inferred (Figure 3c, Figure 3f)? and, conversely, are *Klf4* levels and/or its transcriptional output specifically enriched in these HRC and iCMS3 cells having increased Semaphorin levels and metastasis-seeding capacity?

We thank the reviewer for this suggestion. We found that the expression of the *KLF4* gene as well as of *KLF4* target genes was higher in HR and iCMS3 cells, and coincided with cells showing high MET scores and class IV semaphorin expression. Conversely, expression of *Klf4* target genes and the HRC signature is elevated upon rmPlexin B2 treatment and in mets growing in *Plxnb2* OE livers. Of note, the iCMS3 signature is not increased upon rmPlexin B2 treatment, however AKPS organoids are relatively genomically stable and therefore do not model the microsatellite unstable CRC tumors comprised in the iCMS3 signature¹⁵. We now included these analyses in the manuscript as follows:

SEMA4A, *C*, *D* and *G* are also detected in human colon and CRC, and, in two published scRNASeq datasets of human CRC¹⁶ (Extended Data Fig. 9c,d). Interestingly, expression of the semaphorin genes and *KLF4* target genes is significantly upregulated in high-relapse (HR) epithelial cells¹⁷ and in intrinsic consensus molecular subtype 3 (iCMS3) cells¹⁵, but not in *Lgr5*^{high} cells, suggesting that high semaphorin expression marks a subpopulation in the primary tumor with elevated liver metastatic potential (Extended Data Fig. 9d,e,f). Semaphorins also expression coincides with a MET signature and expression of *KLF4* target genes, indicating that the semaphorin-*KLF4* signaling axis is active in HR cells (Extended Data Fig. 9g). Conversely, the core epithelial HR signature (coreHRC) is significantly upregulated in metastatic cells grown in *Plxnb2* OE livers, as well as in AKPS organoids upon treatment with rmPlexin B2 (Extended Data Fig. 9h). Semaphorins expression is unaltered in liver metastases compared to matched primary tumors in two independent scRNASeq datasets of metastatic CRC^{18,19} (Extended Data Fig. 9i). However, in a large cohort of CRC patients, increased expression of *SEMA4A*, *SEMA4C* and

SEMA4D, but not *SEMA4G*, is associated with reduced recurrence-free survival (Extended Data Fig. 9h). Moreover, copy number variation (CNV) analysis in the COAD dataset²⁰ further revealed that *SEMA4A*, *SEMA4C* and *SEMA4D* are commonly found amplified, while *SEMA4G* is often deleted in CRC patients (Extended Data Fig. 9h). Cumulatively, these data support the role of Plexin B2-semaphorin-Klf4 signaling in promoting liver seeding, and might explain the differential capacity of distinct tumor cell subpopulations for successfully forming hepatic metastases.

Extended Data Fig. 9 | Class IV semaphorin expression in murine and human CRC. **d**, Averaged expression of *Sema4a*, *4c*, *4d*, *4g* in scRNASeq datasets of human primary CRC (epithelial cells only, the Samsung dataset includes 25 patients, the KUL dataset includes 5 patients). Cells grouped as high-relapse cells (HRCs), Lgr5 cells or others. **e**, Averaged expression of *SEMA4A*, *SEMA4C*, *SEMA4D*, *SEMA4G* in scRNASeq datasets of human primary CRC in iCMS2 or iCMS3 cells. **f**, Expression of *KLF4* and *KLF4* targets in HR vs. other cells and iCMS3 vs. iCMS2 cells obtained from the Samsung dataset. **g**, Averaged expression of *SEMA4A*, *SEMA4C*, *SEMA4D*, *SEMA4G* projected on epithelial cells of the Samsung dataset blended with expression of *KLF4* target gene or of a MET signature. **h**, Expression of the core HRC signature in AKPS metastases growing in *Plexnb2* OE vs control livers, and in AKPS organoids treated with or without rmPlexin B2. **i**, Averaged expression of *SEMA4A*, *SEMA4C*, *SEMA4D*, *SEMA4G* in epithelial cells from CRC liver metastasis or matched primaries in two independent datasets (Che *et al* n = 6, Wang *et al* n = 5). **j**, Kaplan-Meier analysis (<http://kmplot.com>) of the recurrence free survival (RFS) of CRC patients stratified by *SEMA4A*, *SEMA4C*, *SEMA4D* and *SEMA4G* expression. n = 1211 patients. Pie charts indicate CNV analysis *SEMA4A*, *SEMA4C*, *SEMA4D* and *SEMA4G* in the COAD dataset. n = 290 patients. d,e,h,i Two-tailed unpaired Wilcoxon test.

Additional minor/related points:

- Fig. 1h & Ext Data Fig. 3a-related: Given that metastasis-seeding capacity in CRC (and other cancer types investigated in this work; PDAC, melanoma) are not exclusively determined by genetic/CNV changes, authors should also highlight seeding-regulating factors that are

transcriptionally dysregulated (upregulated/downregulated) in hepatic metastasis (and/or in specific subpopulations with increased metastasis-seeding potential, eg HRC). As of now, the study appears to consider relevant only seeding-regulating factors that are mutated (amplified/deleted; Fig. 1h), making this analysis somehow disconnected from the new/expanded mechanistic data linking Plxn2-mediated effects to transcriptional and chromatin changes of CRC cells. I suggest expanding (or separately analyzing) transcriptionally dysregulated RLs, to also capture non-genetic mechanisms influencing the effects of the seeding-regulating factors identified in their screen.

We thank the reviewer for the remark. Initially, we did not conduct the suggested analysis due to the paucity of datasets comparing the epithelial fraction of primary CRC tumors and their matched liver metastases. We indeed believe that the differences in microenvironments between the colon and the liver renders bulk RNAseq data not amenable to this analysis.

During the first revision, we have mapped semaphorin expression in recently published single cell datasets (Che *et al*, n = 6 patients and Wang *et al*, n = 5 patients) and now used the same datasets to derive LR that are deregulated in liver mets compared to matched primary tumors. Extended Data Figure 2h now features chordplots mapping the interactions between proximal and distal hits with cognate LR that are deregulated (logFC > |0.25|, adjusted P < 0.01, two-sided Wilcoxon test) between epithelial cells in liver metastases vs matched primary tumors.

Top-scoring hits were also predicted to interact with LR whose expression levels were altered in LM compared to matched primary tumors in two independent scRNASeq datasets of metastatic CRC^{18,19}

h, Predicted interactions between top-scoring hits of the screen (proximal hits in red in top plot, distal hits in green in bottom plot) and cognate LR whose expression is altered (logFC > |0.25|, adjusted P < 0.01, two-sided Wilcoxon test) between epithelial cells in liver metastases vs. matched primary tumors in two independent datasets of metastatic CRC (Wang *et al* n = 5¹⁸, Che *et al* n = 6¹⁹).

- Fig. 2c-d related: it would be convenient to indicate in methods (“Intrasplenic injection of tumor cells followed by splenectomy” section) whether the PDAC cells inoculated were derived from 2D cell lines or 3D organoids, as 3D vs 2D cell line-derived single-cell suspensions may not behave similarly. Also, note a reference to the melanoma cell line is currently missing in this method’s section.

The single cell suspensions from the PDAC cell line KPC were derived from two-dimensional cultures. We now specify this in the method section under “Cell lines”. We also added a reference for the melanoma cell line.

KPC cells: *Ptf1a-Cre;Kras^{G12D/+};Trp53^{fllox/+}* (KPC) pancreatic ductal adenocarcinoma cells (B6J syngeneic) were kindly donated by Ilaria Guccini (ETH Zurich) and cultured in two-dimensional cultures in DMEM:F12 supplemented with 10% FBS and 1% PenStrep. Cells were split every 3-5 days and on the day before surgery. Before intrasplenic injection, KPC cells were detached from culture flasks with 1mM EDTA.

Melanoma cells: the D4M-3A B6 mouse melanoma line was generated previously²¹ and was obtained from Merck Millipore and cultured in Advanced DMEM:F12 supplemented with 10% FBS and 1% PenStrep, and 1x Glutamax (Gibco).

- Fig. 3a / Ext data 8-related: rmPlexinB2 effects in organoids. (i) Please provide UMAP colored by conditions to facilitate the reader’s understanding of which cells came from which rmPlexin B2-treated vs untreated APKS cells.

We now provide a UMAP colored by sample in Fig 3a.

a, UMAP of rmPlexin B2-treated AKPS organoids and controls profiled by scRNASeq. Colors of dots indicate treatment (+ or - rmPlexin B2) or distinct transcriptional clusters (0-3). Significantly enriched gene ontology terms for clusters 0 and 1, as well as cluster proportions per sample are shown.

(ii) Please map the expression of KLF4 (if detected) and of the gene program linked to gained KLF-motif+ ATAC-peaks (from Fig. 3 c-g) in the rmPlexin B2-treated vs untreated APKS cells (ideally also in the Klf4 inhibitor experiment). Also, are the epithelialization genes (eg Elf3, Grhl2) and/or HRC signatures induced upon rmPlexinB2 treatment, or is the major effect proliferation (CFU, Edu+, Ki67, etc) in this ex vivo setting?

Klf4 gene expression was captured in around 30% of all cells, and was upregulated in AKPS organoids upon rmPlexinB2 treatment. We also found Klf4 target genes (obtained from the CHEA Transcription Factor Binding Site Profiles¹⁴) to be upregulated upon rmPlexinB2 treatment. In AKPS metastases growing in *PlexinB2* OE livers, *Klf4* capture rate was <10%, however expression of its target genes was upregulated with respect to metastases growing in control livers. We include this analysis in the text.

Moreover, expression of *Klf4* as well as its predicted target genes was increased in metastases growing in *PlexinB2* OE livers, as well as in AKPS organoids treated with rmPlexin B2 (**Extended Data Fig. 7j**).

j, Expression of *Klf4* and *Klf4* targets in AKPS organoids treated with rmPlexin B2, and of AKPS metastases grown in *PLxn2* OE livers.

The HRC core signature (obtained from¹⁷) was also significantly upregulated in rmPlexin B2-treated AKPS organoids and in metastases growing in *Plxn2* OE livers, suggesting that the plexin-semaphorin-*Klf4* signaling axis is active in subset of cells with high liver metastatic potential.

h, Expression of the core HRC signature in AKPS metastases growing in *Plxn2* OE vs control livers, and in AKPS organoids treated with or without rmPlexin B2.

(iii) Can you clarify if *Zeb1* is suppressed?

Unfortunately, epithelialization/EMT transcription factors were very sparsely captured in the scRNASeq, in particular *Zeb1* was captured in <15 cells so our data cannot be used to infer its suppression upon rmPlexin B2 treatment. Our in vitro assay indicates that *Zeb1* is suppressed in rmPlexin B2 treated organoids, while *Elf3* and *Ghr12* do not vary significantly (see above).

- Fig. 3c-f-related: scRNA/ATAC: (i) Distinct subpopulations of cells are observed: Can authors please describe which one(s) specifically show increased accessibility at KLF-motif containing loci? All, or is there a dominant one?

We thank the reviewer for the remark and computed a per-cell motif activity score by running chromVAR²², which allows to visualize motif activities per cell. All subpopulations appear to show KLF4 activity (motif MA0039.4).

(ii) Related to the above point: are any of these subpopulations (scRNA and/or ATAC) reminiscent of HRC or other metastasis CRC-relevant cell states (eg EMT/MET)? (signatures could be mapped here);

We have mapped the signatures but found no particular population that is strongly reminiscent of HRCs nor EMT/MET. It is important to note that these signatures were defined in primary patient tumors and are not necessarily recapitulated in murine AKPS liver mets. As suggested by the reviewer, we have mapped Klf4 targets and semaphorin expression in scRNASeq datasets of primary tumors and have found that their expression coincides with HRCs (see answer to point 4 above).

core HRC score

EMT score

MET score

(iii) As above, does the gain in accessibility at KLF-moti+ loci correlate with an increased expression of Klf4 (and/or other TF whose motifs are differentially enriched in ATAC peaks of compared conditions), or not necessarily?

We did not find an increased expression of *Klf4* transcripts, however increased expression of Klf4 target genes. However, our stainings indicate that there is an increase in nuclear Klf4 protein so the lack of transcript upregulation could be due to low capture rate of the single-nuclei RNA seq or to post-transcriptional regulation of this factor.

- Extended Data Fig. 7j-related: Can authors please clarify if Klf4 expression is also gained in metastasis forming in regular livers (or only in the PlexinB2-OE setting)? If so, this would further support their conclusion that epithelialization (via Klf4?) is generally required for liver metastasis.

Yes, Klf4 is detected also in unperturbed livers (signal in Fig 3h is > 0), however the staining is mostly cytoplasmic. Klf4 immunoreactivity is completely absent in metastases growing in *Plexnb2* KO livers and in *Sema4KO* metastases. Most recently, Tsanov *et al* found that Klf4 knockdown inhibits PDAC liver metastasis²³. Interestingly, Klf4 target genes are upregulated in liver metastases compared to matched primary tumors in the Wang *et al* dataset but not in the Che *et al* dataset. However, limited patient numbers (n = 6 and 5, respectively) and the extensive interpatient heterogeneity curtail the conclusions that can be drawn from these datasets. Nonetheless, cumulatively, these data suggest that Klf4 activity is generally required for the establishment of CRC and PDAC liver mets.

Referee #4 (Remarks to the Author):

The authors have addressed my comments. I have a few points which in my opinion need to be answered before publication of this manuscript.

1. My first question related to the reliability of the mouse model. While I think that it is sufficient for this study, I would like the authors to add a sentence into the manuscript that this model was reported to have some issues and it is important - as for any other study of course - to do the best possible control experiments. To address the reported issues of the mouse model I asked a few questions, which the authors addressed with data that was already in the paper. However, I would like to raise a few issues with these experiments.

We agree with the reviewer that it is important to characterize the mouse models and know their limitations. We performed extensive validation of the method and of the dCas9-SPH mediated overexpression and are confident that it induces efficient and on-target gene activation. As requested by the reviewer, we added the following sentence to the Method section:

LSL-dCas9-SPH mice were obtained in 2019 and initially exhibited spontaneous urination, as also reported on the Jackson Laboratory website. The phenotype disappeared when mice were outcrossed to B6J and crossed with the AlbCre strain.

a, they perform mRNA quantification in Extended Figure 4c with a statistical Two-tailed unpaired t test. The graph however only shows 1 -2 data points per group. To perform a statistical analysis the authors need at least 3 independent data points. Either take away the statistical test or analyse more data points.

We thank the reviewer for the remark and agree that Student t-test cannot be performed with less than 3 replicates, and therefore removed the test.

b, the staining intensity (how was this measured) in e and f has a 10 fold different x axis labeling (1000 - 4000 in e and 100 - 300 in f). Why is this? Again statistics in f with 2 mice used for the sgNT.

We apologize for the inconsistency and now quantify both stainings with the same methodology (only measuring intensity at the cell borders) yielding values in the same order of magnitude.

Extended Data Fig. 4 | Overexpression of *Plxnb2* in murine hepatocytes after AAV-mediated sgRNA transduction. d,e,f, Representative fluorescent micrographs and quantification of Plexin B2 immunoreactivity in

AlbCre;dCas9-SPH mice injected with AAV8-U6-sg*Plxnb2*OE-EF1a-EGFP (sg*Plxnb2* OE, n = 4) or AAV8-U6-sgNT-EF1a-EGFP (sgNT, n = 3), and *AlbCre;Cas9* mice injected with AAV8-U6-sg*Plxnb2*KO-EF1a-EGFP (sg*Plxnb2* KO, n = 4). DAPI used as a nuclear counterstain. Scale bar, 20 μ m. Two-tailed unpaired t-test.

c, It is argued that the mice respond with similar amounts of target induction and *Glu1* is used as an example. However, the fluctuation of *Glu1* induction in Extended Figure 1b is between 2 to 8 fold - clearly a difference. Again sgNT is again only 2 mice and stats are analysed. Please change. We included *Glu1* expression measurements in additional sgNT control. We'd also like to point out that, except for one datapoint, 3 mice show a fairly consistent *Glu1* upregulation of 2- to 4-fold, which is comparable to the *Plxnb2* upregulation shown in Extended Data Figure 4c.

b, *Glu1* gene expression of whole liver extracts, as assessed by qRT-PCR. Two-tailed unpaired t-test. n = 3 or 4 mice per group.

These experiments clearly suggest that expression is variable, which could be independent of the mouse model, but my concern on this model remains and should be discussed as mentioned above. Too many people use animal models which don't do what they are supposed to be doing and it is important to inform the scientific community.

We agree with the reviewer that the degree of overexpression in *AlbCre;SPH* mice could be variable and target gene-specific. However, we support our findings with regards to the effect of Plexin B2 with multiple independent lines of validation that are not based on CRISPR-a and therefore we believe that our screening method in *AlbCre;SPH* mice enables perturbation of cell-cell interactions *in vivo*.

2, The authors perform bulk RNA seq on liver cells transduced with the sgPlxnB2 sgRNA and compare it to the NT ctrl. I couldn't find any info on the experiment. Did the authors just inject their routine cocktail of sgPlxnB2-OE cells at a 0.5% transduction rate? If so, did they sort the cells before isolating RNA for the RNAseq experiment? If the whole liver population was used it would be surprising to see the differences. Nevertheless, why was PlxnB2 not found? Or it is there and I couldn't see it in their DE blot. Could they please amend the methods and also demonstrate PlxnB2 upregulation in the RNAseq data?

The bulk RNA sequencing in Extended Data Figure 6d-f was performed on whole liver extracts of *AlbCre;SPH* mice injected with AAV8-sg*Plxnb2*-OE or AAV8-sgNT (3 per group), so the transduction rate was likely >90%. We now specify this in the figure legend and Method section. Unfortunately in this experiment *Plxnb2* upregulation was not significant in the bulk RNAseq, however we confirmed Plexin B2 upregulation on the same mice by immunofluorescence in Extended Data Fig 4d,e. We therefore are confident of strong upregulation, and rather ascribe the non-significant gene upregulation in the bulkRNASeq to technical issues in library preparation or transcript alignment. Importantly, we detected significant upregulation of *Plxnb2* in hepatocytes in our multiomic single-nucleus RNA sequencing dataset of *AlbCre;SPH* mice injected with AAV8-sg*Plxnb2*OE and AAV8-sgNT and AKPS organoids.

Left: *Plxnb2* expression density in 27,034 hepatocytes isolated from 2 *AlbCre*SPH mice injected with AAV8-sgPlxnb2OE and 2 *AlbCre*SPH mice injected with AAV8-sgNT. UMAP representation. **Right:** *Plxnb2* expression level in hepatocytes of *AlbCre*;SPH mice injected with AAV8-sgPlxnb2OE and AAV8-sgNT. Two-sided Wilcoxon test, n = 2 mice per condition.

As pointed out above, our results regarding Plexin B2's effect on metastasis are extensively validated using orthogonal methods that are independent of SPH-mediated CRISPR activation. Specifically, we show that lentiviral *Plxnb2* overexpression in immortalized hepatocytes increases colony forming units of mouse CRC organoids when co-cultured in vitro. The same is true for both human and mouse organoids when treated with the recombinant Plexin B2 ectodomain. In vivo, we show that while SPH-mediated overexpression increases hepatic metastatic load, Plexin B2 deletion by both Cas9-mediated gene deletion and Cre-mediated conditional knockout in *Plxnb2*^{fllox/fllox} mice almost completely abolishes metastasis formation. Furthermore, KO and KD of the class IV semaphorin genes that are predicted to interact with Plexin B2 in organoids significantly impairs their grafting ability. All in all, we are confident that our screening method in *AlbCre*;SPH mice is able to yield valid hits for downstream characterization.

3, Immunogenicity of slpmCherry: The authors argue that they have used another pancreas Organoid model and looked at seeding in the liver. Firstly, the blots shown in the rebuttal are not explained well and I assume that the left blot shows B6 injection and the right blot injection of slpmCherry organoids into B6 and NXG mice. It looks ok, but again n=2 only per group and I would have preferred to see the seeding of AKAP organoids into B6 and NXG mice as this is what is used in the paper. It could also be that many of the cells were killed by immune cells after transplantation. Has this been checked? Again the immunogenicity aspect should at least be discussed.

We thank the reviewer and apologize for a lack of legends in the plots provided in the rebuttal, which we report below again for the reviewer's convenience.

Left: Number of metastatic lesions per liver section of n = 2 mice injected with transduced KPC cells (parental) and n = 2 KPC cells lentivirally transduced with the sLP-mCherry construct (sLPmC). **Right:** Number of metastatic lesions per liver section of n = 2 B6 mice and n = 2 NXG mice injected with KPC-sLPmCherry cells.

As requested by the reviewer, we added a few sentences to the discussion regarding the adaptability, specificity and immunogenicity of the screen and niche-labeling system *in vivo*.

“Of note, the adaptability of our method to other experimental systems is dependent on efficiency of sgRNA delivery and metastatic load of the model of choice, as the impact of individual sgRNAs can only be determined if a high enough number of perturbed seeding events occur. Moreover, specificity and immunogenicity of the sLP-mCherry niche labeling system should be assessed.”

It is indeed highly likely that most cells - whether they carry the construct or not - die after intrasplenic injection, however to which extent this is due to recognition by the immune system or simply failure to adapt to the *in vivo* conditions is not well understood. Recent studies have begun to unravel how nascent tumors evade the immune system²⁴, however how this is achieved at the secondary site is still largely unknown. Studies suggest that only a minimal proportion of tumor cells that have successfully disseminated from the primary tumor go on to successfully colonize distant organs and form overt metastases²⁵. Our data suggests that Plexin B2 on hepatocytes promotes hepatic seeding of colorectal, pancreatic and skin cancer *in vivo*. While we did not directly test the involvement of the immune system, our *in vitro* experiments indicate that this effect is directly mediated by tumor-hepatocyte interactions. We'd also like to point out that except for the screening experiments, all validation experiments were carried out with AKPS organoids lacking the sLP-mCherry construct (for example Fig 2b and 2e), and therefore we don't think that the presence of the niche-labeling construct did influence our screening results. Finally, corroborating our results with KPC seeding in the liver, and initial characterization by the Malanchi lab, a study by Liu et al found no immunogenic effect introduced by the sLP-mCherry reporter²⁶. An excerpt from the paper is reported below:

“...we also compared the growth rate of transfected and non-transfected ODTs in the prostate. As shown in Supplemental Figure 4E, no significant difference in tumour weights was detected between these two groups. And no significant differences were found in immune cell infiltration including macrophages, neutrophils, B cells, CD4+ T cells and CD8+ T cells (Supplemental Figure 4F,G), suggesting that mCherry-Luc niche-labelling reporter did not introduce profound immunogenic effect at least in local prostatic microenvironment. Collectively, our results suggested that this niche-labelling reporter was capable to vividly reflect both the status of primary tumour and the liver metastasis at different stages.”

4, PlexinB2 induction, sgRNA efficiency:

I assume the authors wanted to write that sgRNA 2 and 3 are mostly enriched in the proximal to distal analysis as only these two sgRNAs show a 2 fold upregulation *in vitro*. I would have preferred to see these experiments *in vivo* as it is an important control. That would have been an additional 12 mice (3/ group). However, even *in vitro* n numbers are only 2 and the authors could have gone through the effort of at least 3 repeats since they haven't done an *in vivo* experiment.

We added another experimental replicate to the analysis and calculated P values with ordinary one-way ANOVA with Tukey's correction for multiple testing.

Plxnb2 expression levels in AML12-SPH cells transfected with individual sgRNAs targeting *Plxnb2* compared to non-targeting, as quantified by qRT-PCR. Barplots indicate the mean and standard deviation of three independent experiments. In every experiment, 3 wells of a 12 well plate were transfected with one of three *Plxnb2*-targeting sgRNAs, or non-targeting control. *Plxnb2* expression in each well was then quantified by qRT-PCR in three replicate measurements. Individual dots indicate the average log-fold change of the three wells transfected with the same sgRNA. The experiment was independently repeated three times.

5, The apoptosis angle is left out, although the authors still say that PlexinB2 enhances survival and show reduced Caspase3 activity. They also say that the reason for the effect in PlexinB2 livers however is through higher proliferation as shown by increased Ki67. How do they explain then the first observation that the metastatic sides are similar in size between wt and Plexin B2 overexpressing cells in the liver? They also say that proliferation is up as shown by increased Ki67 and lower Caspase 3 activity. Caspase 3 has nothing to do with proliferation. Please correct.

We thank the reviewer for the remark, however nowhere in the revised manuscript do we claim that Plexin B2 on hepatocytes increases survival of metastatic cells in the liver. We now show that Plexin B2 promotes proliferation of tumor cells, and have removed any text or data referring to metastasis size quantification. In the first submitted version of the manuscript, our size estimation was based on tumor area in H&E stains and included both epithelial cells as well as the stroma. Our subsequent immunofluorescent analysis of the lesions revealed that AKPS metastases grown in *Plxnb2*-OE livers are mostly epithelial and exhibit little stroma. Conversely, AKPS metastases grown in control livers are constituted by a high percentage of stromal cells (this is evidenced in Figure 3b). Thus, any quantification that does not differentiate between these two compartments would be misleading. Focusing on the epithelial compartment, we now show that Plexin B2 promotes proliferation. For the same reason, we removed the bulk RNA sequencing analysis of microdissected metastases, and replaced it with differential gene expression of subsetted tumor cells obtained from our single-nucleus multiomic dataset (Extended Data Fig 7).

We amended the text referring to the Caspase 3 staining as follows:

Lesions in *Plxnb2* OE livers further harbored higher density of Ki67⁺ epithelial cells, indicating a proliferative advantage *in vivo*, as well as lower cleaved-caspase 3 levels (Fig. 3b, Extended Data Fig. 7c).

References

1. Zhou, Z. *et al.* A novel small-molecule antagonizes PRMT5-mediated KLF4 methylation for targeted therapy. *EBioMedicine* **44**, 98–111 (2019).
2. Yu, T. *et al.* Krüppel-like factor 4 regulates intestinal epithelial cell morphology and polarity. *PLoS One* **7**, e32492 (2012).
3. Yori, J. L., Johnson, E., Zhou, G., Jain, M. K. & Keri, R. A. Krüppel-like Factor 4 Inhibits Epithelial-to-Mesenchymal Transition through Regulation of E-cadherin Gene Expression*. *J. Biol. Chem.* **285**, 16854–16863 (2010).
4. Subbalakshmi, A. R. *et al.* KLF4 Induces Mesenchymal-Epithelial Transition (MET) by Suppressing Multiple EMT-Inducing Transcription Factors. *Cancers* **13**, (2021).
5. Wang, Z. *et al.* Novel crosstalk between KLF4 and ZEB1 regulates gemcitabine resistance in pancreatic ductal adenocarcinoma. *Int. J. Oncol.* **51**, 1239–1248 (2017).
6. De Craene, B. *et al.* The transcription factor snail induces tumor cell invasion through modulation of the epithelial cell differentiation program. *Cancer Res.* **65**, 6237–6244 (2005).
7. Subbalakshmi, A. R. *et al.* The ELF3 transcription factor is associated with an epithelial phenotype and represses epithelial-mesenchymal transition. *J. Biol. Eng.* **17**, 17 (2023).
8. Sengez, B. *et al.* The Transcription Factor Elf3 Is Essential for a Successful Mesenchymal to Epithelial Transition. *Cells* **8**, (2019).
9. Liu, D. *et al.* ELF3 is an antagonist of oncogenic-signalling-induced expression of EMT-TF ZEB1. *Cancer Biol. Ther.* **20**, 90–100 (2019).
10. Yang, Z., Wu, D., Chen, Y., Min, Z. & Quan, Y. GRHL2 inhibits colorectal cancer progression and metastasis via oppressing epithelial-mesenchymal transition. *Cancer Biol. Ther.* **20**, 1195–1205 (2019).
11. Xiang, J., Fu, X., Ran, W. & Wang, Z. Grhl2 reduces invasion and migration through inhibition of TGFβ-induced EMT in gastric cancer. *Oncogenesis* **6**, e284 (2017).
12. Akhmetkaliyev, A., Alibrahim, N., Shafiee, D. & Tulchinsky, E. EMT/MET plasticity in cancer and Go-or-Grow decisions in quiescence: the two sides of the same coin? *Mol. Cancer* **22**, 90 (2023).
13. Tsai, J. H., Donaher, J. L., Murphy, D. A., Chau, S. & Yang, J. Spatiotemporal regulation of

epithelial-mesenchymal transition is essential for squamous cell carcinoma metastasis. *Cancer Cell* **22**, 725–736 (2012).

14. Lachmann, A. *et al.* ChEA: transcription factor regulation inferred from integrating genome-wide ChIP-X experiments. *Bioinformatics* **26**, 2438–2444 (2010).
15. Joanito, I. *et al.* Single-cell and bulk transcriptome sequencing identifies two epithelial tumor cell states and refines the consensus molecular classification of colorectal cancer. *Nat. Genet.* **54**, 963–975 (2022).
16. Lee, H. O., Hong, Y., Etioglu, H. E., Cho, Y. B. & Pomella, V. Lineage-dependent gene expression programs influence the immune landscape of colorectal cancer. *Nature* (2020).
17. Cañellas-Socias, A. *et al.* Metastatic recurrence in colorectal cancer arises from residual EMP1+ cells. *Nature* **611**, 603–613 (2022).
18. Wang, F. *et al.* Single-cell and spatial transcriptome analysis reveals the cellular heterogeneity of liver metastatic colorectal cancer. *Sci Adv* **9**, eadf5464 (2023).
19. Che, L.-H. *et al.* A single-cell atlas of liver metastases of colorectal cancer reveals reprogramming of the tumor microenvironment in response to preoperative chemotherapy. *Cell Discov* **7**, 80 (2021).
20. Cancer Genome Atlas Network. Comprehensive molecular characterization of human colon and rectal cancer. *Nature* **487**, 330–337 (2012).
21. Jenkins, M. H. *et al.* Multiple murine BRAf(V600E) melanoma cell lines with sensitivity to PLX4032. *Pigment Cell Melanoma Res.* **27**, 495–501 (2014).
22. Schep, A. N., Wu, B., Buenrostro, J. D. & Greenleaf, W. J. chromVAR: inferring transcription-factor-associated accessibility from single-cell epigenomic data. *Nat. Methods* **14**, 975–978 (2017).
23. Tsanov, K. M. *et al.* Metastatic site influences driver gene function in pancreatic cancer. *bioRxiv* 2024.03.17.585402 (2024) doi:10.1101/2024.03.17.585402.
24. Goto, N. *et al.* SOX17 enables immune evasion of early colorectal adenomas and cancers. *Nature* (2024) doi:10.1038/s41586-024-07135-3.
25. Celià-Terrassa, T. & Kang, Y. Distinctive properties of metastasis-initiating cells. *Genes Dev.* **30**, 892–908 (2016).
26. Liu, K. *et al.* A novel mouse model for liver metastasis of prostate cancer reveals dynamic tumour-immune cell communication. *Cell Prolif.* **54**, e13056 (2021).

Reviewer Reports on the Second Revision:

Referee #3 (Remarks to the Author):

The authors have addressed all key points. In my opinion, the manuscript provides strong evidence to support the novel approaches, concepts, and mechanistic links connecting Plexin B2 and liver metastasis proposed by the authors, and I support publication in Nature.

A couple of very minor comments regarding Extended Data Fig. 9g:

1) The current red coloring referring to semaphorins can be unclear to the reader as patterns are different between the two panels. The green coloring does show a clear overlap between both KLF4 targets and the MET signature. Please clarify in the legend or double check in case there is any typo so it is clear to the readers.

2) Please include the list of KLF4 target genes (and, in general, of all signatures used in the omics data analyses) in the Supplementary Tables and/or Github link.

Congrats to the authors for this tour-de-force study.

Referee #4 (Remarks to the Author):

The authors have answered my remaining questions and I have no further comments.

Author Rebuttals to Second Revision:

Referees' comments:

Referee #3 (Remarks to the Author):

The authors have addressed all key points. In my opinion, the manuscript provides strong evidence to support the novel approaches, concepts, and mechanistic links connecting Plexin B2 and liver metastasis proposed by the authors, and I support publication in Nature.

A couple of very minor comments regarding Extended Data Fig. 9g:

1) The current red coloring referring to semaphorins can be unclear to the reader as patterns are different between the two panels. The green coloring does show a clear overlap between both KLF4 targets and the MET signature. Please clarify in the legend or double check in case there is any typo so it is clear to the readers.

We thank the reviewer for the remark, and have amended the legend of the figure.

2) Please include the list of KLF4 target genes (and, in general, of all signatures used in the omics data analyses) in the Supplementary Tables and/or Github link.

Gene sets and signatures are now provided in Supplementary Table S3.

Congrats to the authors for this tour-de-force study.

Referee #4 (Remarks to the Author):

The authors have answered my remaining questions and I have no further comments.